# Photoinduced ynamide structural reshuffling and functionalization

Mohana Reddy Mutra [1] & Jeh-Jeng Wang [1,2] ✉

The radical chemistry of ynamides has recently drawn the attention of synthetic organic chemists to the construction of various *N*-heterocyclic compounds. Nevertheless, the ynamide-radical chemistry remains a long-standing challenge for chemists due to its high reactivity, undesirable byproducts, severe inherent regio- and chemoselective problems. Importantly, the ynamide C(sp)-N bond fission remains an unsolved challenge. In this paper, we observe Photoinduced radical trigger regio- and chemoselective ynamide bond fission, structural reshuffling and functionalization of 2-alkynyl-ynamides to prepare synthetically inaccessible/challenging chalcogen-substituted indole derivatives with excellent step/atom economy. The key breakthroughs of this work includes, ynamide bond cleavage, divergent radical precursors, broad scope, easy to handle, larger-scale reactions, generation of multiple bonds (N-C(sp$^2$), C(sp$^2$)-C(sp$^2$), C(sp$^2$)-SO$_2$R/C-SR, and C-I/C-Se/C-H) in a few minutes without photocatalysts, metals, oxidants, additives. Control experiments and $^{13}$C-labeling experiments supporting the conclusion that sulfone radicals contribute to ynamide structural reshuffling processes via a radical pathway.

[1] Department of Medicinal and Applied Chemistry, Kaohsiung Medical University, No. 100, Shih-Chuan 1st Rd, Sanmin District, Kaohsiung City 807, Taiwan. [2] Department of Medical Research, Kaohsiung Medical University Hospital, No. 100, Tzyou 1st Rd, Sanmin District, Kaohsiung City 807, Taiwan. ✉email: jjwang@kmu.edu.tw

Ynamides are privileged alkyne precursors that bind to nitrogen atoms with an electron-withdrawing group, notably enhancing their stability. In addition, the electron-pushing ability of nitrogen atoms can easily polarize alkyne groups and activate triple bonds, thus realizing high regio- and chemoselectivity in organic transformations[1–8]. Well-established strategies for ynamides include transition-metal-catalyzed[1–8] (metal carbene)/Bronsted acid-mediated[1–14] (keteniminium ion) intermediates, which are then trapped by various nucleophiles/electrophiles to yield various difunctionalization products[14,15] or N-heterocyclic compounds; these intermediates have been studied by various research groups (Liu, Hashmi, Ye, Sahoo, Gandon, and numerous other groups)[1–24] (Fig. 1a). In addition, metal/oxidant/photocatalyst-induced radical addition to the α/β-carbons of ynamides leads to a mixture of E/Z isomers in the products (Fig. 1b)[25–28]. Recently, professors Gandon, Sahoo and coworkers observed an intermolecular radical-triggered reactivity of alkynes vs. ynamides in yne-tethered ynamides with sulfur radicals under traditional reaction conditions (Fig. 1c)[29,30]. In 2020, Ye and coworkers reported intramolecular photoredox-catalyzed regioselective ketyl radical addition on the α-carbon of

ynamide in ketyl-ynamide and radical Smiles rearrangement[31]. Despite their advantages, these limited existing radical strategies require expensive metals/photocatalysts, oxidants, longer reaction times, harmful waste production, and a lack of atom economy. Importantly, selective intermolecular radical-triggered ynamide bond fission and structural reshuffling have remained unanswered challenges in ynamide chemistry until now.

Structural reshuffling is a process involving multiple-bond fission and new bond formation for molecular skeleton reassembly[21,32–36]. However, ynamide structural reshuffling has scarcely been reported[21,33,36]. In 2020, Cui and coworkers reported unusual, ionic divergent intramolecular structural reshuffling of ynamides aided by lithium diisopropylamine (LDA) to produce thiete sulfones, while the additional use of 1,3-dimethyl-tetrahydropyrimidin-2(1H)-one (DMPU) altered the process to produce propargyl sulfonamides (Fig. 1d)[35]. Very recently, Li and coworkers reported intramolecular ionic gold-catalyzed, 1,2-N-migration of ynamides via 1,1-carboalkoxylation in an atom-economical synthesis of tetrahydrofuran-fused 1,4-dihydroquinolines (Fig. 1e)[10]. In addition to ynamides, other types of well-known intramolecular ionic path structural rearrangements

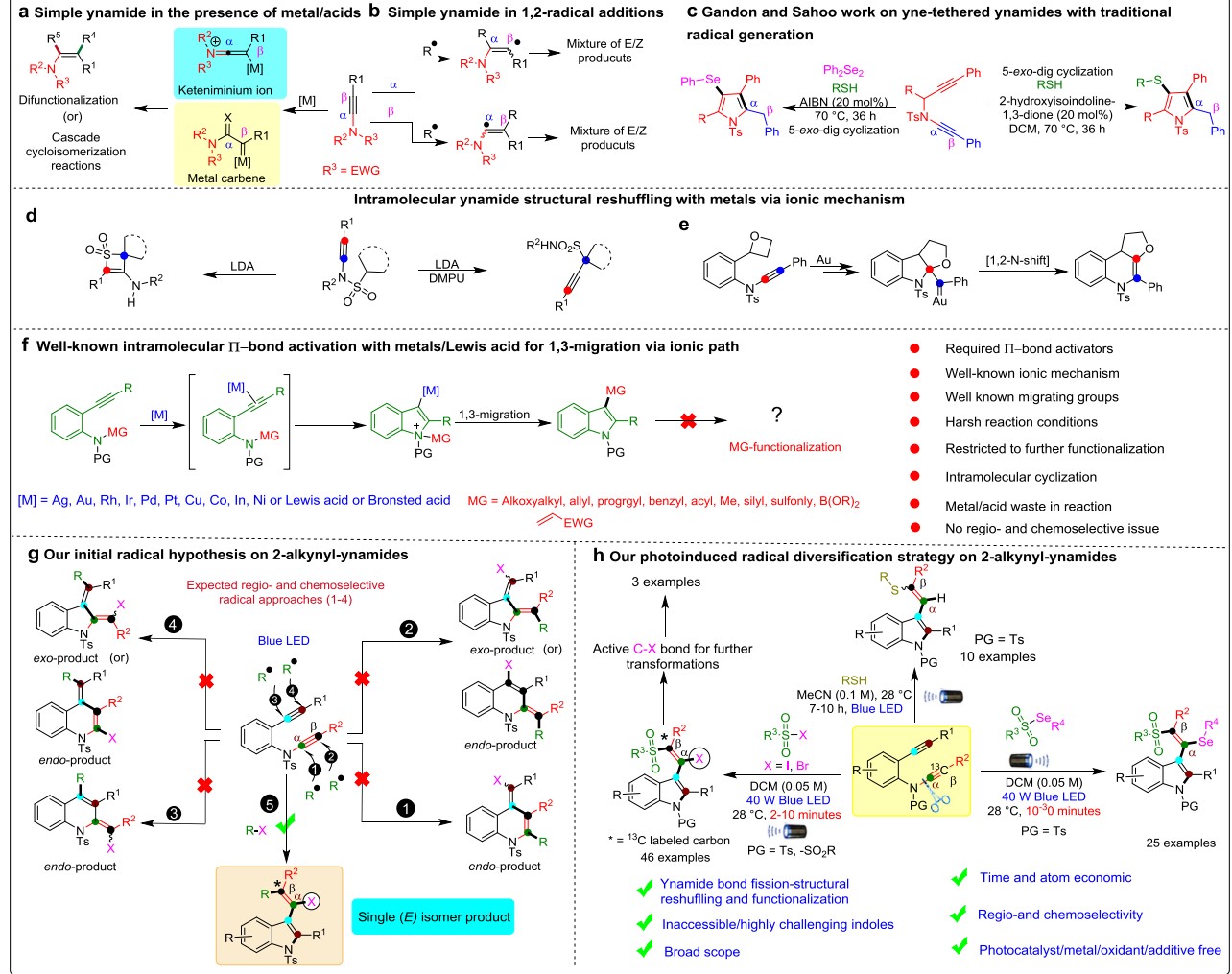

**Fig. 1 Previous literature and background for reaction development. a** Simple ynamide in a metal/Lewis acid. **b** Regioselective radical addition on α,β-carbons. **c** Radical chalcogen-triggered cascade reactions on yne-tethered ynamides. **d** Intramolecular ynamide structural reshuffling with lithium diisopropylamine (LDA) via an ionic pathway. **e** Intramolecular ynamide structural reshuffling with gold-catalyst. **f** Well-known ionic pi-bond activator with a metal/Lewis acid in a skeletal rearrangement of o-alkynylanilines. **g** Our initial hypothesis and the observed ynamide bond fission, structural reshuffling and functionalization processes on 2-alkynyl-ynamides. **h** Photoinduced radical diversification strategy for inaccessible/challenging highly substituted indoles.

**Table 1 Screening of the reaction conditionsa,b.**

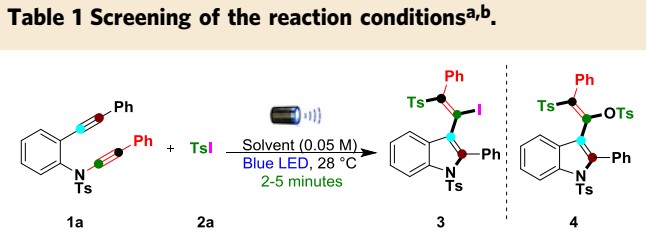

| Entry | Radical initiator | Time (min) | Solvent | Yield 3/4 |
|---|---|---|---|---|
| 1 | Blue LED | 5 | Acetone | 60/Trace |
| 2 | Blue LED | 5 | Toluene | 52/Trace |
| 3 | Blue LED | 5 | THF | 43/Trace |
| 4 | Blue LED | 5 | DCM | 83/Trace |
| 5 | Blue LED | 3 | DCM | 85/– |
| 6 | Blue LED | 5 | H$_2$O | Trace/– |
| 7 | Blue LED | 5 | EtOH | 35/– |
| 8 | Blue LED | 5 | DEC | 58/Trace |
| 9 | Blue LED | 5 | DMSO | NR |
| 10 | Blue LED | 5 | MeCN | 75/Trace |
| 11c | Blue LED | 5 | DCM | 72/<5 |
| 12d | Blue LED | 3 | DCM | 79/Trace |
| 13e | – | 5 | DCM | NR |
| 14f | – | 24 (h) | DCM | 61/<15 |
| 15g | Blue LED | 5 | DCM | 81/– |
| 16h | – | 5 | DCM | –/– |
| 17i | Blue LED | 5 | DCM | 80/– |

*Note*: It is necessary to use freshly prepared 4-methylbenzenesulfonyl iodide in all the reactions. Compound **4** formation can be inhibited by carefully monitoring the reaction time.
aReaction conditions, unless otherwise noted: **1a** (0.10 mmol), **2a** (0.11 mmol), and DCM (0.05 M) were stirred at 28 °C (The fluctuation (depends on local atmosphere) under irradiation with a 40 W Kessil blue LED lamp (Kessil A160WE Tuna Blue controllable LED aquarium light, $\lambda_{max}$ = 462 nm flanked by a second peak at $\lambda$ = 382 nm; more information can be found at Kessil.com) and cooled with a fan, and the reaction mixtures were placed ~8.5 cm from the LED light; 50% intensity of blue light was used.
bIsolated yields.
c0.1 M DCM was used.
d1.5 equiv of **2a** was used.
eStirred at room temperature in the absence of a light source for 5 min.
fStirred at room temperature in the absence of a light source for 24 h.
gStirred under a nitrogen atmosphere.
hStirred at 28 °C under heating in absence of light source.
i40 W PR160L-456 nm blue LED lamp was used (no second emission peak at <400 nm).

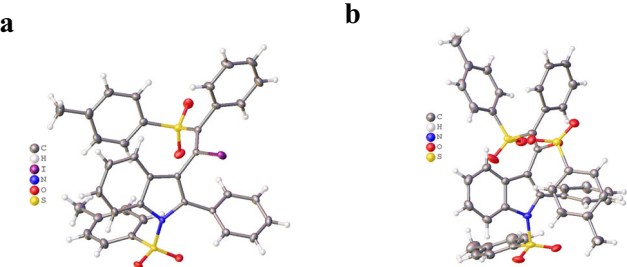

**Fig. 2 Crystal structures of *E* isomers. a** Crystal structure of **3**. **b** Crystal structure of **4**.

mild reaction conditions[47]. Photoinduced organic transformations have potential economic and operational benefits due to their efficacy, commercial availability at a low cost and complementary radical generation compared to traditional metal/oxidants, additives and crucial requirements in chemical transformations[48–53]. We postulated that the generated chalcogen radical triggers either the alkyne or ynamide on 2-alkynyl-ynamides to produce possible regio- and chemoselective products based on the *exo/endo* cyclization mode (Fig. 1g (paths 1–4)).

In this work, we observe radical trigger regio- and chemoselective intermolecular ynamide C(*sp*)–N bond fission, structural reshuffling and functionalization producing synthetically inaccessible/challenging substituted indole derivatives (Fig. 1g, path 5). We are utilizing 2-alkynyl-ynamides with divergent radical precursors to prepare chalcogen-substituted indoles derivatives via the formation of multiple bonds (N–C(*sp*²), C(*sp*²)–C(*sp*²), C(*sp*²)–SO$_2$R/C–SR, and C–I/C–Se/C–H) in a rapid transformation that occurs under mild reaction conditions with excellent step/atom economy (Fig. 1h).

## Results

**Screening of the reaction conditions**. We commenced our photoinduced radical-triggered regio- and chemoselective strategy on ynamides by using 4-methyl-*N*-(phenylethynyl)-*N*-(2-(phenylethynyl)phenyl)benzenesulfonamide (**1a**) with 4-methylbenzenesulfonyl iodide (**2a**) as a model substrate in an acetone solvent under 40 W blue light-emitting diode (LED) light irradiation (Supplementary Figs. 1 and 7). To our delight, we observed radical-triggered ynamide bond fission, structural reshuffling and functionalization to produce a selective single isomer, (*E*)-3-(1-iodo-2-phenyl-2-tosylvinyl)-2-phenyl-1-tosylindole (**3**), in 60% yield and trace amounts of (*E*)-2-phenyl-1-(2-phenyl-1-tosylindol-3-yl)-2-tosylvinyl 4-methylbenzenesulfonate (**4**) (Table 1, entry 1). An extensive solvent-screening process led to these optimized reaction conditions (Table 1, entries 1–10). Such conditions enabled the production of the desired product at maximum yields (85%) with dichloromethane (DCM) solvent within 3 min, without the formation of the compound (*E*)-2-phenyl-1-(2-phenyl-1-tosylindol-3-yl)-2-tosylvinyl 4-methylbenzenesulfonate (**4**). Encouraged by this finding, we altered the solvent molarity ratio from 0.05 to 0.1 M, but the subsequent reaction failed to improve the yields (Table 1, entry 11). Then, the equivalence of compound **2a** was altered, but this failed to improve the yield (Table 1, entry 12). The reported methods revealed the high reactivation of sulfonyl iodides; thus, we carried out the reaction in the absence of a light source, but this reaction did not yield the desired product in a few minutes (Table 1, entry 13). The same reaction continued for 24 h, and compounds **3** and **4** were observed in 61/<15 yields (Table 1, entry 14). Next, we performed the reaction under a N$_2$ atmosphere (Supplementary Figs. 2 and 3), which afforded the

of substituted 2-alkynylanilines have been reported through the utilization of transition metal/Lewis acid-catalyzed cyclization/migration. In this strategy, an alkyne with metal (Pd, Pt, Rh, Ir, Au, Cu, Co)/Lewis acids first activated to induce the reaction, and simultaneous 1,3-migration occurs to produce straightforward indole derivatives (Fig. 1f)[37–40]. Nevertheless, despite the corresponding advances, the above process requires metal/Lewis acids and harsh reaction conditions and follows an intramolecular ionic path, no regio- or chemoselectivity issues are solved, and the reactions are restricted in terms of further functionalization of the migrating group under mild reaction conditions.

Indoles are favorable structural motifs that appear in numerous marketed drugs, the pharmaceutical industry, drug discovery, material chemistry and numerous other fields, including recent therapeutic leads[37–43]. Thus, the construction of inaccessible/challenging indole ring system was a goal for altering the native indole cores, thus enabling access to chalcogen-substituted indoles as potential building blocks for drug discovery in the future. Most importantly, the available active C–I bonds in the products can be used to achieve structural modification of bioactive compounds, drugs and drug leads, as well as natural products[44–46].

We considered the aforementioned challenges and our ongoing research interest in photoinduced radical cascade reactions under

expected desired product in 81% yield (Table 1, entry 15). From these findings, we believe there is no oxygen role in the reaction. The probable reason could be the spontaneous decomposition ability of the weak $-SO_2-I$ bond in 4-methylbenzenesulfonyl iodide to generate an arylsulfonyl radical and iodine radical at room temperature in the absence of light irradiation or any additives[54–59]. The reaction tested with traditional heating at 28 °C (Supplementary Figs. 5 and 6) failed to produce product 3 (Table 1, entry 16) (both starting material visible on TLC).

Next, a blue LED light source which has no second emission peak <400 nm also produced product 3 in 80% yield (Table 1, entry 17) (Supplementary Figs. 4 and 8). According to our observations, the reaction speed and conversion is greatly enhanced by irradiation with visible light; an added advantage is that the reaction also gives a higher final conversion in a shorter reaction time[55,60–64]. Additionally, reaction time monitoring, solvent molarity, and the use of freshly prepared sulfonyl iodide are vital to inhibit compound 4 formation in this rapid transformation. The molecular structures of the products $E$ isomers (Fig. 2), 3 (Fig. 2a), and 4 (Fig. 2b) were unambiguously confirmed by X-ray crystallography (CCDC numbers: 3 (2084458), 4 (2084457)).

**Substrate scope**. With the optimized standard reaction conditions in hand, we focused on the feasibility of the reaction substrate scope as depicted in Fig. 3, a broad range of substituted 2-alkynyl-ynamides (1) were compatible with this transformation to produce the corresponding ($E$)-3-(1-iodo-2-phenyl-2-tosylvinyl)-2-phenyl-1-tosylindole 3–26 with yields ranging from 29% to 88%. Various 2-alkynyl-ynamides 1 (R = Ar) were initially screened, and the reaction produced the desired inaccessible chalcogen-substituted indole derivatives 3–9 in high yields (64–85%) in a few minutes. Long-chain aliphatic and electron-withdrawing groups in the *para* position smoothly tolerated the reaction and produced the desired products. Moreover, this reaction was also carried out with 2-alkynyl-ynamides 1 ($R^1$ = aromatic) with electron-donating groups $p$-Me–Ph, $p$-OMe–Ph, and 3,4-di-OMe–Ph, smoothly producing the desired products (10–12) in efficiently high yields without affecting the functionality. Similarly, 2-alkynyl-ynamides 1 ($R^1$ = aliphatic) with an n-butyl group compatible under standard reaction conditions produced the desired product (13) in good yields (74%). In addition, we were surprised to find that a highly strained cyclopropane ring was readily converted to the desired product in an excellent yield of 82%. This supports the importance of the mild reaction of our divergent radical strategy on the 2-alkynyl-ynamides because cyclopropane is very delicate in the radical homolytic bond fission process. The molecular structure of product 14 was unambiguously confirmed by X-ray crystallography (CCDC number: 14 (2084459)). Importantly, unprotected propan-1-ol gave the desired product (15), albeit in low yield (29%) due to the freely available –OH group in the radical reaction under photoirradiation. Next, various 2-alkynyl-ynamides 1 ($R^2$ = Ar) were studied, and the reactions gave the desired products in moderate to excellent yields of 35–83%. Initially, the electron-donating groups $p$-Me–Ph (16), $p$-ethyl–Ph (17), $m$-OMe–Ph (18) and $p$-OMe–Ph (19) were found to be compatible with the preparation of the desired products in excellent yields (63–83%) without affecting the substituents in the reaction transformation. To our surprise, highly substituted 3,4-5-OMe–Ph (20) produced the desired product with a moderate yield of 47%. Importantly, 2-alkynyl-ynamides 1 ($R^2$ = Ar) with electron-withdrawing substituents $m$-NO$_2$-Ph (21), $p$-COOMe–Ph (22), and $p$-COMe–Ph (23) were compatible with the preparation of the desired products in good yields (42–64%).

Additionally, 2,4-di-Cl-Ph (24) substitutes were also compatible for the preparation of the desired product, which was isolated with a mixture of $E/Z$ isomers (52:48) in moderate yield (43%). The reaction with the product having naphthyl functionality (25) gave a mixture of $E/Z$ isomers (81:19) in moderate yield (55%). Notably, the heterocyclic moiety was smoothly converted to the desired product (26) in low yield (35%) with a mixture of $E/Z$ isomers (76:24), and the probable reason might be the deactivation of the alkyne in ynamide. Next, the $R^2 = -CH_2CH_2Ph$ group was introduced and treated under standard reaction conditions, and the desired product (27) was obtained in moderate yield (59%). This result shows that electronic factors have no effect on the product regioselectivity and chemoselectivity in 2-alkynyl-ynamides. Next, the scope of radical precursor reagents sulfonyl iodides/sulfonyl hydrazides (2) was tested with 2-alkynyl-ynamides 1 to equip the corresponding inaccessible chalcogen-substituted indole derivatives 28–42 with yields ranging from 40% to 90%. Benzene sulfonyl iodide smoothly produced the desired product (28) with an excellent yield of 88%. Due to difficulties in the synthesis/isolation of sulfonyl iodides, a slight modification of the radical precursors was used for the photo-induced radical transformations (sulfonyl iodides were replaced by sulfonyl hydrazides in the presence of oxidizing agents). The reaction proceeded smoothly to give the desired product in the presence of sulfonyl halogens/sulfonyl hydrazide with a series of substituents on the aryl moiety containing electron-donating/drawing groups, such as $p$-OMe–Ph (29) and $p$-$tert$-butyl (30), to give chalcogen-substituted indole derivatives in 71–82% yield. A series of aryl groups with electron-withdrawing groups, $p$-F–Ph (31), $p$-Cl–Ph (32), $p$-Br–Ph (33), $p$-I–Ph (34), $o$-NO$_2$–Ph (35), and $p$-CF$_3$–Ph (36), produced the desired products in good yields of 53–78%. Our reactions were compatible with highly substituted 2,4,5-tri–Cl–Ph sulfonylhydrazine to give desired product 37 in 40% yield. In addition, a series of aliphatic functionalities (methyl (38), ethyl (39)) in compound 2 smoothly generated the desired products in good yields (54–60%). The strained cyclopropane ring was also compatible with the preparation of the corresponding product (40) in excellent yield (77%). The bulky naphthyl (41) moiety smoothly delivered the substituted indole derivatives in excellent yield (90%). Most importantly, the heterocyclic moiety (42) smoothly produced the desired product with a moderate yield of 63%. For all of the above products (except $R^1$ = an aliphatic group), we observe a very broad signal and no sharp peaks (in the case of $R^1$ = an aromatic group, a broad peak is observed at ~7.8 ppm) in the [1]H NMR spectra. We hypothesized that bulky iodine could affect the neighboring aromatic protons so that the orthoprotons may broaden (Supplementary Fig. 9).

To demonstrate the scope and usefulness of our method, we chose to study different starting materials of 2-alkynyl-ynamides (1) and sulfonyl halogens (2), as shown in Fig. 4a. Various $N$-protecting groups (N-SO$_2$Ph (1w), 4-chloro-$N$-(phenylethynyl)-$N$-(2-(phenylethynyl)phenyl)benzenesulfonamide (1x), and N-SO$_2$Et (1y)) with 4-methylbenzenesulfonyl iodide (2a) smoothly afforded desired products 43-45 in 72–77% yield. Next, sulfonyl bromides (2ca) gave the desired products (46 and 47) in 20–52% yield under standard reaction conditions. However, 4-toluenesulfonyl chloride (S17) failed to give the desired product under optimized reaction conditions. The reason for the low reactivity of the sulfonyl bromides/chlorides is due to the relative strength of the sulfone–halogen bond[54–64]. The UV–vis absorption spectra of sulfonyl iodide 2a in various solvents (DCM, MeCN, and THF in $10^{-2}$–$10^{-5}$ M) were measured (Supplementary Figs. 13–24). In more concentrated solutions better absorption in each of these solvents was observed in the blue LED area (our actual reaction condition is even more concentrated than the samples used for

**Fig. 3 Substrate scope for the 2-alkynyl-ynamides, sulfonyl iodide, and sulfonyl hydrazides.** Reaction conditions: **1** (0.10 mmol), **2** (sulfonyl iodides (0.11 mmol)), and DCM (0.05 M) were stirred at 28 °C under irradiation with a 40 W Kessil blue LED lamp (Kessil A160WE Tuna Blue, $\lambda_{max} = 462$ nm) flanked by a second peak at $\lambda = 382$ nm) and cooled with a fan, and reaction vessels were placed ~8.5 cm from the LED light for 2–10 min; isolated yields, a major *E* isomer was formed. [a]Mixtures of *E/Z* isomers were formed (*E/Z* ratios were determined based on indole 4-position aromatic C–H protons in [1]HNMR. Notes: (1) Freshly prepared sulfonyl iodide was used, and the reaction was carefully monitored. (2) Most of the sulfonyl iodides were unstable during synthesis (or) at room temperature. [b]**1** (0.10 mmol), **2** (sulfonyl hydrazide) (1.5 equiv), I$_2$ (0.5 equiv), aq. 70% *tert*-butyl hydroperoxide (TBHP) (3.0 equiv), and DCM (0.05 M) were stirred at 28 °C under irradiation with a 40 W Kessil blue LED lamp (Kessil A160WE Tuna Blue, $\lambda_{max} = 462$ nm flanked by a second peak at $\lambda = 382$ nm) and cooled with a fan, and reaction vessels were placed ~8.5 cm from the LED light for irradiation with a 40 W Kessil blue LED lamp for 10–30 min; isolated yields. [c]4-(*tert*-butyl)benzenesulfonyl iodide was used. [d]Naphthalene-1-sulfonyl iodide was used.

UV–vis measurement). In case of sulfonyl bromide **2ca**, marginal absorption in the blue LED area in various concentrations ($10^{-2}$–$10^{-5}$ M in DCM) was observed (Supplementary Figs. 25–28). Next, we changed the reaction time from minutes to hours under the optimized reaction conditions in Fig. 4b. Sulfonyl iodides (**2b** and **2c**) and sulfonyl hydrazides (2.5 equiv) (**2bc** and **2be**) produced desired products **28** and **30** in a short time under both reaction conditions. In parallel, we performed the

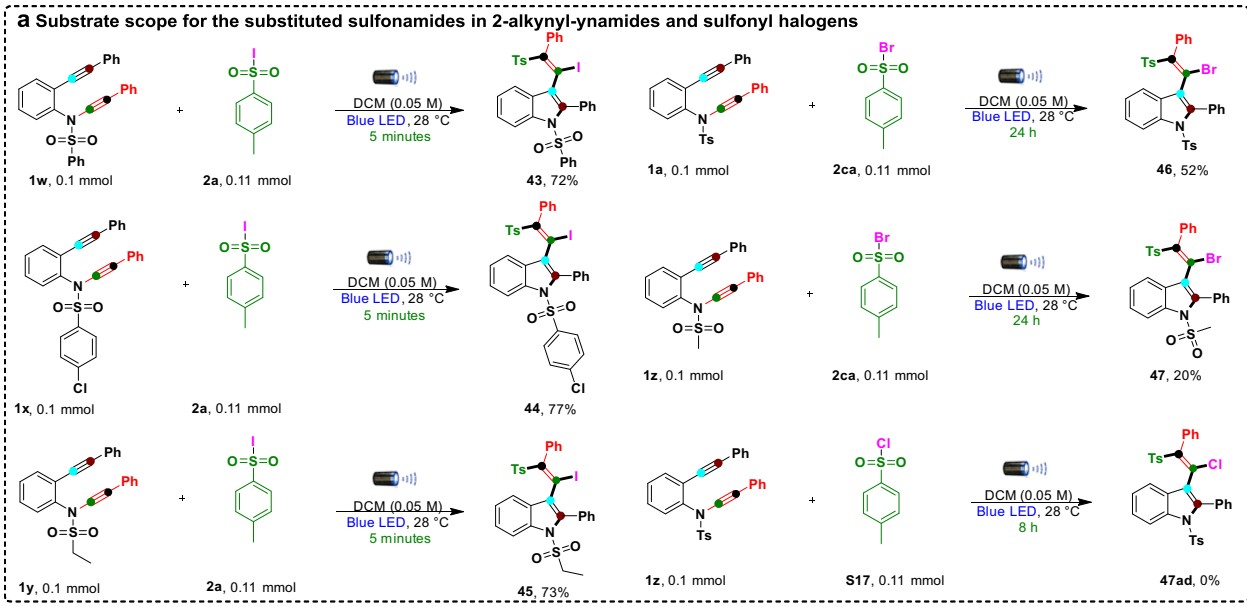

**a** Substrate scope for the substituted sulfonamides in 2-alkynyl-ynamides and sulfonyl halogens

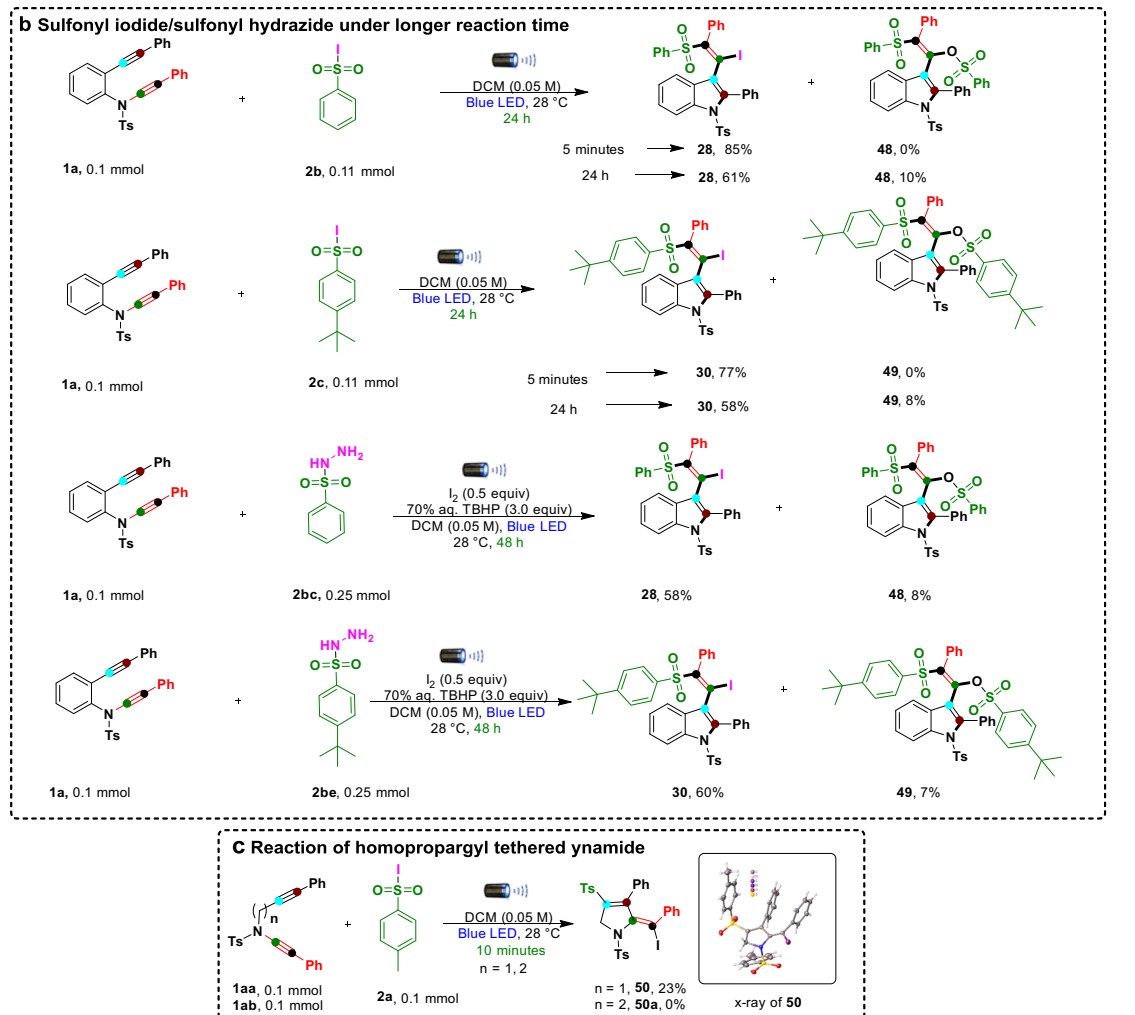

**b** Sulfonyl iodide/sulfonyl hydrazide under longer reaction time

**c** Reaction of homopropargyl tethered ynamide

**Fig. 4 Miscellaneous reactions. a** Substrate scope for the sulfonamides in 2-alkynyl-ynamides and sulfonyl halogens. **b** Reaction with sulfonyl iodides/ sulfonohydrazide. **c** Reaction of propargyl-tethered ynamide and homopropargyl-tethered ynamide with 4-methylbenzenesulfonyl iodide.

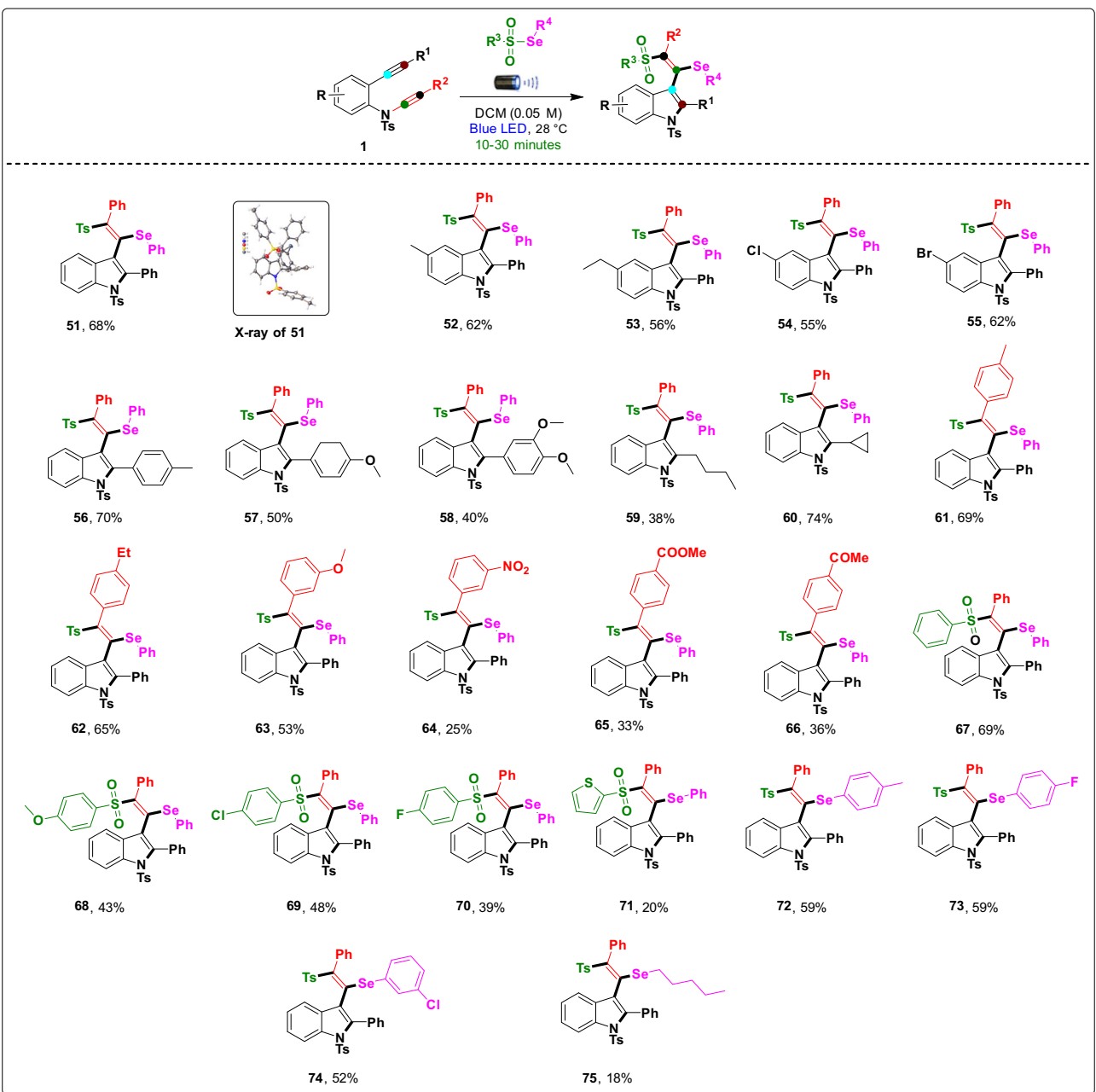

**Fig. 5 Substrate scope of 2-alkynyl-ynamides and areneselenosulfonates.** Reaction conditions: 2-alkynyl-ynamides **1** (0.1 mmol), areneselenosulfonates (0.11 mmol), and DCM (0.05 M) were stirred at 28 °C under irradiation with a 40 W Kessil blue LED lamp (Kessil A160WE Tuna Blue, $\lambda_{max} = 462$ nm flanked by a second peak at $\lambda = 382$ nm) and cooled with a fan, and reaction vessels were placed ~8.5 cm from the LED light for 10–30 min; isolated yields, a major *E* isomer was formed. Note: The reaction was monitored carefully.

same reactions with extended reaction times and observed undesired products **48** and **49** in low yields in addition to the expected major products (**28** and **30**). We hypothesized that the formation of the side products was due to the in situ formation of sulfonic acid from sulfonyl iodides or sulfonyl hydrazides[65]. To validate our hypothesis, we used compound **3** as a starting material and treated it with *p*-toluenesulfonic acid monohydride under optimized reaction conditions. To our surprise, the iodine product (**3**) replaced by the –OTs product (**4**) was formed in moderate yield. Note that when we used 4-toluenesulfonyl bromide as a radical source, product **4** did not form in the reaction due to the bond energy (C$sp^2$–Br) in compound **46** being higher than the C$sp^2$–I bond energy in compound **3** (Supplementary Fig. 10). Next, we extended our methodology by using

propargyl-linked ynamide (**1aa**) under optimized reaction conditions (Fig. 4c) and produced 2-(iodo(phenyl)methylene)-3-phenyl-1,4-ditosyl-2,5-dihydropyrrole as product (**50**) in 23% yield. The molecular structure of product **50** was unambiguously confirmed by X-ray crystallography (CCDC number: **50** (2107787)). However, we did not observe cleavage of the C($sp$)–N bond of ynamide because propargylamide alkyne is more reactive than ynamide[29,30]. The reaction of homopropargyl tethered ynamide (**1ab**) in thin layer chromatography (TLC) showed multiple spots, probably due to the lower stability of starting material **1ab** (spontaneous decomposition in the presence of water (trace amount) within 2–3 days yielded hydration products with ynamide alkynes), and the formed product can easily isomerize into various products[66]. In accordance with these

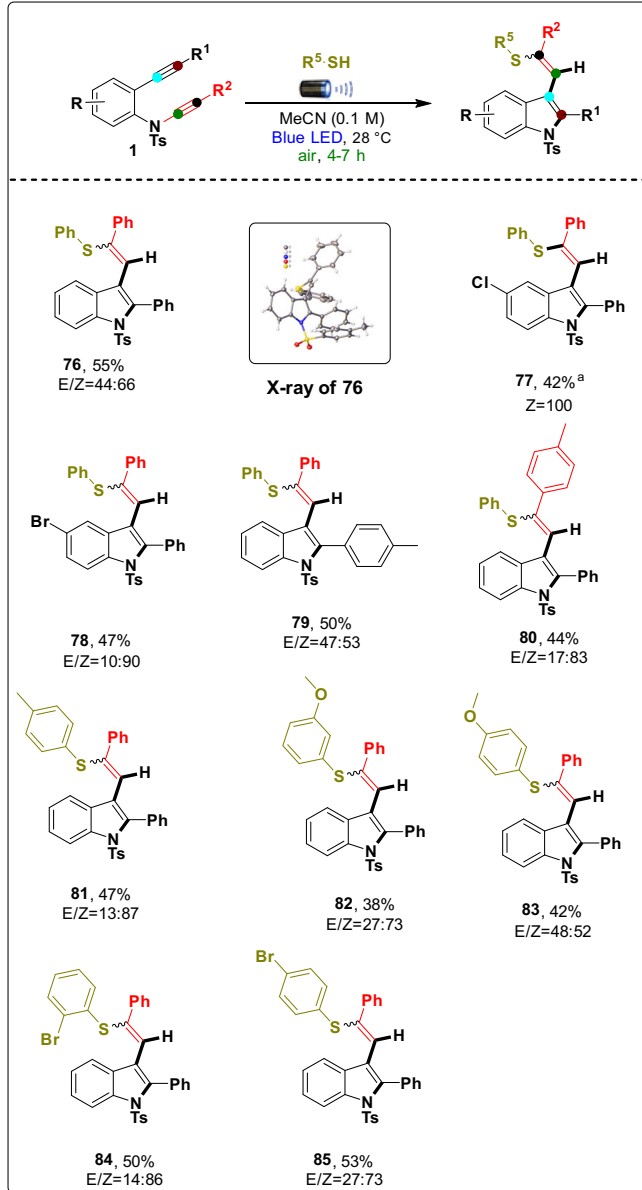

**Fig. 6 Substrate scope of 2-alkynyl-ynamides and aromatic thiols.**
Reaction conditions: 2-alkynyl-ynamides **1** (0.1 mmol), aromatic thiol
(0.25 mmol), and MeCN (0.1 M) were stirred at 28 °C under irradiation
with a 40 W Kessil blue LED for 4–7 h in air; isolated yields of the mixture
of *E* and *Z* isomers were reported, and *E/Z* ratios were determined based on
alkene protons in ¹HNMR. ªZ isomer was formed.

reactions, the aromatic ring between the alkyne and ynamide in
compound **1** is necessary for stability via electron delocalization.

Next, we diversified the radical precursor selenosulfonates
based on our previous work[47]. We focused on the substrate scope
of the reaction, as depicted in Fig. 5. A broad range of substituted
2-alkynyl-ynamides **1** were compatible in the transformation to
equip the corresponding (*E*)-2-phenyl-3-(2-phenyl-1-(phenylse-
lanyl)-2-tosylvinyl)-1-tosylindole derivatives (**51–75**) with yields
ranging from 18% to 74%. 2-Alkynyl-ynamides **1** (R = Ar) with
electron-donating/withdrawing groups –Ph (**51**), *p*-Me–Ph (**52**),
*p*-ethyl–Ph (**53**), *p*-Cl–Ph (**54**), and *p*-Br–Ph (**55**) were investi-
gated first, and the reaction produced the desired inaccessible
selenosulfonate-substituted indole derivatives in good yields of
55–68% in a few minutes. The molecular structure of product **51**

was unambiguously confirmed by X-ray crystallography (CCDC
number: 51 (2084460)). Next, the reaction was used to—test 2-
alkynyl-ynamide **1** (R¹ = aromatic) with electron donating
groups *p*-Me–Ph (**56**), *p*-OMe–Ph (**57**), and 3,4-di-OMe–Ph
(**58**), and the desired products were produced in efficiently good
yields (40–70%) without affecting the functionality. Notably, the
n-butyl group (**59**) and strained cyclopropane (**60**) smoothly
produced the desired products in moderate yields of 38–74%.
Next, we tested various 2-alkynyl-ynamides **1** (R² = Ar) with
electron-donating and electron-withdrawing groups, such as *p*-
Me–Ph (**61**), *p*-ethyl–Ph (**62**), *m*-OMe–Ph (**63**), *m*-NO₂–Ph (**64**),
*p*-COOMe–Ph (**65**), and *p*-COMe–Ph (**66**), which produced the
desired products in low to moderate yields (25–69%). Next, we
used various selenosulfonate radical precursors with electron-
donating/withdrawing groups –Ph (**67**), *p*-OMe–Ph (**68**), *p*-
Cl–Ph (**69**), *p*-F–Ph (**70**) and heterocyclic groups (**71**) compatible
with the sulfone moiety to produce the desired products in
moderate yields (20–69%). Modifications of the selenium moiety
with groups such as *p*-Me–Ph (**72**), *p*-F–Ph (**73**), *m*-Cl–Ph (**74**),
and n-pentyl (**75**) groups smoothly produced the desired
products in 18–59% yield.

Following our previous work[47], we treated 2-alkynyl-ynamides
(**1**) under visible-light irradiation with thiols acting as a radical
precursor (Supplementary Table 1). We focused on the feasibility
of the substrate scope of the reaction, as depicted in Fig. 6. In this
transformation, a series of substituted 2-alkynyl-ynamides (**1**)
were compatible with equipping the corresponding mixture of
(*E/Z*)-2-phenyl-3-(2-phenyl-2-(phenylthio)vinyl)-1-tosylindole
derivatives in low to moderate yields with electron-donating/
withdrawing groups –Ph (**76**), *p*-Cl–Ph (**77**), and *p*-Br–Ph (**78**).
The ratio of the *E* and *Z* mixture in the products changes with the
substituents. In the case of compound **1** (R = Cl or Br), we
exclusively observed the *Z* isomer in product **77** or the major *Z*
isomer in product **78** (*E/Z*, 10:90). We believe that the radical
intermediate (*E*) formed in the mechanism affects the formation
of the single isomer (Supplementary Fig. 54)[67]. The molecular
structure of product **76** was unambiguously confirmed by X-ray
crystallography (CCDC number: **76** (2084461)). Next, the
reaction was also applied to 2-alkynyl-ynamides **1** (R¹ and
R² = aromatic), where the electron-donating groups *p*-Me–Ph
(**79**) and *p*-Me–Ph (**80**) produced the desired products in
moderate yields of 44–50% with a mixture of *E/Z* ratios (47:53
and 17:83). Most importantly, various aromatic thiols containing
electron-donating/electron-withdrawing groups *P*-Me–Ph (**81**),
*m*-OMe–Ph (**82**), *p*-OMe–Ph (**83**), *o*-Br–Ph (**84**), and *p*-Br–Ph
(**85**) produced the desired products in moderate yields of
38–53%. In all cases, we observed the formation of a mixture of
*E/Z* isomers. According to our experimental observations, the
electron-withdrawing group at R gave good selectivity (*Z*)
compared to other substitutes, either in ynamide or thiol
moieties.

**Larger scale synthesis and product synthetic transformations**.
To demonstrate the robustness of our diversified radical strategy
(Fig. 7), we performed larger-scale reactions of 4-methyl-*N*-
(phenylethynyl)-*N*-(2-(phenylethynyl)phenyl)benzenesulfona-
mide **1a** (1.0 g of TsI (**2a**), 0.25 g of Se-phenyl 4-methylbenze-
nesulfonoselenoate (**2da**), and 0.25 g of benzenethiol (**2ae**)) that
underwent smooth transformations to produce the desired pro-
ducts in good yields (**3** (74%), **51** (60%), and **76** (46%)) without
affecting the quantity of the starting material (Fig. 7a). The active
C–I bond in the synthesized products was further transformed
into the respective derivatives, such as *p*-tolylboronic acid (**S26**)
(Suzuki reaction), producing the expected product in a good yield
of 61% (**86**) (eq 1, Fig. 7b). Most importantly, under basic

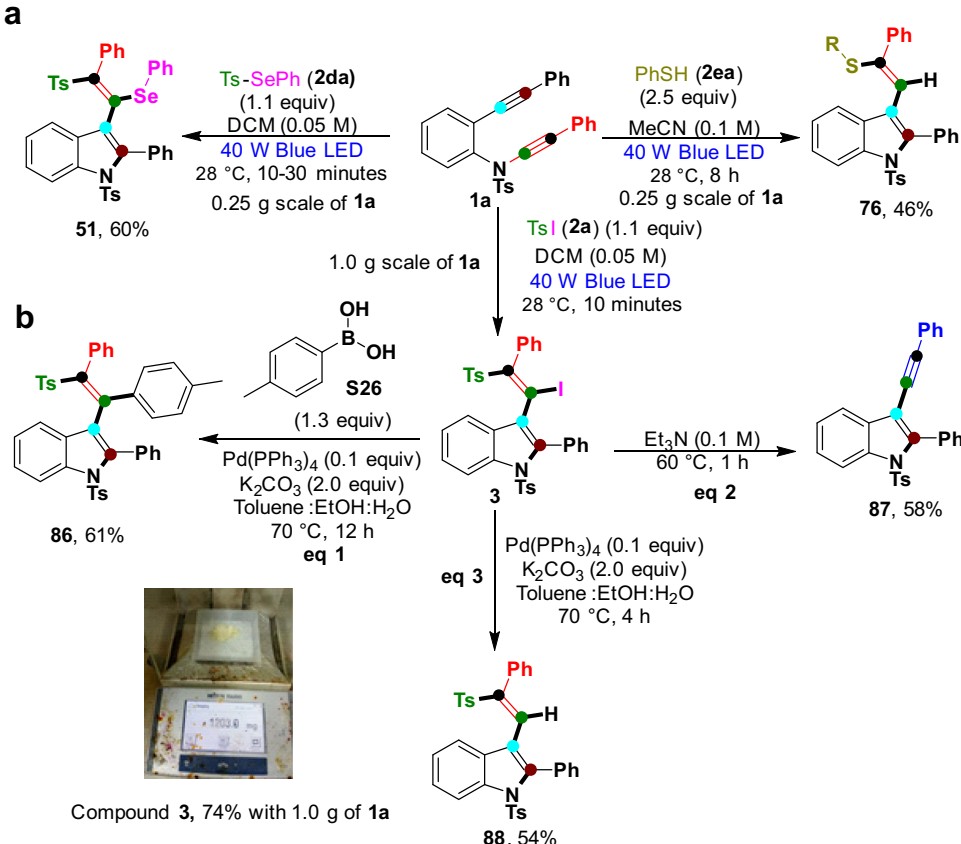

**Fig. 7 Larger scale and product synthetic transformations. a** Reactions on a larger scale by using **1a** with radical precursors **2a**, **2da**, and **2ea**. **b** Synthetic transformations of the product **3**.

reaction conditions, the elimination of sulfone and iodine produced indole 3-substituted alkynes (**87**) in 58% yield (eq 2, Fig. 7b). Next, a deiodination product (**88**) was obtained through palladium catalysis at a moderate yield (54%) (eq 3, Fig. 7b). The products are very interesting but difficult to synthesize in the known protocols reported in the literature, and this synthetic route will be more attractive to chemists working with simple reaction conditions.

**Mechanistic studies.** To gain insights into this reaction mechanism, we conducted several control experiments to form products **3**, **51** in Fig. 8 and **76** in Supplementary Figs. 52–54. First, we conducted radical-trapping experiments using (2,2,6,6-tetramethylpiperidin-1-yl)oxidanyl **89** (TEMPO), ethene-1,1-diyldibenzene (**90**), and butylated hydroxytoluene **91** (BHT) as radical scavengers under standard conditions (Fig. 8a). The TEMPO and ethene-1,1-diyldibenzene radical scavengers completely shut down the desired product formation in the reaction. When we used ethene-1,1-diyldibenzene, sulfone radical-trapping product **92** was observed in 15% yield. The product was confirmed by [1]H nuclear magnetic resonance (NMR), [13]C NMR, and high-resolution mass spectrometry (HRMS) (Supplementary Fig. 49). The above reactions suggest that radical operation is involved in this transformation. However, in the case of the radical scavenger BHT, the desired product was observed in 55% yields with extended reaction times. The probable reason for this result could be the low reactivity of BHT compared to that of 4-methyl-N-(phenylethynyl)-N-(2-(phenylethynyl)phenyl)benzenesulfonamide (**1a**). To determine the importance of the alkynyl group in 2-alkynyl-ynamides (**1**), we performed a reaction with simple

ynamide **93** (Fig. 8b). Surprisingly, this reaction did not produce the expected ynamide bond-fission product (**94**); instead, a regioselective α,β-addition product (**94a**) was observed in 58% yield. The molecular structure of product **94a** was unambiguously confirmed by X-ray crystallography (CCDC number: **94a** (2084462)); this reveals the importance of the 2-alkynyl moiety in 2-alkynyl-ynamides (**1**) for successful transformations. Next, experiments were conducted by using simple ynamide (**93**) and 4-methyl-N-(phenylethynyl)-N-(2-(phenylethynyl)phenyl)benzenesulfonamide (**1a**) to compare the reactivities (ynamide **1a** is more reactive than ynamide **93**) and produce the particular desired product (**3**) at 64% yields (Fig. 8c). We speculated that the 2-alkynyl moiety in 4-methyl-N-(phenylethynyl)-N-(2-(phenylethynyl)phenyl)benzenesulfonamide (**1a**) acts as a directing group in this regio- and chemoselective radical cascade process. Most importantly, to identify the in situ reshuffling groups in these transformations (Ph, alkyne or sulfone), we synthesized [13]C-labeled 4-methyl-N-(phenylethynyl-2-[13]C)-N-(2-(phenylethynyl)phenyl)benzenesulfonamide (**1a′**) and treated it under standard reaction conditions to produce the [13]C-labeled desired products (**3′**) at a higher yield (79%) (Fig. 8d) (Supplementary Fig. 178). This synthesis realizes the first sulfone radical addition to the α-carbon of ynamide and then intramolecular alkyne migration via the expulsion of sulfone radicals and the simultaneous addition of available sulfone radicals to the β [13]C-labeled alkyne. To determine the importance of the electron-withdrawing group in the 2-alkynyl-ynamides (**1**), we replaced –Ts (aromatic) with the –Ms (aliphatic) group (**1z**) and treated it under standard reaction conditions to produce the desired product (**95**) at an excellent yield (80%) (Fig. 8e). The molecular structure of product **95** was unambiguously confirmed by X-ray crystallography (CCDC number: **95** (2084463)). These results

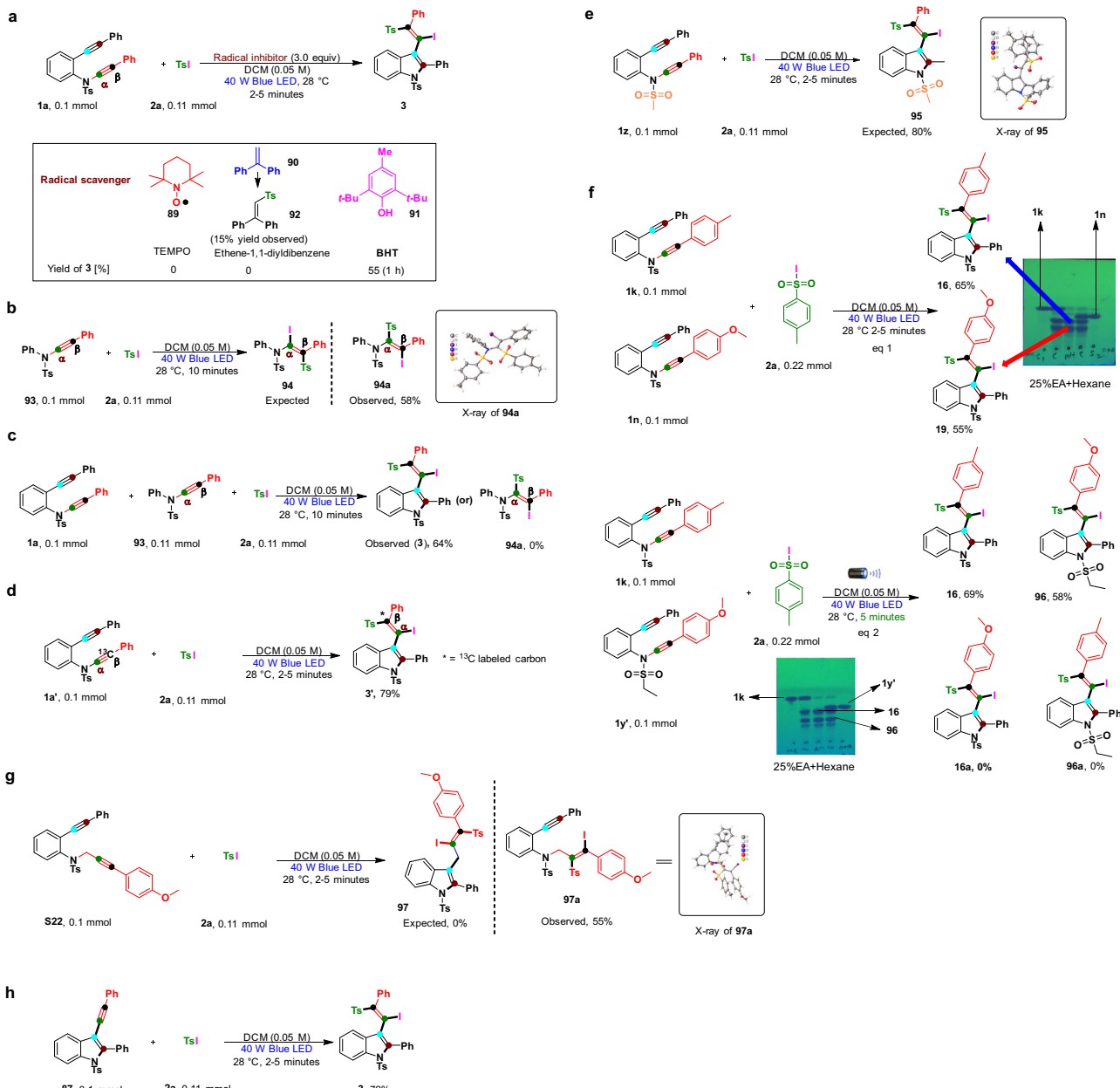

**Fig. 8 Mechanistic studies. a** Radical inhibition experiments with TEMPO, ethene-1,1-diyldibenzene and BHT. **b** To determine the importance of the alkynyl group in 2-alkynyl-ynamides (**1**), we performed a reaction with simple ynamide **93**. **c** One-pot strategy for the reactivity comparison of 2-alkynyl-ynamides **1a** and simple ynamide with **2a**. **d** [13]C-labeled experiment to identify the in situ reshuffling groups in the transformations with help of **1a′** with **2a**. **e** Importance of the electron-withdrawing group in the 2-alkynyl-ynamides by replacing N-Ts with N-Ms in ynamide. **f** Intermolecular or intramolecular alkyne migration was investigated via a crossover experiments. **g** The carbon-linker length effect was tested by using compounds **S22** and **2a**. **h** Reaction tested by using intermediate 2-phenyl-3-(phenylethynyl)-1-tosylindole (**87**) with **2a** under standard condition.

suggest that sulfone derivatives (aliphatic or aromatic) may not be involved in the alkyne migration process, but they will exert a strong influence on ynamide stabilization and isomerization[1–8]. Despite our attempts, we were not able to synthesize other protection groups in compound **1**. The nature of the alkyne migration step was investigated via a crossover experiment with ynamides **1k** and **1n** (eq 1, Fig. 8f). No new crossover products were observed, selectively giving desired products **16** (65%) and **19** (55%). Furthermore, a detailed crossover experiment was performed with ynamides **1k** and **1y′** (eq 2, Fig. 8f), but no crossover products were formed (**16a** and **96a**), selectively giving desired products **16** (69%) and **96** (58%). This step is consistent with previously reported intramolecular migration

processes on *o*-alkynylanilines[37–40]. The carbon-linker length effect was tested by using compounds **S22** and **2a**. To our disappointment, the expected product (**97**) was not observed, and a selective 1,2-addition on propargylamine was observed to produce product **97a** in 55% yield (Supplementary Fig. 181). These results indicate the possibility of isomerization in the 2-alkynyl-ynamides (**1**) (Fig. 8g). The molecular structure of product **97a** was unambiguously confirmed by X-ray crystallography (CCDC number: **97a** (2084593)). In the absence of a radical precursor under standard reaction conditions (Supplementary Fig. 50a), the expected product (**87**) was not formed, which means that light does not induce compound **1a** via cyclization in the first step without a radical source. The sulfone/

**Fig. 9 Plausible reaction mechanism.** Blue LED light initiated ynamide C($sp$)–N bond clevage, intramolecular alkyne reshuffling and functionalization.

thiol radical precursor is vital for this successful transformation. Furthermore, to find the crucial reaction intermediates, under standard conditions, the compounds 4-methyl-$N$-(2-(phenylethynyl)phenyl)benzenesulfonamide (**S8**), phenylacetylene (**S2**), and TsI (**2a**) were treated under standard reaction conditions and expected to yield the desired product (**3**), but the 1,2-addition product **98** (($E$)-1-((2-iodo-2-phenylvinyl)sulfonyl)-4-methylbenzene) was observed in 61% yield with phenylacetylene (Supplementary Fig. 50b). Based on this transformation, 4- methyl-$N$-(2-(phenylethynyl)phenyl)benzenesulfonamide (**S8**) and phenylacetylene (**S2**) are not the crucial reaction intermediates in the reaction. Next, we utilized 4-methyl-$N$-(2-(phenylethynyl)phenyl)benzenesulfonamide (**S8**) under light irradiation, but the expected 2-phenyl-1-tosylindole product (**99**) did not form, confirming that the indole forms via radical addition and ynamide bond fission with a light source (Supplementary Fig. 50c). Ynamide (**1a**) was treated with $I_2$ under standard conditions (Supplementary Fig. 50d), and the expected ($E$)-3-(1,2-diiodo-2-phenylvinyl)-2-phenyl-1-tosyl-1$H$-indole product (**100**) was not observed. Based on this result, iodine does not play a key role in this transformation. Next, we hypothesized that radical addition-ynamide bond fission can produce products **S8** and 1-methyl-4-((phenylethynyl)sulfonyl)benzene (**101**) and promote further recombination of these two fragments, which may provide compound **87** (Supplementary Fig. 50e). Unfortunately, our concept failed due to the absence of radical intermediates (see mechanism intermediates **C** and **D**). To prove homolytic bond cleavage in Ts-C($sp$) (**101**), we chose 2-phenyl-1-tosylindole (**99**) and **101** as starting materials and performed the reaction, but we failed to obtain the expected product (Supplementary Fig. 50f). In addition, we selected the protocol from the literature to prove our hypothesis with **101** ethers as a radical precursor (Supplementary Fig. 50g) under light irradiation[68], delivering expected products (S)-2-(phenylethynyl)tetrahydrofuran (**102**) in 19% yield. Now, this reaction gives a clear idea of homolytic bond fission in Ts-C($sp$). Finally, a key reaction intermediate, 2-phenyl-3-(phenylethynyl)-1-tosylindole

(**87**), was synthesized and treated with **2a** under standard reaction conditions (Fig. 8h). Surprisingly, the key intermediate produced the desired product (**3**) at a good yield of 72%. In addition, we also obtained blue LED emission spectra and UV–vis absorption spectra of various radical precursors (**2a**, **2c**, **2da**, 4-methylbenzenesulfonohydrazide (**2bd**) and **2ea**) (Supplementary Figs. 11–44). The UV–vis absorption spectra show marginal absorption of blue light, which suggests that blue LED light activates the radical precursors by absorption of light. In case of compound 4-methyl-$N$-(phenylethynyl)-$N$-(2-(phenylethynyl)phenyl)benzenesulfonamide (**1a**), UV–vis absorption was not observed (Supplementary Figs. 45–48), which suggests that blue LED light only activate the radical precursors but not ynamides (**1**).

Based on the results of the above control experiment, previous[1–11,37–40,47,54–64,69,70] reports and our own research experience[41], a plausible mechanism is depicted in Fig. 9 for compounds **3′**, **51** and **76** (Supplementary Fig. 55). Blue LED light activate the radical precursor **2a** (TsI) to the excited state **2a′**. Then homolytic bond fission of excited state **2a′** to generate sulfone (**a**) and iodine (**b**) radicals[57]. The generated reactive sulfone radical (**a**) triggers regio- and chemoselectively on the α-carbon of isomerized ynamide intermediate **A**, producing nitrogen-center radical-cation intermediate **B** and then inducing selective 5-$endo$-$dig$ cyclization with an alkyne to generate key reactive intermediate **C**. The nucleophilic β-carbon of ynamide pushes electrons toward nitrogen cations via C($sp$)–N bond fission to generate key intermediate **D**. Simultaneously, intramolecular migration and insertion of 1-methyl-4-((phenylethynyl-2-$^{13}$C)sulfonyl)benzene (**99′**) by discharging sulfone radicals via Ts–C($sp$) homolytic bond fission[68,71,72] generates another key intermediate **E**. Finally, liberated radical sulfone addition to the β-carbon of the alkyne produces vinyl radical intermediate **F**. The available second radical source (iodine radical/selenium) binds with intermediate **F** to produce the final product (**3′/51**).

In summary, we developed a robust approach for photo-induced divergent radical-triggered regio- and chemoselective

ynamide bond fission, skeletal reshuffling, and functionalization of 2-alkynyl-ynamides under mild reaction conditions. These photoinduced radical transformations on ynamides include the ynamide C($sp$)–N bond fission, featuring divergent radical precursors and a very short reaction time. Such transformations are atom economic, have mild reaction conditions and are easy to handle, and their reaction procedures are simple. $E$-isomers are observed in the products of the iodine and selenium derivatives, a mixture of $E/Z$ isomers are observed in the thio derivatives, and a broad substrate scope is achieved. Additionally, there is no need for expensive photocatalysts/metals, oxidants, or additives, and there is high scalability for the synthesis of inaccessible/highly challenging indole derivatives. Moreover, the control experiments and [13]C-labeled studies provide further support regarding the viability of the proposed mechanism. The [13]C-labeled studies revealed that the reaction presumably proceeds in situ with sulfone reshuffling from the α-carbon to the β-carbon in ynamides. Moreover, the products bearing active C–I bonds can be easily converted to essential derivatives, which cannot be achieved through known synthetic methods.

## Methods

**General procedure for the synthesis of ($E$)-3-(1-iodo/bromo-2-phenyl-2-tosylvinyl)-2-phenyl-1-tosylindole derivatives (3, 5-27, 28, 30, 41, 43–47 and 98).** An oven-dried screw-capped, 8 mL vial equipped with a magnetic stir bar was charged with ynamide 1 (0.10 mmol, 1.0 equiv), sulfonyl iodide/sulfonyl bromide (0.11 mmol, 1.1 equiv), and DCM (0.05 M) solvent was added. The resulting solution was stirred up to starting material completion (2–10 min) at 28 °C under a blue LED light (the reaction mixture vial cooled with a fan, and the reaction mixtures were placed ~8.5 cm from the blue LED light). After that, the crude reaction mixture was diluted with water and extracted with DCM. The organic layer was dried over Na$_2$SO$_4$, filtered, and concentrated. The crude material was purified by flash column chromatography to give the corresponding product.

**General procedure for synthesis of ($E$)-3-(1-iodo-2-((4-methoxyphenyl)sulfonyl)-2-phenylvinyl)-2-phenyl-1-tosylindole derivatives (29, 31–40 and 42).** An oven-dried screw-capped 8 mL vial equipped with a magnetic stir bar was charged with ynamide 1 (0.10 mmol, 1.0 equiv) sulfonyl hydrazides (0.15 mmol, 1.5 equiv) in DCM (0.05 M), I$_2$ (0.05 mmol, 0.5 equiv), aq. 70% TBHP (0.30 mmol, 3.0 equiv) was added. The resulting solution was stirred up to starting material completion (30 min) at 28 °C under a blue LED light (the reaction mixture vial was placed ~8.5 cm away from the LED light with a clip fan for cooling). After that, the crude reaction mixture was diluted with water and extracted DCM. The crude material was purified by flash column chromatography to give the corresponding product.

**General procedure for synthesis of ($E$)-2-phenyl-3-(2-phenyl-1-(phenylselanyl)-2-tosylvinyl)-1-tosylindole derivatives (51–75).** An oven-dried screw-capped, 8 mL vial equipped with a magnetic stir bar was charged with ynamide (0.10 mmol, 1.0 equiv), selenosulfonates (0.11 mmol, 1.1 equiv), and DCM (0.05 M) solvent was added. The resulting solution was stirred up to starting material completion (10–30 min) at 28 °C under a blue LED light (the reaction mixture vial cooled with a fan, and the reaction mixtures were placed ~8.5 cm from the blue LED light). After that, the crude reaction mixture was diluted water and extracted with DCM. The organic layer was dried over Na$_2$SO$_4$, filtered, and concentrated. The crude material was purified by flash column chromatography to give the corresponding product.

**General procedure for synthesis of ($E/Z$)-2-phenyl-3-(2-phenyl-2-(phenylthio)vinyl)-1-tosylindole derivatives (76-85).** An oven-dried screw-capped, 8 mL vial equipped with a magnetic stir bar was charged with ynamide (0.10 mmol, 1.0 equiv) aromatic thiols (0.25 mmol, 2.5 equiv), MeCN (0.1 M) was added. The resulting solution was stirred under a blue LED light (the reaction mixture vial cooled with a fan, and the reaction mixtures were placed ~8.5 cm from the blue LED light) up to starting material completion at 28 °C. After that, the crude reaction mixture was diluted with water and extracted with ethyl acetate. The organic layer was dried over Na$_2$SO$_4$, filtered, and concentrated. The crude material was purified by flash column chromatography to give the corresponding product.

## Data availability

All data generated and analyzed during this study are included in this article and its Supplementary Information, and also available from the corresponding author. The X-ray crystallographic coordinates for structures reported in this study have been deposited at the Cambridge Crystallographic Data Center (CCDC), under deposition

numbers CCDC numbers: 3 (2084458), 4 (2084457), 14 (2084459), 50 (2107787), 51 (2084460), 76 (2084461), 94a (2084462), 95 (2084463), 97a (2084593). These data can be obtained free of charge from The Cambridge Crystallographic Data Center via www.ccdc.cam.ac.uk/data_request/cif.

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

## Acknowledgements

The authors gratefully acknowledge funding from the Ministry of Science and Technology (MOST 108-2113-M-037-015-MY3), Taiwan, and the Centre for Research and Development of Kaohsiung Medical University for 400 and 600 MHz NMR, LC–MS.

## Author contributions

M.R.M. conceived, designed the project, performed, analyzed the experimental data and wrote the manuscript. J.J.W. supervised and contributed to the relevant discussions while writing the manuscript.

## Competing interests

The authors declare no competing interests.
