## [Peer Review File · Nature Communications]

REVIEWER COMMENTS

Reviewer #1 (Remarks to the Author):

The manuscript entitled "Photoinduced Radical-Triggered Selective Ynamide Bond-Fission Structural Reshuffling and Functionalization" by Wang and co-worker disclosed photoinduced intermolecular radical diversification of 2-alkynyl-ynamides to substituted indole derivatives. The results presented herein build upon their previous work regarding visible light-promoted alkene vs. alkyne regio- and chemoselective radical cascade cyclization of electronically unbiased 1,6-enynes with chalcogens to synthesize substituted pyrrolidines bearing chalcogens [Green Chem., 2020,22, 2288-2300]. Although synthesis of indole derivatives from 2-alkynylanilines via metal mediated cyclization and subsequent 1,3-migration is known in the literature [for example, see J. Am. Chem. Soc. 2010, 132, 1792–1793; J. Am. Chem. Soc. 2015, 137, 10144–10147; and Org. Lett. 2020, 22, 8550–8554], in this manuscript, the authors disclosed a photoinduced intermolecular radical diversification of 2-alkynyl-ynamides to chalcogens-substituted indole derivatives under metal-free condition which to my knowledge has not been reported. The authors showed remarkable substrate scope with different reactants such as RSO_2I , RSO_2Br , RSO_2SeR , RSH and $\text{RSO}_2\text{NHNH}_2$ in the current manuscript. However, the mechanistic study and the way the entire manuscript is presented lacks the necessary thoroughness and controls to be published in Nature Communications. There are some major issues the authors must solve before the whole work can eventually be assessed on its scientific merits. There might be something interesting there to report but much more work is required and where literature known the necessary references should be included. Underneath detailed comments:

Manuscript:

- 1) The authors presented the desired product **3** formation in 61% yield at room temperature in the absence of a light source for 24 h (see, Table 1, entry 13). Is Blue LED irradiation only reducing reaction time to 5 minutes from 24 h? Is it correct? If yes, authors should justify with a detailed explanation in the manuscript.
- 2) Is it necessary to conduct the reaction under air? Authors should check whether the reaction works in the absence of oxygen by using either nitrogen or argon atmosphere. If oxygen is required for the reaction, then authors should include the oxygen role in the reaction mechanism. In addition, the authors should mention air in the reaction schemes of Table 1 and Scheme 2-4.
- 3) The authors presented twice the absence of a light experiment (entry 12 and 14, Table 1)? Clearly, something is wrong with entries 12 and 14 written by the authors.
- 4) In addition, entry number 4 is written twice in Table 1? Clearly, something is wrong with the numbers given by the authors.
- 5) Product 3a, 3b, 42 and 44 in Scheme 2 should be placed in a new scheme with clearly indicating all the starting materials. Is the product structure 42 and 44 correct? The

IUPAC name indicates the disulfonylated product. Instead of disulfonylation, how did you obtain Ts and OTs? In addition, the missing HRMS data for the products 42 and 44 should be provided.

- 6) To justify RSO₂I absorption of Blue LED absorption, authors should provide absorption spectrum of RSO₂I and the emission spectrum of the Blue LED in the supporting information. In addition, authors should provide absorption spectrum of RSO₂Br, RSO₂SeR, RSH, RSO₂NH₂ and 2-alkynyl-ynamide starting material **1**.
- 7) Did authors use any cooling system for their photoreactor? This information should be clearly mentioned in manuscript and supporting information. By using 34 W lamp, it easily provides more than 35 °C and hence my concern about DCM solvent. What is the distance between the light source and the reaction vial? Authors should provide a clear picture of their photoreaction setup in the supporting information with Blue LED details (emission spectrum and other technical details of Blue LED).
- 8) In Scheme 2, the authors should screen a reaction with 2-alkynyl-ynamides substrate containing R² = alkyl group under the optimized reaction condition.
- 9) Would the reaction work with non-tosyl protecting groups? Would the reaction work with other *N*-protected 2-alkynyl-ynamides substrates for example, *N*-Boc, *N*-Bz or *N*-Piv instead of *N*-Ts, in which the protecting group can easily be removed later in order to obtain the *N*-H products?
- 10) Is there anything known in the literature, regarding excitation of RSO₂I and RSO₂SeR under visible light irradiation? If so, authors should cite all relevant articles in the references.
- 11) The authors underline multiple times that their reaction is environmentally benign, although the reported methods mostly use DCM as solvent in quite dilute conditions.
- 12) Rephrase 'gram-scale reactions' to 'reactions on larger scale', as only one is really gram scale.
- 13) Reference 39 is not present in the manuscript. Reference 40 contains the citation referred to in the manuscript as reference 39? This should be checked for all references.
- 14) Scheme 1, D has been copied wrongly from the initial source (<https://www.nature.com/articles/s41467-020-19467-5.pdf>).
- 15) In all schemes: make sure that PG and EWG are displayed as such, and not as GP or GWE. Examples: Scheme 1,a (EWG); Scheme 1,f (PG).
- 16) The proposed name of **1** does not convey the position of the phenylethynyl group on the *N*-phenyl moiety? Add the position indicator '2'.
- 17) In the manuscript body (page 4, column 1) the text structure has been mixed up.

'To our surprise, the reaction with the naphthyl functionality product produced **25** at the scope of radical precursor reagents sulfonyl halogens **2** were tested with [...]'

18) Appraisals of reaction yields were given inconsistently.

E.g. "Most importantly, the heterocyclic moiety (**41**) smoothly produced the desired product at a **moderate** yield of 63%." and "The active C-I in the synthesized products was further transformed into the respective derivatives, such as boronic acid (Suzuki reaction), to produce the phenyl ring at a **good** yield of 61% (**77**)."

19) The optimization Table is found in the paper itself (Table 1), reference to the SI should be deleted.

"In continuation of this derivatization, we extended the scope of the reaction to consider a higher equivalence of the sulfonyl hydrazide and oxidant and a prolonged reaction time under visible-light irradiation (see the complete optimization in the supporting information)."

20) The complete compound numbering system needs to be checked and corrected in text, schemes and supporting information.

- "Next, we applied various selenosulfate radical precursors bearing electron-donating groups/withdrawing groups, -Ph (**61**), p-OMe-Ph (**62**), p-Cl-Ph (**63**), and p-F-Ph (**64**), in the sulfone moiety and -Ph (**65**) and p-Me-Ph (**66**) in the selenium moiety to produce the desired product in moderate yields of 39%-69%."
- Where is compound number **43**?
- The deiodination product is number **79** in the scheme, not **78** is indicated in the text.
- "Next, a deiodination product (**78**) was obtained through palladium catalysis at a moderate yield (54%). Most importantly, under basic reaction conditions, the elimination of sulfone and iodine produced indole 3-substituted alkynes (**78**) at a 58% yield."
- Place number **78** underneath the expected product in Scheme 4B eq. 5 for clarity.

21) In Scheme 4A, 4-methylbenzene boronic acid was used, while coupled product bears a phenyl group. Correct to the right structure in correspondence with the SI.

22) Attention should be paid to the spelling of toluene. (Scheme 4A)

23) Please write compound numbers consistently in bold.

Supporting information

General comments

- A. The SI should be **carefully checked for English grammar and spelling**. A consistent use of tenses must be followed, and typing mistakes corrected.

Underneath some examples.

- Mass spectra and high-resolution mass spectra (HRMS) **was were** measured using the LTQ Orbitrap XL (Thermo Fischer Scientific) Liquid chromatography–mass spectrometry at National Sun Yat-sen University. (S3)
- Reaction mixture **before** light irradiation (S4)
- Pictures **represent**s the **reaction**s setup under light-irradiation of blue LED source by using compounds **1** and **2** and the progress of the reaction monitored by thin layer chromatography (S4)
- “slowly reaction mixture warming to room temperature.” (S6)
- Then NBS (1.1 equiv) was added in portions**s**. (S5)
- To a stirred solution of sodium sulfite (2 mmol), sodium bicarbonate (2 mmol) and the corresponding sulfonyl chloride (1 mmol) were dissolved in H₂O (S8) → **Sulfonyl chloride was added to a stirred solution of sodium sulfite and sodium bicarbonate.**
- To a dried flask **was-added charged with S7** (1.0 equiv), CuSO₄·5H₂O (0.1 equiv), 1,10-phenanthroline (0.2 equiv) and K₃PO₄ (2.2 equiv) in dry toluene was added bromoalkyne (note: need to prepare freshly before performing reaction) **was** and the mixture was stirred at 80°C 2-12h.
- **In** an oven-dried screw-capped 5 mL vial equipped with a magnetic stir bar was charged with 2-alkynyl-ynamides (1.0 equiv), TsI (1.1 equiv), and DCM (0.05 M) solvent was added. The resulting solution was stirred **utpo** starting material completion at room temperature under blue LED light. After that, the crude reaction mixture **was diluted water** and extracted with DCM (**some times** without extraction directly purified with silica). The organic layer was dried over Na₂SO₄, filtered, and concentrated. The residue was purified by flash chromatography.
- We **commenced (past tense)** our radical strategy by using thiol as a radical precursor with 4-methyl-N-(phenylethynyl)-N-(phenylethynyl)phenyl)benzenesulfonamide (1) in an DCM solvent under 34 W blue light-emitting diode (LED) light irradiation. But, to our disappoint**ment** we **get (present tense)** only trace amount of product 67 (Table S1, entry 1). An extensive solvent screening process **led (past tense)** to these optimized reaction conditions (Table S1, entries 2-6).
Then, the equivalence of compound 2 **was altered** but failed to improve the yield (Table 1, entry, 8). We **carried** out the reaction in the absence of a light source (at room temperature) but this **reaction did** not improve the reaction (Table 1, entry 9). The same reaction **continued** for 24 h, and compounds 3 and 4 were observed in 61/<15 yields (Table 1, entry 13). Thus, we **choose chose** the reaction condition **of** entry 6 for our derivatization.

Preparation of Starting Materials (section 3)

- B. There are **major inconsistencies between the schemes and the text** of section 3, 3.1, 3.3, 3.4, 3.5. These sections should be carefully revised. Problems are not limited to:

- a. Amounts of reagents different in scheme vs text
 - Pd(PPh₃)₄, CuI : experimental of compounds S2 and S6
 - K₂CO₃ : experimental of compound S3, ¹³C-1'
 - TsCl : experimental of compound S7
 - K₃PO₄ : experimental of compound S8
 - Se : experimental of compound 4 (ditolyl diselenide)
 - PIDA : experimental of compound 5 (selenosulfonate)

- b. Wrong reagents in the scheme or the text (pyridine or triethylamine for the synthesis of S7?)
- c. Solvent not mentioned in the text (THF). (compound **S2**)
- d. Amounts not mentioned in the text, only in the scheme (Et₃N) (compound S2)
- e. Typing mistakes: Benzaldehyde (0. 1 equiv.) should be 1 equiv. (compound **S5**, alternative route)
- f. Place either all reaction times in the scheme, or none.
- g. Please add exact purification conditions (heptane/EtOAc ratio) when reporting column chromatography purification.
- h. Please always specify inert gas used to evacuate and refill flasks.

C. Attention to **correct chemical formulation** should be paid:

- Subscripts of CuSO_{4.5}H₂O in scheme of section 3.1
- CuSO₄·5H₂O instead of CuSO₄·5H₂O in experimental for S8 derivatives
- MgSO₄ instead of MgSO₄ in experimental of S5
- Toluene instead of Toluene (many times, among which scheme section 3.5)
- 90 C → 90 °C (3.3 step 1)
- The unit is 'mol%' not 'mole%'.

D. In the Scheme on page S9, there is overlapping text. Also: the second route is also indicated as route-A, and there is no experimental text accompanying this route?

Optimization table (section 4)

E. Table S1 has been placed in section 3.5 (III) by mistake.

Experimental procedures (section 5)

F. Add the gradient used for flash chromatography.

G. Attention to correct chemical formulation should be paid: I₂ instead of I₂

H. Numbering of the compounds is not corresponding with the manuscript.

- General procedure (B) for the synthesis of (*E*)-3-(1-iodo-2-phenyl-2-tosylvinyl)-2-phenyl-1-tosyl-1H-indole derivatives (3) → 3, 5-27. What about the bromo-substituted derivatives (3a and 3b)? Include them in this general procedure ('sulfonyl halides') or make a separate general procedure. Make a separate general procedure for all reactions using sulfonyl hydrazide (28 – 41?)
- General procedure (C) for synthesis of (*E*)-2-phenyl-3-(2-phenyl-1,2-ditosylvinyl)-1-tosyl-1H-indole derivatives (5, 42 and 44) → 5 is not a 1,2-ditosyl compound → 4, 42, 44. Also: take care to account for the oxygen atom issue!
- General procedure (D) for synthesis of (*E*)-2-phenyl-3-(2-phenyl-1-(phenylselanyl)-2-tosylvinyl)-1-tosyl-1H-indole derivatives (45) → 45-66
- General procedure (E) for synthesis of (*E/Z*)-2-phenyl-3-(2-phenyl-2-(phenylthio)vinyl)-1-tosyl-1H-indole derivatives (5) → compound 5 is not a phenylthio derivative!; 67-76

I. The gram scale syntheses are unique procedures, not general procedures. Please add masses and moles of all reactants and reagents, as well as purification details. Use

the correct specific name of the molecules used instead of a general group name, e.g. 'selenosulfonates'. Add reaction times to the experimental procedures and make them correspond to the scheme. Idem for 5.7.

Control studies (section 6)

- J. 6.1: typing mistake on word 'observed' in scheme.
- K. Text underneath scheme 6.2 is copied from 6.1 and wrong for reaction with thiols.

Characterization data:

The reported data should be double checked with the raw data, and corrected in both SI and manuscript if necessary. Data is lacking.

- The title (IUPAC name) of the compounds is not always corresponding to the chemical structures displayed.
- Connect the right compounds to the right general procedure. For all compounds after compound **7**, general procedure A is mentioned, which is clearly wrong.
- Compounds **3a** and **3b** were not synthesized using sulfonyl iodide. Adapt the general procedure.
- HRMS missing for compounds: **3, 26, 28, 42, 44, 51, 61**.
- Compound **43** is never mentioned.
- Compounds missing: **73, 63, 62, 59, 58, 53, 50, 24, 4**
- The mass of pure products **77, 78, 79** is missing. Their yield is different in the manuscript (Scheme 4A).
- The reported yield of **3b** is different in the manuscript (Scheme 2).

NMR Spectra:

The presence of all spectra should be checked again, doubles should be deleted. Furthermore, re-purification of impure isolated products could be considered.

- Some spectra still contain impurities.
 - o E.g. page S66, 68, aromatic impurities in both ^1H as ^{13}C NMR
 - o E.g. page S75, aromatic impurities in ^1H NMR
 - o E.g. page S83, 84, aromatic impurities in both ^1H as ^{13}C NMR
 - o E.g. page S58, 59, aromatic impurities in both ^1H as ^{13}C NMR
- Some spectra are added in duplo, for other compounds one of the spectra is missing.
 - o E.g. page S69 and S70 contain the same spectrum.
 - o E.g. page S81 and 82 contain the same spectrum.
 - o E.g. page S71 and S72 contain the same spectrum, a ^1H spectrum of that compound is missing.
 - o E.g. page S85 and 86 contain the same spectrum, a ^1H spectrum of that compound is missing.
- Please peak pick the solvent signals, especially when used to calibrate the ^{13}C NMR axis.
- Some spectra feature very broad signals. Indicate as 'broad singlet' (brs). E.g. page S89, 91, 226, ...
- In general, peaks are not so 'sharp' and signals are often distorted. Is there a shimming problem with the used NMR?

Reviewer #2 (Remarks to the Author):

Wang and Mutra has presented a useful photoinduced radical-triggered transformations of alkyne tethered ynamides to access synthetically important indole derivatives. The unprecedented regio- and chemoselective ynamide C-N bond fission triggered by photoinduced radical chalcogens, and reshuffling followed by alkyne functionalization provides synthetically challenging indole scaffolds. The direct synthetic method for the preparation of 3-olefin bearing fully functionalized indoles is noteworthy. The reaction was highly efficient, occurs under mild conditions within short reaction time. The transformation makes many new bonds, such as: N-C, C-C, C-SO₂R, C-SR, C-I and C-Se. Important to note that, this reaction does not require expensive metals, photocatalyst, oxidants and additives; only under blue LED and solvent (DCM/CH₃CN), new bonds have been constructed efficiently within a few minutes. Scope of the reaction is very high as various functional group substituted alkyne-ynamides and chalcogens have sustained under the reaction condition. To validate the insights of reaction mechanisms various control experiments have been performed. The site-selective ynamide C-N bond cleavage and in situ sulfone reshuffling on β -carbon of ynamides has been proved with the synthesis of ¹³C-labeled indole product.

Overall, the work shown in this manuscript is expedient and interesting. I, therefore, support this manuscript for publication in Nature Communications after addressing the following queries.

1. To understand the participation of Ts-substituted alkyne (89) by inter or intramolecular fashion, the crossover experiment needs to be conducted with differently substituted alkyne tethered ynamides.
2. Reference numbers 24 and 36 are same.
3. Introduction part; 8th line, "catalyzed1" change to "catalyzed"
4. Show the reaction outcome when alkyl substituted sulfonyl halogens employed in this reaction.
5. Is this reaction appropriate for homopropargyl tethered ynamides with no aryl ring between alkyne and ynamide.
6. Except compound 3b, all the reactions conducted with Ts protected ynamides. Comment about other electron withdrawing groups.
7. Cite the recent work on keteniminium driven functionalization of ynamides (Angew. Chem. Int. Ed. 2020, 59, 10785).
8. Correct spelling of Sahoo instead Shao.
9. Chem. Asian J. 14, 2019, 4282 should be added in ref. 11.
10. Scheme 1a: the keteniminium species should contain both H and M. Modify the Scheme by not only highlighting cycloisomerization but also difunctionalization.
11. GWE should be corrected to EWG in scheme 1a.
12. R is missing in the product of scheme 1c right side.
13. mole% should be corrected to mol% in scheme 1c.
14. scheme 1a, [M] = Metal. or H⁺ should be removed.
15. scheme 1c, space is missing in yne-tethered ynamide.
16. scheme 1e, m of mechanism is merging with the line.
17. scheme 1e, Ts and Au is merging.
18. scheme 1f, GP should be changed to PG.
19. scheme 1f & g, region should be replaced with regio.
20. Sulfonyl bromides and chlorides gave less yields. Any possible explanation?
21. Few examples of various sulfonyl groups of ynamides including alkyl ones should be tried.
22. The reactivity of halogen substituted selenyl compounds along with aliphatic ones should be tried.
23. Suggest the plausible mechanism when thiols are used. Did the authors observe thiol dimerization?

24. Change the deiodination product (79) instead 78.
25. Compound number 80-92 should be bold.
26. The feasibility of Ts-C(SP) bond homolytic cleavage for the conversion of D to E should be supported with few control experiments.
27. Suggest the possible reasons in the attack of Ts radical to b-carbon. As attack at a-carbon could provide stable benzylic radical.

Reviewer #3 (Remarks to the Author):

This manuscript describes a synthetically useful and mechanistically interesting transformation from ynamide precursors under photochemical conditions with cleavage of the C-N bond and recombination to yield a range of indole derivatives. In many cases the addition of the sulfonyl iodide proceeds to form preferentially the E isomer although when thiol is used as the radical source mixture of E and Z isomers are formed. The scope of the transformations has been examined in detail. Following a series of experiments to determine the reaction pathway a reaction mechanism has been proposed.

The authors might clarify why they describe this as 'sulfone reshuffling' – it seems that the sulfonyl iodide adds across an alkyne following a rearrangement to form the indole skeleton.

Overall this work is likely to be of interest to synthetic and medicinal chemists as a route to indole derivatives.

Some detailed comments :

Page 4 Column 1 line 2 2-Phenyl should be 2-phenyl

Page 4 Column 1 line 18 – it is not correct to say 'different alkyl chain groups such as n-butyl groups' as there is only one example. 'a different alkyl chain group which is a n-butyl group' The cyclopropyl substituted derivative is discussed separately in the next sentence.

Page 4 Column 1 line 8 from end – there is some text missing here – the sentence does not make any sense

Page 4 Column 1 line 2 from end – not clear which compound is intended here – in Scheme 3 88% yield is given for 27. In text 82% but not sure if it is the same compound.

Page 4 Column 2 first paragraph – text is inconsistent – is there an effect or not?

Page 4 Column 2 2nd paragraph. The reference to use of sulfonyl hydrazides in place of sulfonyl iodides is not clear – other than the footnote b in Scheme 2 there is no detail on this. In the SI for most of the indoles the experimental says use of general procedure A - should this be B or C depending on whether the sulfonyl iodide or sulfonyl hydrazide and I2 are used? Suggest this is clarified. Should the text read sulfonyl hydrazide and iodine?

Scheme 3 – the structure of the product should contain R1 and R2 in place of Ph

Page 7 first paragraph – not at all clear what this sentence means 'However....stage'

Page 7 first paragraph – n-butyl-Ph – not at all clear to me what this means – is the butyl group on the alkyne, not on a phenyl?

Page 7 for the thiol reactions E and Z mixtures are formed. There is no discussion of the alteration in stereochemical outcome. Also in the SI I could find no information on the E:Z ratio – important to provide this information in the spectral assignment. Does ratio alter with substituents?

Page 7 column 2 rephrase 'See for spectra' to 'For spectra see'

Page 9 the sentence 'This synthesis concludedcarbons' must be rephrased as it is not clear. I am not clear on what is meant by sulfone reshuffling – it seems that the sulfonyl moiety is simply adding across an alkyne after the migration

Page 9 column 1 3 lines from end 'radical beard indole' – what is meant by 'beard'

Conclusions

'exclusively E isomer' – this is not the case with thiol trapping experiments

I have marked these and some further points on the attached PDF.

In the SI in many of the indole products there is a broad signal in the H NMR spectra at ~ 7.8 ppm – it would be helpful if the authors might clarify which protons account for this signal and why this is broadened

A POINT-BY-POINT RESPONSE TO REVIEWER COMMENTS

Manuscript ID: NCOMMS-21-22456-T

Title: Photoinduced Radical-Triggered Selective Ynamide Bond-Fission Structural Reshuffling and Functionalization

Author (s): Mohana Reddy Mutra, Jeh-Jeng Wang

Dear Reviewers,

Thank you very much for your suggestion. We have revised this manuscript according to your comments and suggestions. The corrections in detail were given in the revised manuscript and were highlighted in yellow color. The detailed answers were given for revision was listed as follows:

REVIEWER COMMENTS

Reviewer: 1

The manuscript entitled “Photoinduced Radical-Triggered Selective Ynamide Bond-Fission Structural Reshuffling and Functionalization” by Wang and co-worker disclosed photoinduced intermolecular radical diversification of 2-alkynyl-ynamides to substituted indole derivatives. The results presented herein build upon their previous work regarding visible light-promoted alkene vs. alkyne regio- and chemoselective radical cascade cyclization of electronically unbiased 1,6-enynes with chalcogens to synthesize substituted pyrrolidines bearing chalcogens [Green Chem., 2020, 22, 2288-2300]. Although synthesis of indole derivatives from 2-alkynylanilines via metal mediated cyclization and subsequent 1,3-migration is known in the literature [for example, see J. Am. Chem. Soc. 2010, 132, 1792–1793; J. Am. Chem.Soc. 2015, 137, 10144–10147; and Org. Lett. 2020, 22, 8550–8554], in this manuscript, the authors disclosed a photoinduced intermolecular radical diversification of 2-alkynyl-ynamides to chalcogens-substituted indole derivatives under metal-free condition which to my knowledge has not been reported. The authors showed remarkable substrate scope with different reactants such as RSO_2I , RSO_2Br , RSO_2SeR , RSH and $\text{RSO}_2\text{NHNH}_2$ in the current manuscript. However, the mechanistic study and the way the entire manuscript is presented lacks the necessary thoroughness and controls to be published in Nature Communications. There are some major issues the authors must solve before the whole work can eventually be assessed on its scientific merits. There might be something interesting there to report but much more work is required and where literature known the necessary references should be included. Underneath detailed comments:

Answer: Thank you very much to the reviewer for taking the time and effort to review our manuscript and gave positive comments and valuable suggestions on our work to improve the quality and quantity of the work. In our reaction, we used 40 W Kessil A160WE Controllable LED Aquarium Light (in the previous version of the manuscript we have given 34 W, We are extremely sorry for this mistake) According to your suggestion, we conducted additional control experiments to support our hypothesized reaction mechanism and improved the presentation of the manuscript in the revised version. In addition, we also added necessary references in the revised manuscript.

Manuscript:

1). The authors presented the desired product 3 formation in 61% yield at room temperature in the absence of a light source for 24 h (see, Table 1, entry 13). Is Blue LED irradiation only reducing reaction time to 5 minutes from 24 h? Is it correct? If yes, authors should justify with a detailed explanation in the manuscript.

Answer: We are thankful for your query. In the Table 1, the radical precursor (4-methylbenzenesulfonyl iodide) in the absence of light source also produced the desired product in 61% yield. The probable reason could be the spontaneous decomposition ability of the weak -SO₂-I bond in the 4-methylbenzenesulfonyl iodide to generate an arylsulfonyl radical and a iodine radical at room temperature in absence of light irradiation or any additives. The reaction speed and conversion is greatly enhanced by irradiation with visible light to initiate the radical reaction; an added advantage is that the reaction also gives a higher final conversion in short reaction time. The related references (56-61) were cited in the revised manuscript.

2). Is it necessary to conduct the reaction under air? Authors should check whether the reaction works in the absence of oxygen by using either nitrogen or argon atmosphere. If oxygen is required for the reaction, then authors should include the oxygen role in the reaction mechanism. In addition, the authors should mention air in the reaction schemes of Table 1 and Scheme 2-4.

Answer: We are thankful for your query and suggestions. As per your query, we performed the reaction under N₂ atmosphere by using 4-methyl-*N*-(phenylethynyl)-*N*-(2-(phenylethynyl)phenyl)benzenesulfonamide with 4-methylbenzenesulfonyl iodide. In this reaction transformation, under inert atmosphere also afford desired product with 81% yield. With this reaction, we believe there is no key role of oxygen in the reaction transformation with 4-methylbenzenesulfonyl iodide as radical precursor. The complete handling of the reaction setup as shown below and included in the supporting information for your kind reference.

Supplementary Figure 2: Pictures represent the N₂ atmosphere reaction setup under light-irradiation of blue LED source by using compounds **1a** and **2a** and the progress of the reaction monitored by thin-layer chromatography.

Next, we performed the reaction by using 4-methyl-*N*-(phenylethynyl)-*N*-(2-(phenylethynyl)phenyl)benzenesulfonamide with 4-methylbenzenesulfonyl iodide under N₂ atmosphere at room temperature in absence of light source, to our surprise this transformation also afford desired products 46% and 10% after 24 h. With this reaction support, we hypothesis when the reaction preformed at room temperature there was little influence air in the homolytic cleavage of the radical precursor. The complete handling of the reaction setup as shown below included in the supporting information for your reference.

Supplementary Figure 3: Pictures represent the N₂ atmosphere reaction setup at room temperature using compounds **1a** and **2a** and the progress of the reaction monitored by thin-layer chromatography.

To know in details role of the air in the reaction, we also carried the reaction with other radical precursors such as Se-phenyl 4-methylbenzenesulfonoselenoate, 4-methylbenzenesulfonyl bromide and benzenethiol. In these three radical precursors only benzenethiol delivered the desired product in 49% followed by 4-methylbenzenesulfonyl bromide in 9% but in the presence of Se-phenyl 4-methylbenzenesulfonoselenoate the reaction failed to delivered the product. From all these reaction transformations, we believe light play a crucial role in the radicals generation, followed by radical cascades. In case of thiol as a radical precursor the reaction working at room temperature because air involve in the radical generation.

3). The authors presented twice the absence of a light experiment (entry 12 and 14, Table 1)? Clearly, something is wrong with entries 12 and 14 written by the authors.

Answer: We are thankful for your careful observation. We corrected in the revised manuscript and highlighted with yellow color.

4). In addition, entry number 4 is written twice in Table 1? Clearly, something is wrong with the numbers given by the authors.

Answer: We are thankful for your careful observation. We corrected in the revised manuscript and highlighted with yellow color.

5). Product 3a, 3b, 42 and 44 in Scheme 2 should be placed in a new scheme with clearly indicating all the starting materials. Is the product structure 42 and 44 correct? The IUPAC name indicates the disulfonylated product. Instead of disulfonylation, how did you obtain Ts and OTs? In addition, the missing HRMS data for the products 42 and 44 should be provided.

Answer: We are thankful for your suggestion. We have placed these compounds in **Fig. 4**. Yes, the product structures are corrected and IUPAC names are revised in the manuscript as well as in supporting information. We have carried out reaction with sulfonyl iodide derivatives and sulfonyl hydrazine derivatives as shown below. The reaction in short reaction time afford desired iodine products in short reaction time either different reaction conditions. Next, another batch of the same reactions we carried to longer reaction to check the outcome of the product, when we continued to longer reaction time we were observed formation of minor product of OTs along with major iodine products. Based on this reaction we hypothesized, a trace amount insitu

formation of TsOH involve in the reaction via nucleophilic substitution. To prove our hypothesis we have we choose compound 3 as starting material with TsOH in our standard reaction condition. In this case we observed the formation of product 4 in 44% yields, along with a small amount of unreacted compound 3 in the reaction as shown below (The progress of the reaction checked on TLC after 24 h). From these results, we believe the oxygen atom in the products may come from insitu generated TsOH in the reaction..

6). To justify RSO₂I absorption of Blue LED absorption, authors should provide absorption spectrum of RSO₂I and the emission spectrum of the Blue LED in the supporting information. In addition, authors should provide absorption spectrum of RSO₂Br, RSO₂SeR, RSH, RSO₂NHNH₂ and 2-alkynyl-ynamide starting material 1.

Answer: We are thankful for your query. Emission spectra were measured using Ocean Optics USB 2000+ Spectrometer. Spectra were normalized to 1.0 at the emission maximum. This emission spectra was provided by Miss Angela Liou, Sales Specialist, DiCon Fiberoptics & DiCon Lighting, aliou@diconfiberoptics.com,

Kaohsiung, Taiwan, DiCon Brands - Kessil | Fiilex | Cielux. We sincerely thanks to her support.

In addition to Blue LED emission spectra, we have provided the absorption spectra of the compounds TsBr, Se-phenyl 4-methylbenzenesulfonoselenoate, 4-methyl-*N*-(phenylethynyl)-*N*-(2-(phenylethynyl)phenyl)benzenesulfonamide, benzenethiol as shown below and also included in the supporting information. But, there is no light absorption in that specific region of light in compounds absorption spectra. Thus, we have modified the reaction mechanism according to experimental results obtained from the absorption spectra. In this case, the light source enhances the homolytic bond fission without excitation state. We have literature evidence, the sulfonyl halogens bond cleavage enhances under light irradiation and the related references shown below and also cited in the revised manuscript and highlighted.

Blue LED Emission Spectra

Supplementary Figure 6. Emission spectrum from a 40 W Kessil A160WE Tuna Blue LED shown as blue color line with emission maximum at $\lambda_{\text{max}} = 462$ nm flanked by a second peak at $\lambda = 382$ nm.

Supplementary Figure 7. Absorption spectra of 4-methyl-N-(phenylethynyl)-N-(2-(phenylethynyl)phenyl)benzenesulfonamide (10^{-5} M) in DCM, and blue LED emission.

Supplementary Figure 8. Absorption spectra of 4-methylbenzenesulfonyl iodide (10^{-5} M) in DCM, and blue LED emission. Inset: zoom spectrum zone.

Supplementary Figure 9. Absorption spectra of 4-methylbenzenesulfonyl bromide (10^{-5} M) in DCM, and blue LED emission. Inset: zoom spectrum zone.

Supplementary Figure 10.. Absorption spectra of Se-phenyl 4-methylbenzenesulfonoselenoate (10^{-5} M) in DCM, and blue LED emission. Inset: zoom spectrum zone.

Supplementary Figure 11. Absorption spectra of 4-methylbenzenesulfonylhydrazide (10^{-5} M) in DCM, and blue LED emission. Inset: zoom spectrum zone.

Supplementary Figure 12. Absorption spectra of benzenethiol (10^{-5} M) in DCM, and blue LED emission. Inset: zoom spectrum zone.

7). Did authors use any cooling system for their photoreactor? This information should be clearly mentioned in manuscript and supporting information. By using 34 W lamp, it easily provides more than 35°C and hence my concern about DCM solvent. What is the distance between the light source and the reaction vial? Authors should provide a clear picture of their photoreaction setup in the supporting information with Blue LED details (emission spectrum and other technical details of Blue LED).

Answer: We are thankful for your query. Yes, we used clip fan as a cooling system (our laboratory have air condition $\approx 25^{\circ}\text{C}$, we believe, this also minor influence in the cooling system). The distance between the reaction mixture vial and light source is 8.5 cm and the height is 1 cm as shown below. The complete reaction setup and emission spectrum of blue LED and other technical details of the Blue LED provided as shown below and also included in the supporting information.

Supplementary Figure 4: Pictures represent the complete reaction setup and other technical details of the Kessil A160WE Controllable LED Aquarium Light

You can go through the below links for further details.

(https://www.kessil.com/aquarium/saltwater_A160.php

https://www.kessil.com/support/downloadfiles/aquarium/A160WE_UserManual.pdf (user manual))

8). In Scheme 2, the authors should screen a reaction with 2-alkynyl-ynamides substrate containing $R^2 = \text{alkyl}$ group under the optimized reaction condition.

Answer: We are thankful for your suggestion. As for reviewer suggestions, we have synthesized the starting material (4-methyl-*N*-(4-phenylbut-1-yn-1-yl)-*N*-(2-(phenylethynyl)phenyl)benzenesulfonamide) and screened under standard reaction condition, afforded the desired product in 59% yield as shown below.

In addition to above example, we also tried to synthesize few more 2-alkynyl-ynamides substrate containing R^2 = alkyl groups but unfortunately we are failed to synthesize respective starting material. In most of the below transformations, we observed indole product formation due to the less reactivity of the aliphatic bromoalkyne in the ynamide synthesis. To avoid indole formation in the reaction, we choose the *N*-(2-iodophenyl)-4-methylbenzenesulfonamide but this reaction showed major starting material in TLC.

9). Would the reaction work with non-tosyl protecting groups? Would the reaction work with other *N*-protected 2-alkynyl-ynamides substrates for example, *N*-Boc, *N*-Bz or *N*-Piv instead of *N*-Ts, in which the protecting group can easily be removed later in order to obtain the *N*-H products?

Answer: We are thankful for your query and suggestions. As for reviewer suggestions, we tried to synthesize other protecting groups in 2-alkynyl-ynamides, such as *N*-Boc, *N*-Bz, *N*-Bn or *N*-Piv as shown below. But

unfortunately we failed to achieve N-Boc, N-Bz or N-Piv, N-Bn groups. We observed most of the transformations starting materials intact in the reactions and few transformations observed unknown products formation. We are extremely sorry for this failure.

10) Is there anything known in the literature, regarding excitation of RSO_2I and RSO_2SeR under visible light irradiation? If so, authors should cite all relevant articles in the references.

Answer: We are thankful for your suggestion. As for reviewer suggestions, we cross checked the availability of the known literature, regarding excitation of RSO_2I but there is no literature support. However, we find in the case of RSO_2SeR in the presence of photocatalyst but in the absence of photocatalyst there is no relevant articles in the literature.

Ref: for RSO_2SeR excitation under metal (J. Org. Chem., **84**, 12324-12333 (2019)). Without any metal or photocatalyst there is no literature support for excitation. Thus we did not cite any related references to this query.

11). The authors underline multiple times that their reaction is environmentally benign, although the reported methods mostly use DCM as solvent in quite dilute conditions.

Answer: We are sorry for this mistake. We agree with the reviewers concern because DCM is not a green solvent. We revised “environmentally benign” to “mild reaction conditions” and highlighted with yellow color

in the revised manuscript.

12). Rephrase ‘gram-scale reactions’ to ‘reactions on larger scale’, as only one is really gram scale.

Answer: We are sorry for this mistake. We corrected in the revised manuscript and highlighted with yellow color.

13) Reference 39 is not present in the manuscript. Reference 40 contains the citation referred to in the manuscript as reference 39? This should be checked for all references.

Answer: We are sorry for the inconvenience caused. We corrected in the revised manuscript. Please see in the revised manuscript.

14). Scheme 1, D has been copied wrongly from the initial source (<https://www.nature.com/articles/s41467-020-19467-5.pdf>).

Answer: We are sorry for this mistake. We corrected in the revised manuscript. Please see in the revised manuscript Fig. 1.

15). In all schemes: make sure that PG and EWG are displayed as such, and not as GP or GWE. Examples: Scheme 1,a (EWG); Scheme 1,f (PG).

Answer: We are sorry for this mistake. We corrected in the revised manuscript. Please see in the revised manuscript.

16). The proposed name of 1 does not convey the position of the phenylethynyl group on the N-phenyl moiety? Add the position indicator ‘2’.

Answer: We are sorry for this mistake. We corrected in the revised manuscript. Please see in the revised manuscript.

17). In the manuscript body (page 4, column 1) the text structure has been mixed up. ‘To our surprise, the reaction with the naphthyl functionality product produced 25 at and the scope of radical precursor reagents sulfonyl halogens 2 were tested with [...]’

Answer: We are sorry for the inconvenience caused. We corrected and include the missed text in the revised manuscript and highlighted with yellow color

18) Appraisals of reaction yields were given inconsistently. E.g. “Most importantly, the heterocyclic moiety (41) smoothly produced the desired product at a moderate yield of 63%.” and “The active C-I in the synthesized products was further transformed into the respective derivatives, such as boronic acid (Suzuki reaction), to produce the phenyl ring at a good yield of 61% (77).”

Answer: We are sorry for the typo mistakes. We corrected in the revised manuscript.

19) The optimization Table is found in the paper itself (Table 1), reference to the SI should be deleted. “In continuation of this derivatization, we extended the scope of the reaction to consider a higher equivalence of the sulfonyl hydrazide and oxidant and a prolonged reaction time under visible-light irradiation (see the complete

optimization in the supporting information).”

Answer: We are thankful for your careful observation. We have deleted the “supporting information” in the Table 1 text.

20) The complete compound numbering system needs to be checked and corrected in text, schemes and supporting information. • “Next, we applied various selenosulfate radical precursors bearing electron-donating groups/withdrawing groups, –Ph (61), p-OMe-Ph (62), p-Cl-Ph (63), and p-F-Ph (64), in the sulfone moiety and –Ph (65) and p-Me-Ph (66) in the selenium moiety to produce the desired product in moderate yields of 39%-69%.”

• Where is compound number 43?

Answer: We are sorry for the mistake. We completely revised the compounds numbers and highlighted with yellow color in the revised manuscript and also in supporting information.

• The deiodination product is number 79 in the scheme, not 78 is indicated in the text.

Answer: We are thankful for your careful observation. We corrected in the revised manuscript.

• “Next, a deiodination product (78) was obtained through palladium catalysis at a moderate yield (54%). Most importantly, under basic reaction conditions, the elimination of sulfone and iodine produced indole 3-substituted alkynes (78) at a 58% yield.”

• Place number 78 underneath the expected product in Scheme 4B eq. 5 for clarity.

Answer: We are thankful for your careful observation. As for your suggestion, we included the revised compound number.

21) In Scheme 4A, 4-methylbenzene boronic acid was used, while coupled product bears a phenyl group. Correct to the right structure in correspondence with the SI.

Answer: We are thankful for your careful observation. We corrected in the revised manuscript.

22) Attention should be paid to the spelling of toluene. (Scheme 4A)

Answer: We are sorry for the inconvenience caused. We carefully checked the spelling of the toluene in the manuscript as well as SI.

23) Please write compound numbers consistently in bold.

Answer: We are thankful for your careful observation. Revised all compound numbers in bold and highlighted with yellow color in the revised manuscript.

Supporting information:

General comments:

A. The SI should be carefully checked for English grammar and spelling. A consistent use of tenses must be followed, and typing mistakes corrected. Underneath some examples.

o Mass spectra and high-resolution mass spectra (HRMS) were measured using the LTQ Orbitrap XL (Thermo Fischer Scientific) Liquid chromatography–mass spectrometry at National Sun Yat-sen University. (S3).

Answer: We are thankful for your English corrections. We revised the English grammar and spellings in the supporting information.

o. Reaction mixture before light irradiation (S4)

Answer: We are thankful for your English correction. We revised the English grammar and spellings in the supporting information.

o. Pictures represent the reactions setup under light-irradiation of blue LED source by using compounds 1 and 2 and the progress of the reaction monitored by thin layer chromatography (S4) o “slowly reaction mixture warming to room temperature.” (S6)

Answer: We are thankful for your English correction. We revised the English grammar and spellings in the supporting information.

o Then NBS (1.1 equiv) was added in portions. (S5)

Answer: We are thankful for your English correction. We revised the English grammar and spellings in the supporting information.

o To a stirred solution of sodium sulfite (2 mmol), sodium bicarbonate (2 mmol) and the corresponding sulfonyl chloride (1 mmol) were dissolved in H₂O (S8) → Sulfonyl chloride was added to a stirred solution of sodium sulfite and sodium bicarbonate.

Answer: We are thankful for your English correction. We revised the English grammar and spellings in the supporting information.

o To a dried flask was added charged with S7 (1.0 equiv), CuSO₄·5H₂O (0.1 equiv), 1,10-phenanthroline (0.2 equiv) and K₃PO₄ (2.2 equiv) in dry toluene was added bromoalkyne (note: need to prepare freshly before performing reaction) was and the mixture was stirred at 80°C 2-12h.

Answer: We are thankful for your English correction. We revised the English grammar and spellings in the supporting information.

o In an oven-dried screw-capped 5 mL vial equipped with a magnetic stir bar was charged with 2-alkynyl-ynamides (1.0 equiv), TsI (1.1 equiv), and DCM (0.05 M) solvent was added. The resulting solution was stirred upto starting material completion at room temperature under blue LED light. After that, the crude reaction mixture was diluted water and extracted with DCM (some times without extraction directly purified with silica). The organic layer was dried over Na₂SO₄, filtered, and concentrated. The residue was purified by flash chromatography.

Answer: We are thankful for your English correction. We revised the English grammar and spellings in the

supporting information.

o We commenced (past tense) our radical strategy by using thiol as a radical precursor with 4-methyl-N-(phenylethynyl)-N-(phenylethynyl)phenyl)benzenesulfonamide (1) in an DCM solvent under 34 W blue light-emitting diode (LED) light irradiation. But, to our disappointment we get (present tense) only trace amount of product 67 (Table S1, entry 1). An extensive solvent screening process led (past tense) to these optimized reaction conditions (Table S1, entries 2-6). Then, the equivalence of compound 2 was altered but failed to improve the yield (Table 1, entry, 8). We carried out the reaction in the absence of a light source (at room temperature) but this reaction did not improve the reaction (Table 1, entry 9). The same reaction continued for 24 h, and compounds 3 and 4 were observed in 61/<15 yields (Table 1, entry 13). Thus, we choose chose the reaction condition of entry 6 for our derivatization.

Answer: We are thankful for your English correction. We revised the English grammar and spellings in the supporting information.

Preparation of Starting Materials (section 3)

B. There are major inconsistencies between the schemes and the text of section 3, 3.1, 3.3, 3.4, 3.5. These sections should be carefully revised. Problems are not limited to:

a. Amounts of reagents different in scheme vs text

- Pd(PPh₃)₄, CuI : experimental of compounds S2 and S6

Answer: We are thankful for your careful observation. We corrected and revised in the supporting information.

- K₂CO₃ : experimental of compound S3, 13C-1'

Answer: We are thankful for your careful observation. We corrected and revised in the supporting information.

- TsCl : experimental of compound S7

Answer: We are thankful for your careful observation. We corrected and revised in the supporting information.

- K₃PO₄ : experimental of compound S8

Answer: We are thankful for your careful observation. We corrected and revised in the supporting information.

- Se : experimental of compound 4 (ditolyl diselenide)

Answer: We are thankful for your careful observation. We corrected and revised in the supporting information.

- PIDA : experimental of compound 5 (selenosulfonate)

Answer: We are thankful for your careful observation. We corrected and revised in the supporting information.

b. Wrong reagents in the scheme or the text (pyridine or triethylamine for the synthesis of S7?)

Answer: We are thankful for your careful observation. We corrected and revised in the supporting information.

c. Solvent not mentioned in the text (THF). (Compound S2)

Answer: We are thankful for your careful observation. We corrected and revised in the supporting information.

d. Amounts not mentioned in the text, only in the scheme (Et₃N) (compound S2)

Answer: We are thankful for your careful observation. We corrected and revised in the supporting information.

e. Typing mistakes: Benzaldehyde (0.1 equiv.) should be 1 equiv. (compound S5, alternative route)

Answer: We are thankful for your careful observation. We corrected and revised in the supporting information.

f. Place either all reaction times in the scheme, or none.

Answer: We are thankful for your careful observation. We corrected and revised in the supporting information.

g. Please add exact purification conditions (heptane/EtOAc ratio) when reporting column chromatography

Answer: We are thankful for your careful observation. We corrected and revised in the supporting information.

h. Please always specify inert gas used to evacuate and refill flasks.

Answer: We are thankful for your careful observation. We corrected and revised in the supporting information.

C. Attention to correct chemical formulation should be paid:

- Subscripts of $\text{CuSO}_4 \cdot 5\text{H}_2\text{O}$ in scheme of section 3.1

Answer: We are thankful for your careful observation. We corrected and revised in the supporting information.

- $\text{CuSO}_4 \cdot 5\text{H}_2\text{O}$ instead of $\text{CuSO}_4 \cdot 5\text{H}_2\text{O}$ in experimental for S8 derivatives

Answer: We are thankful for your careful observation. We corrected and revised in the supporting information.

- MgSO_4 instead of MgSO_4 in experimental of S5

Answer: We are thankful for your careful observation. We corrected and revised in the supporting information.

- Toluene instead of Toluene (many times, among which scheme section 3.5)

Answer: We are thankful for your careful observation. We corrected and revised in the supporting information.

- 90 C \rightarrow 90 °C (3.3 step 1)

Answer: We are thankful for your careful observation. We corrected and revised in the supporting information.

- The unit is 'mol%' not 'mole%'.

Answer: We are thankful for your careful observation. We corrected and revised in the supporting information.

D. In the Scheme on page S9, there is overlapping text. Also: the second route is also indicated as route-A, and there is no experimental text accompanying this route?

Answer: We are thankful for your careful observation. We corrected and included experimental text for that route in the supporting information.

Optimization table (section 4)

E. Table S1 has been placed in section 3.5 (III) by mistake.

Answer: We are thankful for your careful observation. We corrected in the supporting information.

Experimental procedures (section 5)

F. Add the gradient used for flash chromatography.

Answer: We are thankful for your suggestion. We add the gradient used for flash chromatography in the revised supporting information.

G. Attention to correct chemical formulation should be paid: I2 instead of I₂

Answer: We are thankful for your careful observation. We corrected and revised in the supporting information.

H. Numbering of the compounds is not corresponding with the manuscript.

o General procedure (B) for the synthesis of (E)-3-(1-iodo-2-phenyl-2-tosylvinyl)-2-phenyl-1-tosyl-1H-indole derivatives (3) → 3, 5-27. What about the bromo-substituted derivatives (3a and 3b)? Include them in this general procedure ('sulfonyl halides') or make a separate general procedure. Make a separate general procedure for all reactions using sulfonyl hydrazide (28 – 41?)

Answer: We are sorry for the inconvenience caused. We revised the compounds numbers carefully and include the general procedure for corresponding compounds. Please see in the revised supporting information.

o General procedure (C) for synthesis of (E)-2-phenyl-3-(2-phenyl-1,2-ditosylvinyl)-1-tosyl-1H-indole derivatives (5, 42 and 44) → 5 is not a 1,2-ditosyl compound → 4, 42, 44. Also: take care to account for the oxygen atom issue!

Answer: We are sorry for the inconvenience caused. We revised general experimental procedures to corresponding product. To check the oxygen atom issue we have carried out several experiments in the question 5.

o General procedure (D) for synthesis of (E)-2-phenyl-3-(2-phenyl-1-(phenylselanyl)-2-tosylvinyl)-1-tosyl-1H-indole derivatives (45) → 45-66

Answer: We are sorry for the inconvenience caused. We carefully adopted each general procedure for respective series of the compounds.

o General procedure (E) for synthesis of (E/Z)-2-phenyl-3-(2-phenyl-2-(phenylthio)vinyl)-1-tosyl-1H-indole derivatives (5) → compound 5 is not a phenylthio derivative!; 67-76

Answer: We are sorry for the inconvenience caused. We carefully adopted each general procedure for respective series of the compounds.

I. The gram scale syntheses are unique procedures, not general procedures. Please add masses and moles of all reactants and reagents, as well as purification details. Use the correct specific name of the molecules used instead of a general group name, e.g. 'selenosulfonates'. Add reaction times to the experimental procedures and make them correspond to the scheme. Idem for 5.7. Control studies (section 6).

Answer: We are sorry for the inconvenience caused. We revised the compounds numbers carefully and include the general procedure for the larger scale synthesis. Please see in the revised supporting information.

J. 6.1: typing mistake on word 'observed' in scheme.

Answer: We are sorry thankful for your careful observation. Corrected in the revised supporting information.

K. Text underneath scheme 6.2 is copied from 6.1 and wrong for reaction with thiols.

Answer: We are sorry for the inconvenience caused. Corrected in the revised supporting information.

Characterization data:

The reported data should be double checked with the raw data, and corrected in both SI and manuscript if necessary. Data is lacking.

Answer: We are extremely sorry for the inconvenience caused. We have double checked the data and provided in the revised manuscript as well as supporting information.

- The title (IUPAC name) of the compounds is not always corresponding to the chemical structures displayed.

Answer: We are sorry extremely for the mistake. We carefully revised the IUPAC name of the compounds numbers

- Connect the right compounds to the right general procedure. For all compounds after compound 7, general procedure A is mentioned, which is clearly wrong.

Answer: We are sorry for the mistake. We connected the each general procedure for respective compounds.

- Compounds 3a and 3b were not synthesized using sulfonyl iodide. Adapt the general procedure.

Answer: We are sorry for the mistake. We adapted the general procedure for the respective compounds.

- HRMS missing for compounds: 3, 26, 28, 42, 44, 51, 61.

Answer: We are sorry for the inconvenience caused. We add the HRMS data for the compounds 3, 26, 42, 44, 51, 61.

- Compound 43 is never mentioned.

Answer: We are thankful for your careful observation. We carefully revised and updated compounds numbers in the manuscript as well as supporting information.

- Compounds missing: 73, 63, 62, 59, 58, 53, 50, 24, 4

Answer: We are sorry for the inconvenience. We add the missing compounds in the supporting information.

- The mass of pure products 77, 78, 79 is missing. Their yield is different in the manuscript (Scheme 4A).

Ans: We are sorry for the inconvenience. We add the mass of the pure products of the revised compound numbers **86, 87, 88**.

- The reported yield of 3b is different in the manuscript (Scheme 2).

Answer: We are sorry for the inconvenience. Corrected the mistake.

NMR Spectra:

The presence of all spectra should be checked again, doubles should be deleted. Furthermore, re-purification of impure isolated products could be considered.

- Some spectra still contain impurities.

o E.g. page S66, 68, aromatic impurities in both ^1H as ^{13}C NMR

Answer: We are sorry for the inconvenience. We purified the compounds.

o E.g. page S75, aromatic impurities in ^1H NMR

Answer: We are sorry for the inconvenience. We purified the compounds.

o E.g. page S83, 84, aromatic impurities in both ^1H as ^{13}C NMR

Answer: We are sorry for the inconvenience. We purified the compounds.

o E.g. page S58, 59, aromatic impurities in both ^1H as ^{13}C NMR

Answer: We are sorry for the inconvenience. We purified the compounds.

- Some spectra are added in duplo, for other compounds one of the spectra is missing.

Answer: We are sorry for the inconvenience. We carefully cross checked spectra and the duplicated was deleted.

o E.g. page S69 and S70 contain the same spectrum.

Answer: We are sorry for the inconvenience. We carefully cross checked spectra and the duplo was deleted.

o E.g. page S81 and 82 contain the same spectrum.

Answer: We are sorry for the inconvenience. We carefully cross checked spectra and the duplo was deleted.

o E.g. page S71 and S72 contain the same spectrum, a ^1H spectrum of that compound is missing.

Answer: We are sorry for the inconvenience. We carefully cross checked spectra and the duplo was deleted.

o E.g. page S85 and 86 contain the same spectrum, a ^1H spectrum of that compound is missing.

Answer: We are sorry for the inconvenience. We carefully cross checked spectra and the duplo was deleted.

- Please peak pick the solvent signals, especially when used to calibrate the ^{13}C NMR axis.

Answer: We are thankful for your valuable suggestion. Add the solvent signals in the ^{13}C NMR

- Some spectra feature very broad signals. Indicate as 'broad singlet' (brs). E.g. page S89, 91, 226,

Answer: We are thankful for your valuable suggestion. We add the broad singlets where it requires.

- In general, peaks are not so 'sharp' and signals are often distorted. Is there a shimming problem with the used NMR?

Answer: We are thankful for your valuable query. We asked NMR technician at Kaohsiung Medical University (NMR instrumentation center) but she say there is no issue in the shimming of NMR machine. In this case, to know the exact reason, first we checked compound **3** in CDCl_3 by using 600 MHz (proton and carbon). In addition, we also used various solvents and mixture of solvents to check outcome of the spectra but there is no major change in the shape of the peaks and the spectra were provided below and also included in the supporting information.

Solvent CDCl_3
Spectrometer Frequency 597.24

Solvent CDCl_3
Spectrometer Frequency 150.19

Based on these results the peaks shape is not influenced by solvent or shimming in the NMR instrument. Next, we tried to find the possible reason for the shape of the peak and the signal broadening in the ^1H NMR (we observed broad signals in the (*E*)-3-(1-iodo-2-phenyl-2-tosylvinyl)-2-phenyl-1-tosylindole derivatives. In this case, we checked the distance and bond angles of crystal compound of 3 as shown below. The bond distance between the I-H17: 4.146 Å, I-H21: 4.491 Å and bond angle C22-I-H17: 39.48°, C22-I-H21: 77.73°. Based on this outcome, we hypothesized the bulky iodine may influence the neighboring aromatic protons to make the ortho protons broadened in R¹ when aromatic substitution in the iodine products (compound 3 derivatives) and also it may also influence peak shapes by nuclear quadrupole moments of iodine. The related references were shown below and also provided google links for your kind reference

Supplementary Figure 5: Iodine and phenyl group bond distance and bond angles in the crystal structure of (*E*)-3-(1-iodo-2-phenyl-2-tosylvinyl)-2-phenyl-1-tosylindole

In addition, an indirect confirmation of the broad peaks in the products can confirm by proton NMR of the R¹=aliphatic the peak shape good in the (*E*)-3-(1-iodo-2-phenyl-2-tosylvinyl)-2-phenyl-1-tosylindole derivatives as

shown below.

References and useful links

<https://organicchemistrydata.org/hansreich/resources/nmr/?page=07-multi-04-quadrupolar/>

<https://organicchemistrydata.org/hansreich/resources/nmr/?page=08-tech-01-relax/#08-tech-01-relax-qr>

http://nmrwiki.org/wiki/index.php?title=Quadrupolar_coupling

Reviewer: 2

Wang and Mutra has presented a useful photoinduced radical-triggered transformations of alkyne tethered ynamides to access synthetically important indole derivatives. The unprecedented regio- and chemoselective ynamide C-N bond fission triggered by photoinduced radical chalcogens, and reshuffling followed by alkyne functionalization provides synthetically challenging indole scaffolds. The direct synthetic method for the preparation of 3-olefin bearing fully functionalized indoles is noteworthy. The reaction was highly efficient, occurs under mild conditions within short reaction time. The transformation makes many new bonds, such as: N-C, C-C, C-SO₂R, C-SR, C-I and C-Se. Important to note that, this reaction does not require expensive metals, photocatalyst, oxidants and additives; only under blue LED and solvent (DCM/CH₃CN), new bonds have been constructed efficiently within a few minutes. Scope of the reaction is very high as various functional group substituted alkyne-ynamides and chalcogens have sustained under the reaction condition. To validate the insights of reaction mechanisms various control experiments have been performed. The site-selective ynamide C-N bond cleavage and in situ sulfone reshuffling on α -carbon of ynamides has been proved with the synthesis of ¹³C-labeled indole product.

Overall, the work shown in this manuscript is expedient and interesting. I, therefore, support this manuscript for publication in Nature Communications after addressing the following queries.

Answer: Thank you very much to the reviewer for taking the time and effort to review our manuscript and gave positive comments on our work. In our reaction we used 40 W Kessil A160WE Controllable LED Aquarium Light (in the previous version of the manuscript we given 34 W, We are extremely sorry for tthis mistake).

1. To understand the participation of Ts-substituted alkyne (89) by inter or intramolecular fashion, the crossover experiment needs to be conducted with differently substituted alkyne tethered ynamides.

Answer: We are thankful for your valid suggestion. According to your suggestion, the alkyne migration step was investigated via a crossover experiment with substituted yndiamides **1k** and **1n**. No crossover products were observed under standard condition, suggesting that the alkyne migration is an intramolecular process,

which is consistent with previously reported functional groups on 2-alkynylanilines.

2. Reference numbers 24 and 36 are same.

Answer: We are sorry for the inconvenience caused. The reference section completely modified and updated in the revised manuscript.

3. Introduction part; 8th line, “catalyzed1” change to “catalyzed”

Answer: We are thankful for careful observation. Corrected in the revised manuscript.

4. Show the reaction outcome when alkyl substituted sulfonyl halogens employed in this reaction.

Answer: We are thankful for your query. According to reviewer query, we treated various alkyl substituted sulfonyl halogens in our standard reaction conditions. Most of the sulfonyl chlorides/bromides failed to produce desired product as shown in the following reaction except ethanesulfonyl iodide. The probable reason could be the low reactivity compared sulfonyl iodides. The related references were cited in the revised manuscript.

5. Is this reaction appropriate for homopropargyl tethered ynamides with no aryl ring between alkyne and ynamide?

Answer: We are thankful for your query. As for your suggestion, we synthesized the 4-methyl-*N*-(phenylethynyl)-*N*-(3-phenylprop-2-yn-1-yl)benzenesulfonamide and treated with 4-methylbenzenesulfonyl

iodide under standard reaction condition. But, we did not observed the N-C(SP) bond cleavage in 4-methyl-*N*-(phenylethynyl)-*N*-(3-phenylprop-2-yn-1-yl)benzenesulfonamide, due to the reversal reactivity of the propargyl amine compared to alkyne ynamide based on profs: . This type of ynamides well studied by Profs: Gandon and Sahoo with radical precursors and the related references were also cited in the manuscript.

6. Except compound 3b, all the reactions conducted with Ts protected ynamides. Comment about other electron withdrawing groups.

Answer: We are thankful for your query. According to your query, we tried to synthesize other protecting groups in 2-alkynyl-ynamides, such as N-Boc, N-Bz, N-Bn or N-Piv as shown below. But unfortunately we failed to achieve N-Boc, N-Bz or N-Piv, N-Bn groups. We observed most of the transformations starting materials were intact in the reactions and few transformations observed unknown products.

7. Cite the recent work on keteniminium driven functionalization of ynamides (Angew. Chem. Int. Ed. 2020, 59,

10785).

Answer: We are thankful for your suggestion. We have add the suggested article in the revised along with another article refs: 14 and 15

8. Correct spelling of Sahoo instead Shaoo.

Answer: Thank you very much for careful observation. We corrected in the revised manuscript.

9. Chem. Asian J. 14, 2019, 4282 should be added in ref. 11.

Answer: We are thankful for your suggestion. Added in the ref:34

10. Scheme 1a: the keteniminium species should contain both H and M. Modify the Scheme by not only highlighting cycloisomerization but also difunctionalization.

Answer: We are thankful for your suggestion. According to your suggestions, we have revised in the current version of the manuscript.

11. GWE should be corrected to EWG in scheme 1a.

Answer: We are thankful for your careful observation. We corrected it

12. R is missing in the product of scheme 1c right side.

Answer: We are thankful for your careful observation. We corrected it in the revised manuscript.

13. mole% should be corrected to mol% in scheme 1c.

Answer: We are thankful for your careful observation. We corrected it in the revised manuscript.

14. Scheme 1a, [M] = Metal. or H⁺ should be removed.

Answer: We are thankful for your careful observation. We corrected it in the revised manuscript.

15. Scheme 1c, space is missing in yne-tethered ynamide.

Answer: We are thankful for your careful observation. We corrected it in the revised manuscript

16. Scheme 1e, m of mechanism is merging with the line.

Answer: We are thankful for your careful observation. We corrected it in the revised manuscript

17. Scheme 1e, Ts and Au is merging.

Answer: We are thankful for your careful observation. We corrected it in the revised manuscript

18. Scheme 1f, GP should be changed to PG.

Answer: We are thankful for your careful observation. We corrected it in the revised manuscript

19. Scheme 1f & g, region should be replaced with regio.

Answer: We are thankful for your careful observation. We corrected it in the revised manuscript

20. Sulfonyl bromides and chlorides gave less yields. Any possible explanation?

Answer: We are thankful for your query. Sulphonyl iodide is more reactive than the corresponding bromide and the bromides are more reactive as the corresponding chloride. This is the result of the relative strength of sulphur-halogen bonds. The related references were cited in the revised manuscript. Please see the revised

21. Few examples of various sulfonyl groups of ynamides including alkyl ones should be tried.

Answer: We are thankful for your suggestion. According to your suggestion, we prepared respective sulfonyl groups of ynamides including alkyl one, tried in our standard reaction condition and all the reaction smoothly affords the desired products with good yields as shown below. The synthesized new derivatives added in the revised manuscript. Please see in the revised manuscript. Please see in the revised manuscript.

22. The reactivity of halogen substituted selenyl compounds along with aliphatic ones should be tried.

Answer: We are thankful for your query. According to your query, we prepared respective halogen substituted selenyl compounds along with aliphatic one tried in our standard reaction condition and all the reaction afford except aliphatic we observed low yields due less reactivity compared to aromatic selenyl compounds. The new synthesized derivatives were added in the revised manuscript. Please see in the revised manuscript.

23. Suggest the plausible mechanism when thiols are used. Did the authors observe thiol dimerization?

Answer: We are thankful for your query. Based on the control experiments and previous reports we have proposed possible mechanism for thiol radical cascade cyclization and migration. In this case, we also observed minor unknown structure but not dimerization, at this stage we don't know the exact structure (previously, we have mentioned this information in the supporting information optimization table foot note). We tried to prepare crystal but unfortunately, we failed to get crystal. For your kind reference, we have given ^1H NMR and ^{13}C NMR spectra of the unknown product. Please find the below attached spectra.

24. Change the deiodination product (79) instead 78.

Answer: We are thankful for your careful observation. We corrected in the revised manuscript

25. Compound number 80-92 should be bold.

Answer: We are thankful for your careful observation. We corrected it in the revised manuscript

26. The feasibility of Ts-C(SP) bond homolytic cleavage for the conversion of D to E should be supported with few control experiments.

Answer: We are thankful for your suggestion. According to your suggestion, first we tried the reactions with 2-phenyl-1-tosylindole with 1-methyl-4-((phenylethynyl)sulfonyl)benzene but failed to get the expected product due to the absence of radical on third position of indole in first reaction. Next, to prove the feasibility of Ts-C(SP) bond homolytic cleavage we selected compound 1-methyl-4-((phenylethynyl)sulfonyl)benzene as radical acceptors and tetrahydrofuran source in this visible light induced reaction, in this reaction we observed expected product.

27. Suggest the possible reasons in the attack of Ts radical to β -carbon. As attack at α -carbon could provide stable benzylic radical.

Answer: We are thankful for your query. We agree with your concept in general, the benzylic radical is more stable because of more resonance structures. But in our reaction mechanism, we hypothesized, the sulfone radical addition at β -carbon to provide more stable radical in the intermediate-E in the reaction. The formed radical have opportunity get more resonance structures in the indole compared to benzylic position in case of radical addition on α -carbon

Reviewer: 3

This manuscript describes a synthetically useful and mechanistically interesting transformation from ynamide precursors under photochemical conditions with cleavage of the C-N bond and recombination to yield a range of indole derivatives. In many cases the addition of the sulfonyl iodide proceeds to form preferentially the E isomer although when thiol is used as the radical source mixture of E and Z isomers are formed. The scope of the transformations has been examined in detail. Following a series of experiments to determine the reaction pathway a reaction mechanism has been proposed.

Answer: Thank you very much to the reviewer for taking the time and effort to review our manuscript and gave positive comments on our work. In our reaction we used 40 W Kessil A160WE Controllable LED Aquarium Light (in the previous version of the manuscript we given 34 W, We are extremely sorry for this mistake).

The authors might clarify why they describe this as ‘sulfone reshuffling’ – it seems that the sulfonyl iodide adds across an alkyne following a rearrangement to form the indole skeleton.

Answer: We are thankful for your valuable query. We used the term “sulfone reshuffling” because the initial generated radical addition on α -carbon of the intermediate A in the mechanism. Then, simultaneous radical cased via liberation of the sulfone radical (please see in the reaction mechanism intermediate C to D), followed by radical addition on β –carbon of the intermediate D (in this entire process the sulfone is moving from α -carbon to β –carbon of the ynamide alkyne. Thus, we believe the word “sulfone reshuffling” may fit for this process).

Overall this work is likely to be of interest to synthetic and medicinal chemists as a route to indole derivatives.

Some detailed comments:

Answer: We greatly appreciate your highly positive comments.

Page 4 Column 1 line 2 2-Phenyl should be 2-phenyl

Answer: We are thankful for your careful observation. We corrected it in the revised manuscript

Page 4 Column 1 line 18 – it is not correct to say ‘different alkyl chain groups such as n-butyl groups ‘as there is only one example. ‘a different alkyl chain group which is a n-butyl group‘ The cyclopropyl substituted derivate is discussed separately in the next sentence.

Answer: We are thankful for your careful suggestion. We corrected in the revised manuscript and highlighted with yellow color.

Page 4 Column 1 line 8 from end – there is some text missing here – the sentence does not make any sense

Answer: We are thankful for your careful observation. Added the missing text in the revised manuscript and highlighted with yellow color.

Page 4 Column 1 line 2 from end – not clear which compound is intended here – in Scheme 3 88% yield is given for 27. In text 82% but not sure if it is the same compound.

Ans: We are thankful for your careful observation. We corrected in the revised manuscript and highlighted with yellow color

Page 4 Column 2 first paragraph – text is inconsistent – is there an effect or not?

Answer: We are sorry for the inconvenience. Corrected in the revised manuscript and highlighted with yellow color.

Page 4 Column 2 2nd paragraph. The reference to use of sulfonyl hydrazides in place of sulfonyl iodides is not clear – other than the footnote b in Scheme 2 there is no detail on this. In the SI for most of the indoles the experimental says use of general procedure A - should this be B or C depending on whether the sulfonyl iodide or sulfonyl hydrazide and I₂ are used? Suggest this is clarified. Should the text read sulfonyl hydrazide and iodine?

Answer: We are sorry for the inconvenience. We completely revised the supporting information with general procedure for corresponding products. Please see in the revised version of the supporting information.

Scheme 3 – the structure of the product should contain R1 and R2 in place of Ph

Answer: We are sorry for the inconvenience. Corrected in the revised manuscript.

Page 7 first paragraph – not at all clear what this sentence means ‘However....stage’

Answer: We are sorry for the inconvenience. We corrected and revised the manuscript.

Page 7 first paragraph – n-butyl-Ph – not at all clear to me what this means – is the butyl group on the alkyne, not on a phenyl?

Answer: We are extremely sorry for the inconvenience. Corrected in the revised manuscript.

Page 7 for the thiol reactions E and Z mixtures are formed. There is no discussion of the alteration in stereochemical outcome. Also in the SI I could find no information on the E:Z ratio – important to provide this information in the spectral assignment. Does ratio alter with substituents?

Answer: We are thankful for valuable suggestion. In the revised manuscript, we have provided the ration of the E:Z in the manuscript based on the proton NMR by using alkene proton. Yes, the ratio of E and Z mixture in the products, alter with substituents. In case of the compound **1** (R = Cl or Br) we observed exclusive E product (**77**) or major isomer E (**78**). We believe the, the formed radical intermediate (E) in the mechanism influence the single product formation. But remaining, all the cases we observed mixture of E:Z isomers. The following reference was added in the revised manuscript. Cited article in ref: 64

Page 7 column 2 rephrase ‘See for spectra’ to ‘For spectra see’

Answer: We are sorry for the inconvenience. According to your suggestion we have revised the sentence in the revised manuscript.

Page 9 the sentence ‘This synthesis concludedcarbons’ must be rephrased as it is not clear. I am not clear on what is meant by sulfone reshuffling – it seems that the sulfonyl moiety is simply adding across an alkyne after

the migration

Answer: We are thankful for your valuable query. We used the term “sulfone reshuffling” because the initial generated radical addition on α -carbon of the intermediate A in the mechanism. Then, simultaneous radical cased via liberation of the sulfone radical (please see in the reaction mechanism intermediate C to D), followed by radical addition on β -carbon of the intermediate D (in this entire process the sulfone is moving from α -carbon to β -carbon of the ynamide alkyne. Thus, we believe the word “sulfone reshuffling” may fit for this process).

In the revised manuscript, we completely revised in this section. Please see the revised manuscript.

Page 9 column 1 3 lines from end ‘radical beard indole’ – what is meant by ‘beard’

Answer: We are sorry for the inconvenience. We have revised the sentence in the manuscript.

Conclusions ‘exclusively E isomer’ – this is not the case with thiol trapping experiments.

Answer: We are sorry for the inconvenience. We removed the “E isomer” to “major single isomer”.

I have marked these and some further points on the attached PDF.

Answer: We are thankful for your support. We corrected and revised the manuscript

In the SI in many of the indole products there is a broad signal in the H NMR spectra at ~ 7.8 ppm – it would be helpful if the authors might clarify which protons account for this signal and why this is broadened.

Answer: We are thankful for valuable query. According to your suggestion, we tried to find the reason for the signal broadening in the ^1H NMR (we observed broad signals in the (*E*)-3-(1-iodo-2-phenyl-2-tosylvinyl)-2-phenyl-1-tosylindole derivatives. In this case, we checked the distance and bond angles of crystal compound of 3 as shown below. The bond distance between the I-H17: 4.146 Å, I-H21: 4.491 Å and bond angle C22-I-H17: 39.48° , C22-I-H21: 77.73° . Based on this outcome, we hypothesized the bulky iodine may influence the neighboring aromatic protons to make the ortho protons broadened.

Supplementary Figure 5: Iodine and phenyl group bond distance and bond angles in the crystal structure (*E*)-3-(1-iodo-2-phenyl-2-tosylvinyl)-2-phenyl-1-tosylindole

In addition, an indirect confirmation of the broad peaks in the products can confirm by proton NMR of the R¹= aliphatic in the (*E*)-3-(1-iodo-2-phenyl-2-tosylvinyl)-2-phenyl-1-tosylindole derivatives as shown below.

Solvent $CDCl_3$
Spectrometer Frequency 100.69

Solvent $CDCl_3$
Spectrometer Frequency 400.40

Solvent $CDCl_3$
Spectrometer Frequency 100.69

Solvent $CDCl_3$
Spectrometer Frequency 400.40

Solvent CDCl_3
Spectrometer Frequency 100.69

Some other changes were also made and marked in YELLOW color. Finally, we would like to show our great respects to all the reviewers. Your efforts have improved the quality of this manuscript. We hope that the revised manuscript will reach the level for publication in Nature Communications.

Thank you once again. We are looking forward to hearing from you.

Sincerely,

Prof. Jeh-Jeng Wang.

REVIEWER COMMENTS

Reviewer #1 (Remarks to the Author):

The authors have clearly done a major effort in revising their original manuscript, including the mechanistic investigation. This, together with a more logical and consistent build-up of the supporting information has added significantly to the scientific merit of the manuscript. Unfortunately, some problems remained or have appeared in the added text. Still, some numbering issues persist, as well as inconsistent yields for some products. Additionally, mixed sentences make the explanation of the added mechanistic study slightly dubious.

Some detailed comments:

- Please correct the following textual and content problems:

Next, we performed the reaction under N₂ atmosphere ~~under~~ (Supplementary Fig. 2-3), ~~and which afforded the~~ expected desired product ~~afford~~ in 81% yield (Table 1, entry 15).

From these findings, we believe there is no oxygen role in the reaction, ~~and the~~ blue LED light source is important for radical generation from sulfonyl iodide.

As depicted in Fig. 3, a broad range of substituted 2-alkynyl-ynamides (1) was compatible with this transformation to produce the corresponding (E)-3-(1-iodo-2-phenyl-2-tosylvinyl)-2-phenyl-1-tosylindole 3-26 with yields ranging from ~~35%~~ 29% to 85%.

This ~~contradicts supports~~ the importance of the mild reaction of our divergent radical strategy on the 2-alkynyl-ynamides because cyclopropane is very delicate in the radical homolytic bond fission process.

n-propanol → *n*-propanol, or better propan-1-ol

Various N-protecting groups (N-SO₂Ph (1w), 4-chloro-N-(phenylethynyl)-N-(2-(phenylethynyl)phenyl) benzenesulfonamide (1x), NSO₂Et (1y)) with 4-methylbenzenesulfonyl iodide (2a) smoothly delivered desired products 43-45 in ~~72-73-~~77% yields.

Next, we ~~altere~~d the reaction time from minutes to hours under optimized reaction conditions (a and b) in Fig. 4eb.

The carbon-linker length effect was These results indicate the possibility [...]. ~~Please correct the incomplete sentence.~~

Finally, a key reaction intermediate, 2-phenyl-3-(phenylethynyl)-1-tosylindole (87), was synthesized and treated with 2a under standard reaction conditions (Fig. 7, eq ~~12~~ 15).

~~Please revise the text on page 10, especially in the second column, the sentence structure has been mixed up. An example:~~

In addition to this we also conducted our blue LED emission and absorption of spectra of various radical precursors (Supplementary Fig. 6-12) tested by using compound S22 and 2a. To our disappointment, the expected product (92) was not observed, and a selective 1,2-addition on propargylamine was observed to produce product 97a at a 58% yield. ~~The spectra were indeed found in the SI, and the other text seems to belong to Fig 7 eq. 14, although product numbering (92 instead of 97 in the Fig.) and yield (58% instead of 55% in the Fig.) are reported. Please compare with the crude data and correct.~~

~~In the explanation of the mechanism on the same page and the first sentence of page 12 is a comparable mixing of sentences! Please revise this part of the text, explaining each step of the mechanism in the right order.~~

Conclusion: Please revise the text on English grammar **carefully**, especially the conjugation of verbs and the consistent use of tenses, and especially in the newly written yellow text. What has been mentioned above is no full list but just some examples explaining the need to revise carefully.

- In the mechanism of Figure 7, please draw the expulsion product and Ts-C(SP) bond fission and attachment to the indole radical for clarity.
- The authors clarified the structure of side-product **4**, generated at longer reaction times. However, please comment on the origin of the water, necessary for the *in-situ* TsOH formation from TsI. The SI mentions some trace water impurity due to the synthesis of TsI. Is the side product also observed in the reactions with the other sulfonyl-generating reactants,

such as sulfonyl bromides? Please discuss this explicitly, e.g. in the supporting information as a separate section.

- Still some numbering is not in bold in the text (e.g. p 5 second column; 2-alkynyl-ynamides **1**). Additionally, add numbers to the general schemes to the starting materials which have been allocated one (such as 2a and 1).
- Some instances of 'gram scale reactions' is still present. Please adjust to 'gram scale reaction' or 'larger scale reactions'.
- The Ms-group is also electron withdrawing. Please adjust the text on page 10 accordingly. The exact reason why RSO₂ as a protecting group is key to the success remains elusive. Please try to rationalize for the reader.
- In the caption of Figure 3, there is no explanation for superscript 'a', while product **30** has an 'a' in superscript next to its yield.

SUPPORTING INFORMATION

- Please add the result of the reaction executed in 2.2 to Supplementary Figure 3. In the rebuttal, two yields (46% and 10 %) were mentioned?
- 'To know the exact reasons for signal very broad and peaks not very sharp (¹H NMR) in the (E)-3-(1-iodo-2-phenyl-2-tosylvinyl)-2-phenyl-1-tosylindole derivatives.' Please correct this sentence for English grammar.
- Please add reaction times to the schemes in the SI. This has only been done to some of them at this point (following the reviewer remark).
- Why are there dotted structures in SI section 4.2? Is it possible to add the yields of these starting material syntheses to the scheme?
- In SI 4.3, the number of **1aa** is inconsistent with Figure 4c of the main text, where it is labelled **1ab**. Correct where appropriate. **1ab** is later referred to as the ¹³C containing starting material.
- In the title of 4.3, the final product is referred to as 'B'. Please exchange by either **1aa** or **1ab** depending on the outcome of the previous point.
- In 4.3, the spelling of 'toluene' should still be corrected once.
- 4.4 is no 'general' procedure, as the text is specific for sodium 4-methylbenzenesulfinate. Please exchange by a 'general' procedure, or add the note that other sulfinates were obtained in a similar fashion, or that it is a 'representative example'. Please check this for all general procedures.
- In 4.5, the compound name which was added in (200 mg, 1.12 mmol, 1.0 equivl) is missing. Also: equiv. instead of equivl !
- Add 30°C to the first step in the scheme of 4.7.
- Add nitrogen atmosphere in the scheme of 4.8. Also add the KOH to the scheme!
- Add the equivalence of sulfinate in the scheme of 4.8 in both pathways. Please also check this for other schemes.
- In 4.9, correct the name of 4-methyl-N-(2-(phenylethynyl)phenyl)benzenesulfonamide (**S7**) and check its numbering. Isn't this **S21**? **S7** is 2-(phenylethynyl)aniline?
- Add the gas with which the flask was evacuated and filled in 4.10 in the text.

- In 4.10: the pentahydrate indication is also subscript. Correct this. Also: 'toluene' spelling here.
- In 4.10 iii), the starting material is not *N*,4-dimethylbenzenesulfonamide, as indicated in the text.
- In 6.1, in the second scheme (after 'Or'), again a sulfonyl iodide is shown, while sulfonyl hydrazides have been used in combination with I₂ and TBHP.
- 6.9.4: why the dotted lines? Also: the product name and yield in the SI text are not consistent with respectively the product in the scheme and the yield in the main text. Probably by mistake copied from 6.9.3?
- Compound **3** is shown five (!) times in the characterization data. Please remove 4 instances and keep one with HRMS data!
- Compound 3-(2-(2,4-dichlorophenyl)-1-iodo-2-tosylvinyl)-2-phenyl-1-tosylindole (**24**) has been isolated as a mixture of E/Z. Please show this in Figure 3 in the main text by a wiggly bond. Do this consistently both in the main text as well as in the SI.
- Compound **30** has been isolated in 71% via procedure D according to the characterization section, while in the main text, 71% belongs to procedure E using the sulfonohydrazide? Please correct this if necessary.
- Compound **70** has a yield of 20% in the main text Figure 5, while in the characterization data, yield is 25%. Kindly compare with raw data and correct. Compound **64** also has this problem: 25% in main text and 53% in characterization data. Compound **56** has this as well. Please re-check all yields in the main text to the supporting information and compare them with the raw data. Other compounds were not checked as such. Possibly, these errors were generated by repurification?
- Compound **63** only has a yield and no isolated mass in the characterization data.
- Compound **70**: structure is wrong!
- Please use a wiggly bond consistently in order to show unseparable E/Z mixtures. Remove the wiggly bond for compound **77** in Figure 5, as this was isolated as 100% Z.
- Compound **91** in the characterization data is not compound **91** in Fig. 7a.
- The characterization data of compound **94** shows two different yields.

The manuscript has been revised thoroughly. However, some concerns need to be addressed before publication.

Comments:

1. The performed cross-over experiments does not really prove that migration is intramolecular. It should have been done using the following yne-ynamides, so that the cross-over products can be characterized.

2. It was suggested to use homo-propargyl tethered ynamide with no aryl group in between alkyne and ynamide. The author misunderstood it completely and performed the reaction with propargyl tethered ynamide. Needs to be done again.

3. Is it possible to propose some structure for the available NMR data?
4. In 'abstract' and throughout the manuscript, the hybridizations are written in unscientific ways, like '(N-C(SP²), C(SP²)...etc)'. Hybridization occurs through the combination of 's', 'p' etc. orbitals not 'S', 'P' (represent atomic/molecular term-symbol). Those should be written in small letters.
5. In 'Introduction', instead of "Professors Gandon and Sahoo et al.", it should be written as "Professors Gandon, Sahoo and co-workers."
6. Reference '33' is nowhere mentioned in the manuscript.
7. Fig. 1, Scheme a) in 'Difunctionalization' product, length of R⁴ and R⁵ are not same. In 'Metal Carbene' [M] should be in center of bond. In Scheme c), C-Se bond lengths are not same. Scheme d), [Au] should be in the center of the bond.
8. In "Table 1. Screening of the reaction conditions^a", author mentioned "b" as isolated yield, but in table nowhere it mentioned.
9. In 'Discussion' section;
 - A) In 'Figure 3', the general reaction equation lacks 'R' in the benzenoid ring.
 - B) Below 'Figure 3', author mentioned "c Isolated yields. d A major E isomer was formed". However, this is not mentioned in the table.
 - C) Product '24' contains '1,3-di-Cl-Ph' moiety, not '1,2-di-Cl-Ph (24)'. The yield is mentioned as 43% in the Table, while it has been shown as "...the desired product at a good yield (52%)".
 - D) Yield '25' is mentioned as 55% in the Table. See the text "...the naphthyl functionality product produced 25 at an excellent yield (82%)."
 - E) Isolated yield of '37' is 40% in the Table. But it is mentioned as 65%.
 - F) In 'Figure 5', particular reaction conditions are not mentioned for specific substrate.

Reviewer #3 (Remarks to the Author):

The authors have made efforts to address the issues raised by the referees in the revised manuscript and corrected many of the minor issues which have been specifically identified in the reviews.

A POINT-BY-POINT RESPONSE TO REVIEWER COMMENTS

Manuscript ID: NCOMMS-21-22456A

Title: Photoinduced Radical-Triggered Selective Ynamide Bond-Fission Structural Reshuffling and Functionalization

Author (s): Mohana Reddy Mutra, Jeh-Jeng Wang

Dear Reviewers,

Thank you very much for your time and efforts to review our manuscript. We have revised the manuscript according to your comments and suggestions. The corrections in detail were given in the revised manuscript and were highlighted in yellow color. The detailed answers were given for revision was listed as follows:

REVIEWER COMMENTS

Reviewer: 1

The authors have clearly done a major effort in revising their original manuscript, including the mechanistic investigation. This, together with a more logical and consistent build-up of the supporting information has added significantly to the scientific merit of the manuscript. Unfortunately, some problems remained or have appeared in the added text. Still, some numbering issues persist, as well as inconsistent yields for some products. Additionally, mixed sentences make the explanation of the added mechanistic study slightly dubious.

Answer: We thank again this reviewer for taking the time and efforts to review our manuscript and giving positive comments and valuable suggestions on our work to improve the quality and quantity of the work before publication. According to this reviewer's suggestion, we have completely revised our manuscript and supporting information and highlighted it with yellow color in the revised manuscript.

Some detailed comments:

- Please correct the following textual and content problems:

Next, we performed the reaction under N₂ atmosphere ~~under~~ (Supplementary Fig. 2-3), ~~and which afforded the~~ expected desired product ~~afford~~ in 81% yield (Table 1, entry 15).

Answer: We are thankful for your English correction. We follow your suggestions, corrected and highlighted it with yellow color in the revised manuscript.

From these findings, we believe there is no oxygen role in the reaction, **and the** blue LED light source is important for radical generation from sulfonyl iodide.

Answer: We are thankful for your English correction. We corrected it according to your suggestions and highlighted it with yellow color in the revised the manuscript.

As depicted in Fig. 3, a broad range of substituted 2-alkynyl-ynamides (1) was compatible with this transformation to produce the corresponding (E)-3-(1-iodo-2-phenyl-2-tosylvinyl)-2-phenyl-1-tosylindole 3-26 with yields ranging from ~~35%~~ 29% to 85%.

Answer: We are thankful for your careful observation. We corrected in the revised manuscript.

This ~~contradicts-supports~~ the importance of the mild reaction of our divergent radical strategy on the 2-alkynyl-ynamides because cyclopropane is very delicate in the radical homolytic bond fission process.

Answer: We are thankful for your English correction. We corrected it according to your suggestions and highlighted it with yellow color in the revised the manuscript.

n-propanol → **n**-propanol, or better propan-1-ol

Answer: We are thankful for your careful observation. We corrected in the revised manuscript.

Various N-protecting groups (N-SO₂Ph (**1w**), 4-chloro-N-(phenylethynyl)-N-(2-(phenylethynyl)phenyl) benzenesulfonamide (**1x**), NSO₂Et (**1y**)) with 4-methylbenzenesulfonyl iodide (**2a**) smoothly delivered desired products **43-45** in ~~72-73~~-77% yields.

Answer: We are thankful for your careful observation. We corrected in the revised manuscript.

Next, we alter**ed** the reaction time from minutes to hours under optimized reaction conditions (a and b) in Fig. 4**eb**.

Answer: We are thankful for your English correction. We corrected it according to your suggestions and highlighted it with yellow color in the revised the manuscript.

The carbon-linker length effect was These results indicate the possibility [...]. **Please correct the incomplete sentence.**

Answer: We are thankful for your careful observation. We corrected the incomplete sentence in the revised manuscript. Please see in the revised manuscript.

Finally, a key reaction intermediate, 2-phenyl-3-(phenylethynyl)-1-tosylindole (**87**), was synthesized and treated with **2a** under standard reaction conditions (Fig. 7, eq ~~12~~ 15).

Answer: We are thankful for your careful observation. We corrected in the revised manuscript.

Please revise the text on page 10, especially in the second column, the sentence structure has been mixed up. An

example: In addition to this we also conducted our blue LED emission and absorption of spectra of various radical precursors (Supplementary Fig. 6-12) tested by using compound **S22** and **2a**. To our disappointment, the expected product (**92**) was not observed, and a selective 1,2-addition on propargylamine was observed to produce product **97a** at a 58% yield. The spectra were indeed found in the SI, and the other text seems to belong to Fig 7 eq. 14, although product numbering (92 instead of 97 in the Fig.) and yield (58% instead of 55% in the Fig.) are reported. Please compare with the crude data and correct.

Answer: We are sorry for the inconvenience caused in case of inconsistent compounds numbering and few yields mismatching from manuscript to supporting information. We thoroughly checked, corrected and revised in the manuscript. Please check in the revised manuscript.

In the explanation of the mechanism on the same page and the first sentence of page 12 is a comparable mixing of sentences! Please revise this part of the text, explaining each step of the mechanism in the right order.

Conclusion: Please revise the text on English grammar carefully, especially the conjugation of verbs and the consistent use of tenses, and especially in the newly written yellow text. What has been mentioned above is no full list but just some examples explaining the need to revise carefully.

Answer: We are thankful for your valid suggestions. We have thoroughly checked the typo and grammatical errors in the manuscript and revised according to your suggestions. Please see the revised manuscript highlighted it with yellow color.

- In the mechanism of Figure 7, please draw the expulsion product and Ts-C(SP) bond fission and attachment to the indole radical for clarity.

Answer: We are thankful for your suggestion. We revised the reaction mechanism as per your suggestion. Please see Fig. 8 in the revised manuscript.

- The authors clarified the structure of side-product 4, generated at longer reaction times. However, please comment on the origin of the water, necessary for the in-situ TsOH formation from TsI. The SI mentions some trace water impurity due to the synthesis of TsI. Is the side product also observed in the reactions with the other sulfonyl-generating reactants, such as sulfonyl bromides? Please discuss this explicitly, e.g. in the supporting information as a separate section.

Supplementary Figure 6: Investigation of side product (*E*)-2-phenyl-1-(2-phenyl-1-tosylindol-3-yl)-2-tosylvinyl 4-methylbenzenesulfonate (**4**) with various radical precursors under longer reaction time.

Answer: Generation of the side product (*E*)-2-phenyl-1-(2-phenyl-1-tosylindol-3-yl)-2-tosylvinyl 4-methylbenzenesulfonate (**4**) was investigated with various radical precursors as shown above Supplementary Figure 6. First, we choose 4-methyl-*N*-(phenylethynyl)-*N*-(2-(phenylethynyl)phenyl)benzenesulfonamide (**1a**) with 4-methylbenzenesulfonyl iodide (**2a**) as radical precursor under standard reaction conditions for 24 h (Supplementary Figure 6. Eq. 1). At short reaction time (2-5 minutes), we did not observe the formation of the compound **4** but under longer reaction time, observed formation of the side product (**4**) with 11% yield. We hypothesized that the probable reason for the side product formation could be the in situ generation of TsOH from **2a** with a trace amount of water (The compound **2a** was synthesized from sodium *p*-toluenesulfinate and I₂ in H₂O as a solvent. After formation of yellow solid in the reaction, filtered by using Büchner funnel. The yellow solid was washed with water (2-3 times) to remove unreacted starting material (sodium *p*-toluenesulfinate or I₂). The yellow solid was dried 5-10 minutes (longer time, the compound will decompose) under high vacuum pump and carried to the next step without further workup or purification. In this entire process, we believe there is a trace amount of water in the compound **2a**).

In case of 4-methylbenzenesulfonohydrazide (**2ab**) as a radical precursor under standard conditions for 48 h (Supplementary Figure 6. Eq. 2), observed side product **4** formation in 8% yield (herein, the water source is aq. TBHP).

Next, we choose compound **3** as a starting material with commercially available TsOH.H₂O under standard reaction conditions for 24 h (Supplementary Figure 6. Eq. 3) and the reaction afford the side product **4** formation in 44% yield. This reaction suggests that the in situ generated TsOH can act as a nucleophile with weak C-I bond in the product **3**.

In addition to the above radical precursors, we choose the 4-methylbenzenesulfonyl bromide (**2ca**) as radical precursors to check the formation of side products in our standard reaction condition (24-48 h) as shown above (Supplementary Figure 6. Eq. 4) but we did not observe any side products (**4**). Next, we choose compound **3** as a starting material with commercially available TsOH.H₂O under standard reaction condition for 24 h (Supplementary Figure 6. Eq. 5) and the reaction did not afford the side product **4** except a trace amount on TLC. The probable reason could be stronger C_{sp}²-Br bond compared to C_{sp}²-I bond in compound **46** for the in situ generated TsOH.

This Supplementary Figure 6 an text included in the Supporting information.

- Still some numbering is not in bold in the text (e.g. p 5 second column; 2-alkynyl-ynamides **1**). Additionally, add numbers to the general schemes to the starting materials which have been allocated one (such as **2a** and **1**).

Answer: We are sorry for the inconvenience caused. We gave compounds numbers in bold and added numbers to the general schemes to the starting materials.

- Some instances of ‘gram scale reactions’ is still present. Please adjust to ‘gram scale reaction’ or ‘larger scale reactions’.

Answer: We are sorry for the inconvenience. We corrected in the revised manuscript.

- The Ms-group is also electron withdrawing. Please adjust the text on page 10 accordingly. The exact reason why RSO₂ as a protecting group is key to the success remains elusive. Please try to rationalize for the reader.

Answer: We are thankful for your valid suggestions. We revised and provided the probable reason as follow:

To determine the importance of the electron-withdrawing group in the 2-alkynyl-ynamides (**1**), we replaced –Ts (aromatic) with the –Ms (aliphatic) group and treated it under standard reaction conditions to produce the desired product (**98**) at an excellent yield (80%) (Fig. 7a, eq 9). The molecular structure of product **98** was unambiguously confirmed by X-ray crystallography.⁶² These results suggest that sulfone derivatives (aliphatic or aromatic) may not be involved in alkyne migration process, but they will exert a strong influence on ynamide stabilization and isomerization¹⁻⁸ (Despite our attempts, we were not able to synthesize other protection groups in compound **1**).

Please follow the refs: 1-8 and *J. Am. Chem. Soc.* 138, 13135–13138 (**2016**) for the importance of the protecting group in the ynamide (Note: JACS article was not cited in the reference section due to the limited reference number).

- In the caption of Figure 3, there is no explanation for superscript ‘a’, while product **30** has an ‘a’ in superscript next to its yield.

Answer: We are sorry for the inconvenience caused. We corrected in the revised manuscript.

SUPPORTING INFORMATION

- Please add the result of the reaction executed in 2.2 to Supplementary Figure 3. In the rebuttal, two yields (46% and 10 %) were mentioned?

Answer: We are thankful for your careful observation. We corrected in the revised manuscript.

‘To know the exact reasons for signal very broad and peaks not very sharp (¹H NMR) in the (*E*)-3-(1-iodo-2-phenyl-2-tosylvinyl)-2-phenyl-1-tosylindole derivatives.’ Please correct this sentence for English grammar.

Answer: We are thankful for your suggestion. We revised in the Supporting information.

- Please add reaction times to the schemes in the SI. This has only been done to some of them at this point (following the reviewer remark).

Answer: We are thankful for your suggestion. We added the reaction times to the schemes in the revised Supporting information.

- Why are there dotted structures in SI section 4.2? Is it possible to add the yields of these starting material syntheses to the scheme? In SI 4.3, the number of **1aa** is inconsistent with Figure 4c of the main text, where it is labelled **1ab**. Correct where appropriate. **1ab** is later referred to as the ¹³C containing starting material.

Answer: We are sorry for the inconvenience caused. We corrected in the revised Supporting information.

- In the title of 4.3, the final product is referred to as ‘B’. Please exchange by either **1aa** or **1ab** depending on the outcome of the previous point.

Answer: We are sorry for the inconvenience caused. We corrected in the revised Supporting information.

- In 4.3, the spelling of ‘toluene’ should still be corrected once.

Answer: We are sorry for the repeated mistake in the spelling of toluene. We corrected in the revised Supporting information.

- 4.4 is no ‘general’ procedure, as the text is specific for sodium 4-methylbenzenesulfinate. Please exchange by a ‘general’ procedure, or add the note that other sulfonates were obtained in a similar fashion, or that it is a ‘representative example’. Please check this for all general procedures.

Answer: We are thankful for your careful observation. We corrected in the revised Supporting information.

- In 4.5, the compound name which was added in (200 mg, 1.12 mmol, 1.0 equivl) is missing. Also: equiv. instead of equivl !

Answer: We are thankful for your careful observation. We corrected in the revised Supporting information.

- Add 30°C to the first step in the scheme of 4.7.

Answer: We are thankful for your careful observation. We corrected in the revised Supporting information.

- Add nitrogen atmosphere in the scheme of 4.8. Also add the KOH to the scheme!

Answer: We are thankful for your careful observation. We corrected in the revised Supporting information.

- Add the equivalence of sulfinate in the scheme of 4.8 in both pathways. Please also check this for other schemes.

Answer: We are thankful for your careful observation. We corrected in the revised Supporting information.

- In 4.9, correct the name of 4-methyl-*N*-(2-(phenylethynyl)phenyl)benzenesulfonamide (**S7**) and check its numbering. Isn't this **S21**? **S7** is 2-(phenylethynyl)aniline?

Answer: We are thankful for your careful observation. We revised and given the corrected compound number.

- Add the gas with which the flask was evacuated and filled in 4.10 in the text.

Answer: We are thankful for your suggestion. We added N₂ gas for this respective text.

In 4.10: the pentahydrate indication is also subscript. Correct this. Also: 'toluene' spelling here.

Answer: We are sorry for our repeated mistake in the spelling of toluene. Corrected in the revised Supporting information.

- In 4.10 iii), the starting material is not *N*,4-dimethylbenzenesulfonamide, as indicated in the text.

Answer: We are sorry for the inconvenience caused. We corrected in the revised Supporting information.

- In 6.1, in the second scheme (after ‘Or’), again a sulfonyl iodide is shown, while sulfonyl hydrazides have been used in combination with I₂ and TBHP.

Answer: We are thankful for your careful observation. We revised the text with sulfonyl hydrazides in combination with I₂ and TBHP.

6.9.4: why the dotted lines? Also: the product name and yield in the SI text are not consistent with respectively the product in the scheme and the yield in the main text. Probably by mistake copied from 6.9.3?

Answer: We are sorry for the inconvenience caused. While converting word to pdf file the dotted line appeared. In the revised Supporting information, we corrected yield of the product with main text manuscript.

Compound **3** is shown five (!) times in the characterization data. Please remove 4 instances and keep one with HRMS data!

Answer: We are thankful for your suggestion. We deleted 4 instances and kept one with HRMS data.

Compound 3-(2-(2,4-dichlorophenyl)-1-iodo-2-tosylvinyl)-2-phenyl-1-tosylindole (**24**) has been isolated as a mixture of E/Z. Please show this in Figure 3 in the main text by a wobble bond. Do this consistently both in the main text as well as in the SI.

Answer: We are thankful for your careful observation. We give wobble bond for isolated mixture of E/Z isomers.

- Compound **30** has been isolated in 71% via procedure D according to the characterization section, while in the main text, 71% belongs to procedure E using the sulfonohydrazide? Please correct this if necessary.

Answer: We are thankful for your careful observation. We revised the procedure from D to E.

- Compound **70** has a yield of 20% in the main text Figure 5, while in the characterization data, yield is 25%. Kindly compare with raw data and correct. Compound **64** also has this problem: 25% in main text and 53% in characterization data. Compound **56** has this as well. Please recheck all yields in the main text to the supporting information and compare them with the raw data. Other compounds were not checked as such. Possibly, these errors were generated by repurification?

Answer: We are sorry for the inconvenience caused. We checked thoroughly and corrected the

yields of the products from main text to Supporting information.

Compound **63** only has a yield and no isolated mass in the characterization data.

Answer: We are thankful for your careful observation. Provided the isolated mass in the characterization data of revised Supporting information.

- Compound **70**: structure is wrong!

Answer: We are thankful for your careful observation. Corrected the structure of the compound in the revised Supporting information.

Please use a wiggly bond consistently in order to show unseparable E/Z mixtures. Remove the wiggly bond for compound **77** in Figure 5, as this was isolated as 100% Z.

Answer: We are thankful for your careful observation. Removed wiggly bond in compound **77**.

Compound **91** in the characterization data is not compound **91** in Fig. 7a.

Answer: We are thankful for your careful observation. We give exact compound characterization data in the revised Supporting information.

- The characterization data of compound **94** shows two different yields.

Answer: We are thankful for your careful observation. Correct in the Supporting information.

Reviewer: 2

The manuscript has been revised thoroughly. However, some concerns need to be addressed before publication.

Answer: We thank this reviewer for supporting our work in the prestigious journal “*Nature communications*”. We also thank this reviewer for providing valuable suggestions in our work to improve quality and quantity of the manuscript.

Comments:

1. The performed cross-over experiments does not really prove that migration is intramolecular. It should have been done using the following yne-ynamides, so that the cross-over products can be characterized.

Answer: We are thankful for your valid suggestion. According to your suggestion, we synthesized the above yne-ynamides (**1k** and **1y'**) and performed the reaction under standard reaction conditions (see Fig. 7, eq 12 in the manuscript). But no crossover products were formed (**16a** and **100a**) and selectively intramolecular alkyne migration products were obtained **16** and **100** (69 and 58%) in yields. This crossover experiments suggest that the alkyne migration step is intramolecular.

- It was suggested to use homo-propargyl tethered ynamide with no aryl group in between alkyne and ynamide. The author misunderstood it completely and performed the reaction with propargyl tethered ynamide. Needs to be done again.

Answer: We are extremely sorry for our misunderstanding in the previous revision. According to your suggestion, we synthesized the compound **1ab** without aryl group in between alkyne and ynamide as shown below. We also provided the ^1H NMR and ^{13}C NMR spectra for your kind reference.

¹H NMR and ¹³C NMR of compound 1ab

After synthesizing compound **1ab**, we performed the reaction with radical precursor (**2a**) as shown below. But the reaction of homo-propargyl tethered ynamide (**1ab**), showed multiple spots on thin layer chromatography (TLC), probably due to the less stability of the starting material **1ab** (spontaneous decomposition in the presence of water (trace amount) within 2-3 days to get hydration product with ynamide alkyne) and also the formed product can easily isomerization into various products.⁶⁴ In accordance with these reactions, the aromatic ring between the alkyne and ynamide in compound **1** is necessary for stability via electron delocalization.

3. Is it possible to propose some structure for the available NMR data?

Answer: We are thankful for your query. But we are unable to provide structures with available NMR data, due to new series of compound derivatives obtained in our research.

4. In 'abstract' and throughout the manuscript, the hybridizations are written in unscientific ways, like '(N-

C(SP²), C(SP²)...etc)’. Hybridization occurs through the combination of ‘s’, ‘p’ etc. orbitals not ‘S’, ‘P’ (represent atomic/molecular term-symbol). Those should be written in small letters.

Answer: We are thankful for your careful observation. We corrected in the revised manuscript and highlighted it with yellow color.

5. In ‘Introduction’, instead of “Professors Gandon and Sahoo et al.”, it should be written as “Professors Gandon, Sahoo and co-workers.”

Answer: We are thankful for your suggestion. We added “co-workers” in the revised manuscript and highlighted it with yellow color.

6. Reference ‘33’ is nowhere mentioned in the manuscript.

Answer: We are thankful for careful observation. Reference 33 was included in the page 1, column 2 and highlighted with yellow color. Please check in the revised the manuscript.

7. Fig. 1, Scheme a) in ‘Difunctionalization’ product, length of **R4 and R5** are not same. In ‘Metal Carbene’ [M] should be in center of bond. In Scheme c), C-Se bond lengths are not same. Scheme d), [Au] should be in the center of the bond.

Answer: We are thankful for your careful observation. Corrected in the Fig. 1, Please check in the revised manuscript.

8. In “**Table 1. Screening of the reaction conditions**”, author mentioned “**b**” as isolated yield, but in table nowhere it mentioned.

Answer: We are thankful for your careful observation. Footnote “b” mentioned in Table 1 and highlighted it with yellow color. Please check in the revised manuscript.

9. In ‘**Discussion**’ section;

A) In ‘**Figure 3**’, the general reaction equation lacks ‘**R**’ in the benzenoid ring.

Answer: We are thankful for your careful observation. Included “R” in the benzenoid ring in Fig. 3. Please check in the revised manuscript.

B) Below ‘**Figure 3**’, author mentioned “**c Isolated yields. ^d A major E isomer was formed**”. However, this is not mentioned in the table.

Answer: We are thankful for your careful observation. We completely revised footnotes of the Fig. 3. Please check in the revised Fig. 3 footnote in the manuscript.

C) Product ‘**24**’ contains ‘**1,3-di-Cl-Ph**’ moiety, not ‘1,2-di-Cl-Ph (**24**)’. The yield is mentioned as **43%** in the Table, while it has been shown as “...the desired product at a good yield (52%)”.

Answer: We are thankful for your careful observation. We corrected from 1,2-di-Cl-Ph (**24**) to 2,4-di-Cl-Ph (**24**) and highlighted it with yellow color in the text. In addition, the product yield also corrected.

D) Yield ‘**25**’ is mentioned as **55%** in the Table. See the text “...the naphthyl functionality product produced **25** at an excellent yield (82%).”

Answer: We are thankful for your careful observation. We corrected the yield of the product in the revised manuscript. Please check in the revised the manuscript.

E) Isolated yield of ‘**37**’ is **40%** in the Table. But it is mentioned as 65%.

Answer: We are thankful for your careful observation. We corrected the yield of the product in the revised manuscript. Please check in the revised the manuscript.

F) In ‘**Figure 5**’, particular reaction conditions are not mentioned for **specific substrate**.

Answer: We are thankful for your suggestions. We completely revised the footnote of the Fig. 5, in the revised manuscript. Please check in the revised the manuscript.

Reviewer: 3

The authors have made efforts to address the issues raised by the referees in the revised manuscript and corrected many of the minor issues which have been specifically identified in the reviews.

Answer: **Answer:** We thank this reviewer for supporting our work in the prestigious journal “*Nature communications*” as an article.

In summary, we feel all the suggestions and concerns from the referees have been carefully addressed or explained with experimental support. Finally, thank you very much again for the time you spend reviewing our manuscript.

Sincerely,

Prof. Jeh-Jeng Wang.

REVIEWER COMMENTS

Reviewer #1 (Remarks to the Author):

The authors once again have revised their manuscript according to the comments of all reviewers. Most importantly, the inconsistency issues regarding the yields were carefully solved. Some things do require further careful attention before the manuscript is ready for publication.

MANUSCRIPT

- In the abstract, “gram-scale reactions” is still mentioned.
- The commonly used name is selenosulfonate, instead of selenosulfate. Please correct this where necessary.
- The authors write that “the blue LED light source is important for radical generation from sulfonyl iodide”. On the other hand, they later write “but, we did not observe specific wave length of blue LED light absorbance by compounds **1a**, **2a**, **2c**, **2da**, **2bd**, **2ea**. This suggests that blue LED light only induce radical generation”. This seems contradictory: if no light absorption occurs by the radical generating species, then why is the reaction so fast under light irradiation?
 - The authors should clarify this ‘enhancement of the homolytic bond fission without excited state’. Is there any literature support for such a phenomenon?
 - The authors should check the reactivity of their reaction with a Blue LED source which has no second emission peak at 382 nm in order to back the claim that the reaction is sped-up by ‘visible light irradiation’, as it is below 400 nm. (see quote below)
 - The authors mentioned in a previous rebuttal that the temperature was kept constant using a fan in an air-conditioned lab. However, the local temperature can rise quickly when using a strong 40 W lamp. What is the temperature of a reaction vessel with pure DCM after 5 minutes? Can a (slight) rise in temperature be responsible for the increased rate of homolytic cleaving (see also the reaction in absence of light after 24 h – Table 1, entry 14).
 - It is not clear for a reader who did not read the SI what is meant by compounds **2da**, **2bd** and **2ea**, as they are never mentioned or drawn in the manuscript.

*“The reaction speed and conversion is greatly enhanced by **irradiation with visible light** to initiate the radical reaction; an added advantage is that the reaction also gives a higher final conversion in short reaction time.”*

- The authors show a proposed mechanism for the reaction using thiols as radical source in the supporting information. A reaction with the corresponding disulfide did not generate the desired product, seemingly indicating that the authors rule out the intermediacy of a disulfide. However, the authors did not add a hydrogen atom source in this reaction, which might explain the lack of reactivity. A better way of proving the intermediacy of a disulfide would be a scrambling experiment by the addition of ditolyl disulfide (**S20**) to a reaction with benzenethiophenol (see reaction below). Incorporation of the tosyl group would then support a role for a disulfide species. Disulfides have been reported as blue light absorbers and are thought to be in equilibrium with their thiyl radicals under blue LED irradiation. This would solve the apparent absence of blue LED absorption by the thiol.

suggested control reaction:

- Unfortunately, the language of the manuscript is still not up to the level required in Nature Communications. At times, this complicates the interpretation of the message of the authors. Some examples are shown below, but this list is not exhaustive.

A) *The items in the following list do not all correspond to the verb 'require' i.e. "the strategies require a lack of atom economy".*

"Despite their advantages, these limited existing radical strategies require expensive metals/photocatalysts, oxidants, longer reaction times, harmful waste production and a lack of atom economy."

B) *The following sentence loses cohesion from 'no regio- and chemoselectivity [...]'. Verbs are missing. Again, not all items in the list correspond to the verb 'require'.*

"Nevertheless, despite the corresponding advances, the above process requires metal/Lewis acid, harsh reaction conditions, follows an intramolecular ionic path, and no regio- and chemoselectivity issues in the reaction and restricted to further functionalization of migrating group under mild reaction conditions."

Suggestion:

"Nevertheless, despite the corresponding advances, the above process requires metal/Lewis acid, harsh reaction conditions, follows an intramolecular ionic path, no regio- or chemoselectivity issues are solved, and the reactions are restricted in terms of further functionalization of the migrating group."

C) *The newly written parts suffer most from strange sentence structures, not limited to:*

"But, we did not observe specific wave length of blue LED light absorbance by compounds (1a, 2a, 2c, 2da, 2bd, 2ea). This suggests that blue LED light only induce radical generation."

Suggestion:

"However, the absorbance of specific wavelengths of the blue LED light source by reactants 1a, 2a, 2c, 2da, 2bd, 2ea was not observed, which suggests that blue LED light only induces radical generation." (This of course depending on the outcome of the analysis as outlined earlier.)

D) *In the following sentence, please use a consistent notation of sp^2 with the rest of the manuscript (as per the comment of another reviewer). Additionally, clarify that it is compound 3 that has the C-I bond, and 46 which has the C-Br bond.*

"We note that, when we used 4-toluenesulfonyl bromide as a radical source, the product 4 did not form in the reaction due to higher bond energy ($C_{sp^2}-Br$) than $C_{sp^2}-I$ bond in compound 3."

E) *Please use the past tense consistently in a paragraph.*

“To prove homolytic bond fission in Ts-C(sp) (**99**), we choose **96** and **99** as starting material and performed the reaction but failed to get expected product [...].

SUPPORTING INFORMATION

- The past tense of ‘choose’ is ‘chose’. Please adjust on page S10 and where it is furthermore needed. Also change SP² to sp² on page S10.
- There is an ‘S’ standing below the structure of iodobenzene in the figure of 5.3 on page S15.
- Compound **2ca** is called **2ac** in the title of 5.8 in the SI, yet again **2ca** in the scheme itself.
- Add ‘**S22**’ to the title of 5.10.
- The text of page S24 mentions compound **67** while it is in fact compound **76**.
- Has compound **41** been synthesized by using an isolated sulfonyl iodide or from the sulfonyl hydrazide? In the characterization, it mentions General Procedure E (sulfonyl hydrazide), but footnote b) is not placed at this compound in Figure 3 in the manuscript, and entry **41** is omitted in the title of section 7.2 in the SI.
- The yields of the both reactions in section 7.4 is not corresponding to those reported in Figure 4b in the main manucript! This for both reactions.
- In 10.1, the yield of the adduct **92** is not given in the supporting information. Additionally, a ‘-’ is given instead of 0 % for the reaction with this radical trap.
- In 10.2: typo in scheme: ‘observed’.
- In the characterization, the compounds **1a** and **1b** are indicated to be isolated ‘as a brown color’. Please exchange this by ‘as a brown colored solid’ or ‘as a brown solid’.

Reviewer #2 (Remarks to the Author):

The Authors have addressed the queries. The quality of the revised manuscript has been significantly improved. In my opinion, this fine piece of work is now suitable for publication in Nature Communications.

A POINT-BY-POINT RESPONSE TO REVIEWER COMMENTS

Manuscript ID: NCOMMS-21-22456B

Title: Photoinduced Radical-Triggered Selective Ynamide Bond-Fission Structural Reshuffling and Functionalization

Author (s): Mohana Reddy Mutra, Jeh-Jeng Wang

Dear Reviewers,

Once again thank you very much for your time and efforts to review our manuscript. We have revised the manuscript according to your comments and suggestions. The corrections in detail were given in the revised manuscript and highlighted it in yellow color. The detailed answers were given for revision was listed as follows:

REVIEWER COMMENTS

Reviewer: 1

The authors once again have revised their manuscript according to the comments of all reviewers. Most importantly, the inconsistency issues regarding the yields were carefully solved. Some things do require further careful attention before the manuscript is ready for publication.

Answer: We thank again this reviewer for taking the time and efforts to review our manuscript and accepting by providing valuable suggestions and comments. We believe all the suggestions and concerns from this reviewer have been carefully addressed or explained with experimental support. We have revised Fig.1h graphical abstract because we feel the side product formation makes confusion while reading.

MANUSCRIPT:

- In the abstract, “gram-scale reactions” is still mentioned:

Answer: We are sorry for our repeated mistake. We revised according to your suggestion and highlighted it with yellow color in the revised manuscript.

- The commonly used name is selenosulfonate, instead of selenosulfate. Please correct this where necessary.

Answer: We are thankful for your careful observation. Revised and highlighted it with yellow color in the

- The authors write that “the blue LED light source is important for radical generation from sulfonyl iodide”. On the other hand, they later write “but, we did not observe specific wave length of blue LED light absorbance by compounds **1a**, **2a**, **2c**, **2da**, **2bd**, **2ea**. This suggests that blue LED light only induce radical generation”. This seems contradictory: if no light absorption occurs by the radical generating species, then why is the reaction so fast under light irradiation?

Answer: We are thankful for your query. Sulfonyl iodides have the ability to spontaneous homolytic bond fission (or) as decomposition (Please see Table 1, entry 14). We believe blue LED light source helps shorten the reaction time and also enhancement of the product yields by providing photo-energy in the cleavage of weak RSO₂-I bond in compound **2a** via homolytic bond fission”. We have provided following references for your kind references.

- 1) Da Silva Correa, C. M. M. & Waters, W. A. Reactions of the free p-toluene sulfonyl radical. I. Diagnostic reactions of free radicals. *J. Chem. Soc., C*, 1874-1879 (1968).
- 2) Liu, L. K., Chi, Y. & Jen, K.-Y. Copper-catalyzed additions of sulfonyl iodides to simple and cyclic alkenes. *J. Org. Chem.* 45, 406-410 (1980).
- 3) Kang, S.-K., Ha, Y.-H., Kim, D.-H., Lim, Y. & Jung, J. Toluene-p-sulfonyl-mediated radical cyclization of bis(allenes) utilizing p-TsBr and p-TsSePh. *Chem. Commun.* 1306-1307 (2001).
- 4) Buathongjan, C., Beukeaw, D. & Yotphan, S. Iodine-Catalyzed Oxidative Amination of Sodium Sulfinates: A Convenient Approach to the Synthesis of Sulfonamides under Mild Conditions. *Eur. J. Org. Chem.* 1575-1582 (2015).
- 5) Hwang, S. J., Shyam, P. K. & Jang, H.-Y. Synthesis of Vinyl Sulfones via I₂-mediated Alkene Sulfonylations with Thiosulfonates. *Bull. Korean Chem. Soc.* **39**, 535-539 (2018).
- 6) Kostyukov, S. G., Masterova, Y. Y., Burtasov, A. A. & Romanova, E. V. Photochemical Selenosulfonation of Tricyclo[4.1.0.0^{2,7}]heptane Derivatives. *Russ. J. Org. Chem.* **56**, 1006-1013 (2020).
- 7) Pan, X., Gao, J., Liu, J., Lai, J., H. Jiang & Yuan, G. Synthesis of sulfonamides via I₂-mediated reaction of sodium sulfinates with amines in an aqueous medium at room temperature. *Green Chem.* **17**, 1400-1403 (2015).
- 8) Katrun, P. et al. Regioselective C2 Sulfonylation of Indoles Mediated by Molecular Iodine. *J. Org. Chem.* **79**, 1778-1785 (2014).

- The authors should clarify this ‘enhancement of the homolytic bond fission without excited state’. Is there any literature support for such a phenomenon?

Answer: We are thankful for your query. There are literature reported methods compounds **2a** and **2da** derivatives can easily bond cleavage occurs under light irradiation. We have provided the all the related references for your kind reference.

- 1) Alabugin, I. V., Timokhin, V. I., Abrams, J. N., Manoharan, M., Abrams, R. & Ghiviriga, I. In Search of Efficient 5-Endo-dig Cyclization of a Carbon-Centered Radical: 40 Years from a Prediction to Another Success for the Baldwin Rules. *J. Am. Chem. Soc.* **130**, 10984-10995 (2008).
- 2) Zhang, Y. & Vessally, E. Direct halosulfonylation of alkynes: an overview. *RSC Adv.* **11**, 33447-33460 (2021).
- 3) Mei, H., Pajkert, R., Wang, L., Li, Z., Röschenthaler, G.-V. & Han, J. Chemistry of electrochemical oxidative reactions of sulfinate salts. *Green Chem.* **22**, 3028-3059 (2020).
- 4) Loccufier, J., Torfs, R. & Sauvageot, M. Solder mask inkjet inks for manufacturing printed circuit boards. WO2018087059A1 (2018).
- 5) Truce, W. E. & Wolf, G. C. Adducts of sulfonyl iodides with acetylenes. *J. Org. Chem.* **36**, 1727-1732 (1971).
- 6) Truce, W. E., Heuring, D. L. & Wolf, G. C. Addition of sulfonyl iodides to allenes. *J. Org. Chem.* **39**, 238-244 (1974).
- 7) Brumwell, J. E., Simpkins, N. S. & Terrett, N. K. Radical cyclization of dienes and enynes using phosphorus- and sulfur-centered radicals. *Tetrahedron* **50**, 13533-13552 (1994).
- 8) Kostryukov, S. G. & Masterova, Y. Y. On Radical Reactions of 1-Bromotricyclo[4.1.0.0^{2,7}]heptane with Phenylethynyl Sulfones. *Russ. J. Org. Chem.* **56**, 741-745 (2020).
- 9) Hiltunen, J. et al. Process for the preparation of arylsulfonylpropenenitriles by photocatalytic reactions. FI128424B (2020).
- 10) Aaltonen, T. et al. Process for the preparation of arylsulfonylpropenenitriles by photocatalytic reactions. WO2019043289A1 (2019).
- 11) Ho, H. T., Ito, O., Iino, M. & Matsuda, M. Studies of sulfonyl radicals. 4. Flash photolysis of aromatic sulfones. *J. Phys. Chem.* **82**, 314-319 (1978).
- 12) Serra, A. C., Correa, C. M. M. d. S., Vieira, M. A. M. S. A. & Gomes, M. A. Radical chain cyclization of allylic compounds. *Tetrahedron* **46**, 3061-3070 (1990).
- 13) Takahara, Y., Iino, M. & Matsuda, M. Studies of sulfonyl radicals. II. Relative reactivities of addition reactions of sulfonyl free radicals to vinyl monomers. *Bull. Chem. Soc. Jpn.* **49**, 2268-2271 (1976).

(Note: Due to the limited space in the reference section, selected references were cited in the revised manuscript and highlighted it yellow color).

- The authors should check the reactivity of their reaction with a Blue LED source which has no second emission peak at 382 nm in order to back the claim that the reaction is sped-up by ‘visible light irradiation’, as it is below 400 nm. (see quote below)

Answer: We are thankful for your query. As per your query, we purchased 40 W blue LED lamp (PR160L-456 nm) (which has no second emission peak at <400 nm) from Fiberoptics & DiCon Lighting, Kaohsiung, Taiwan. See the following pictures for more details.

Outside box appearance of Kessil 40 W PR160L-456 nm blue LED light

Kessil PR160L-456 nm blue LED light and supporting power cables inside the box

Kessil PR160L-456 nm blue LED light and supporting power cables outside the box

40 W blue LED light and adapter

40 W blue LED light

Kessil PR160L-456 nm

50% intensity of blue LED used for our reaction

Frontside view of Blue LED light

Technical Specifications

Specifications

Power Consumption	= 456 nm (max 40W)
Input Voltage	= 100-240 VAC
Operating Temperature	= 0 - 40 °C / 32 - 104 °F
Beam Angle	= 56°
Average Intensity of PR160 series	= 352mW/cm ² (measured from 1 cm distance)
Dimensions	= 4.49" x 2.48" / 11.4cm x 6.3cm (H x D)

For more details please find the following link

<https://kessil.com/science/PR160L.php>

Supplementary Figure 8: Pictures represent the 40 W blue LED lamp (PR160L-456 nm) and other technical

details.

Blue LED Emission Spectra

Emission spectra were measured using Ocean Optics USB 2000+ Spectrometer. Spectra were normalized to 1.0 at the emission maximum. This emission spectra was provided by Miss Angela Liou, Sales Specialist, DiCon Fiberoptics & DiCon Lighting, aliou@diconfiberoptics.com, Kaohsiung, Taiwan, +886 7 815-8055 Ext 485, DiCon Brands - Kessil | Fiilex | Cielux.

Supplementary Figure 12. Emission spectrum from a 40 W Kessil PR160L-456 nm Blue LED shown as blue color line with emission maximum at $\lambda_{\text{max}} = 456 \text{ nm}$

Next, we tried the reaction under PR160L-456 nm wavelength light by using compounds **1a** and **2a**. After 5 minutes, the progress of the reaction monitor by TLC, diluted with water and extracted with DCM and purified column purification. The reaction given 80% yield of compound **3**. Base on this reaction, blue LED light can induce the radical process in a fast and efficient manner.

Supplementary Figure 4. Pictures represent the reaction setup under light-irradiation of 40 W PR160L-456 nm source by using compounds **1a** and **2a** and the progress of the reaction monitored by thin-layer chromatography.

- The authors mentioned in a previous rebuttal that the temperature was kept constant using a fan in an air-conditioned lab. However, the local temperature can rise quickly when using a strong 40 W lamp. What is the temperature of a reaction vessel with pure DCM after 5 minutes? Can a (slight) rise in temperature be responsible for the increased rate of homolytic cleaving (see also the reaction in absence of light after 24 h – Table 1, entry 14).

Answer: We are thankful for your suggestion. We agree with your concern, there will be a slight rise in the temperature (The temperature can also fluctuate depends on the place and surrounding environment). According to your suggestion, we monitor the pure DCM temperature after 5 minutes under blue LED light irradiation by using thermometer. The pictures are shown below and revised in manuscript as well as in the Supporting information. We noticed, the temperature is at 28 °C. As discussed above there may be slight fluctuation in the temperature depends on place and surrounding environment.

**Kessil A160WE Tuna Blue LED
light source on**

Pure DCM under Blue LED

Turn off the light after 5 minutes

**Immediately checked the temperature
of pure DCM by using thermometer**

**Zoom appearance of the
thermometer temperature**

28 °C

**Checked oil bath temperature under blue
LED light**

**Zoom appearance of the
thermometer temperature**

28 °C

Supplementary Figure 5. Pictures represent the pure DCM solvent and oil bath temperature checking under light-irradiation (5 minutes) of 40 W Kessil A160WE Tuna Blue LED.

Next, we tried the reaction under similar reaction temperature in the absence of light source as shown below but the reaction failed afford desired product after 5 minutes. This reaction also suggests the light influence the reaction time and yields by cleavage of the weak $\text{RSO}_2\text{-I}$ bond.

Reaction under similar temperature under heating condition

Close appearance of the reaction

25% EA+Hexane
Reaction mixture
TLC after 5 minutes

Supplementary Figure 6. Pictures represent the similar temperature in absence of light source (40 W Kessil A160WE Tuna Blue LED).

- It is not clear for a reader who did not read the SI what is meant by compounds **2da**, **2bd** and **2ea**, as they are never mentioned or drawn in the manuscript.

Answer: We are thankful for your suggestion. We have inserted the structures of compounds **2da**, **2bd** and **2ea** in the reference 74. Please check in the revised manuscript.

Structures of compounds **2da**, **2bd** and **2ea** are as are given below.

“The reaction speed and conversion is greatly enhanced by irradiation with visible light to initiate the radical reaction; an added advantage is that the reaction also gives a higher final conversion in short reaction time.”

Answer: We are thankful for your suggestion. We followed your suggestion and revised in the manuscript.

- The authors show a proposed mechanism for the reaction using thiols as radical source in the supporting information. A reaction with the corresponding disulfide did not generate the desired product, seemingly

indicating that the authors rule out the intermediacy of a disulfide. However, the authors did not add a hydrogen atom source in this reaction, which might explain the lack of reactivity. A better way of proving the intermediacy of a disulfide would be a scrambling experiment by the addition of ditolyl disulfide (S20) to a reaction with benzenethiophenol (see reaction below). Incorporation of the tosyl group would then support a role for a disulfide species. Disulfides have been reported as blue light absorbers and are thought to be in equilibrium with their thiyl radicals under blue LED irradiation. This would solve the apparent absence of blue LED absorption by the thiol.

Answer: We are thankful for your valuable suggestions and details explanation of the possible reaction mechanism for the formation of compound **76**. We agree with your suggestion and performed the reaction by using compounds **1a**, **S20** and **2ea** under blue light irradiation. In this reaction we observed mixture of the products **76** and **81** with 3:1 ratio based on proton NMR (Both products are same R_F on TLC). We assume the addition reactivity of thiophenol is faster than the disulfide (**S20**). This might suggest the intermediate disulfide may form. However, benzenethiol-radical will attack ynamide (**1a**) faster than disulfide **S20**. We have provided ¹HNMR and ¹³C NMR spectra for your kind reference and included in the Supporting Information.

Supplementary Figure 22: ^1H NMR and ^{13}C NMR spectra of mixture of 76 and 81.

- Unfortunately, the language of the manuscript is still not up to the level required in Nature Communications. At times, this complicates the interpretation of the message of the authors. Some examples are shown below, but this list is not exhaustive.

Answer: We are sorry for the inconvenience caused in the English language. We took help from professional ACS English editing service and attached a proof of certificate as shown below for your kind reference.

A) *The items in the following list do not all correspond to the verb 'require' i.e. "the strategies require a lack of atom economy".*

“Despite their advantages, these limited existing radical strategies require expensive metals/photocatalysts, oxidants, longer reaction times, harmful waste production and a lack of atom economy.”

Answer: We are sorry for the inconvenience caused in the English language. We took help from professional ACS English editing service and also we follow your suggestions.

B) *The following sentence loses cohesion from ‘no regio- and chemoselectivity [...]’. Verbs are missing. Again, not all items in the list correspond to the verb ‘require’.*

“Nevertheless, despite the corresponding advances, the above process requires metal/Lewis acid, harsh reaction conditions, follows an intramolecular ionic path, and no regio- and chemoselectivity issues in the reaction and restricted to further functionalization of migrating group under mild reaction conditions.”

Suggestion:

“Nevertheless, despite the corresponding advances, the above process requires metal/Lewis acid, harsh reaction conditions, follows an intramolecular ionic path, no regio- or chemoselectivity issues are solved, and the reactions are restricted in terms of further functionalization of the migrating group.”

Answer: We are sorry for the inconvenience caused in the English language. We took help from professional ACS English editing service and also we follow your suggestions.

C) *The newly written parts suffer most from strange sentence structures, not limited to:*

“But, we did not observe specific wave length of blue LED light absorbance by compounds (1a, 2a, 2c, 2da, 2bd, 2ea). This suggests that blue LED light only induce radical generation.”

Suggestion:

“However, the absorbance of specific wavelengths of the blue LED light source by reactants **1a, 2a, 2c, 2da, 2bd, 2ea** was not observed, which suggests that blue LED light only induces radical generation.” (This of course depending on the outcome of the analysis as outlined earlier.)

Answer: We are sorry for the inconvenience caused in the English language. We took help from professional ACS English editing service and also we follow your suggestions.

D) In the following sentence, please use a consistent notation of sp² with the rest of the manuscript (as per the comment of another reviewer). Additionally, clarify that it is compound 3 that has the C-I bond, and 46 which has the C-Br bond.

“We note that, when we used 4-toluenesulfonyl bromide as a radical source, the product 4 did not form in the reaction due to higher bond energy (C_{sp²}-Br) than C_{sp²}-I bond in compound 3.”

Answer: We are sorry for the inconvenience caused. We used consistent in the notation of sp² in the entire manuscript and also in the Supporting Information. Actually, the compound **3** has the C-I bond and compound 46 has the C-Br bond. The sentence was corrected by ACS English editor and highlighted it yellow color in the manuscript.

E) Please use the past tense consistently in a paragraph.

“To prove homolytic bond fission in Ts-C(sp) (**99**), we choose **96** and **99** as starting material and performed the reaction but failed to get expected product [...].

Answer: We are sorry for the inconvenience caused in the English language. We took help from professional ACS English editing service and also we follow your suggestions.

SUPPORTING INFORMATION

- The past tense of ‘choose’ is ‘chose’. Please adjust on page S10 and where it is furthermore needed. Also change SP² to sp² on page S10.

Answer: We are sorry for the inconvenience caused. We follow your suggestion and corrected in the Supporting Information.

- There is an ‘S’ standing below the structure of iodobenzene in the figure of 5.3 on page S15.T.

Answer: We are thankful for your careful observation. We removed that extra S in the figure 5.3.

- Compound 2ca is called 2ac in the title of 5.8 in the SI, yet again 2ca in the scheme itself.

Answer: We are sorry for the inconvenience caused. We corrected in the revised Supporting Information.

- Add ‘S22’ to the title of 5.10.

Answer: We are thankful for your careful observation. We added the compound number at respective place in the revised Supporting Information.

- The text of page S24 mentions compound 67 while it is in fact compound 76.

Answer: We are thankful for your careful observation. Revised with correct compound number 76 in the revised Supporting Information.

- Has compound 41 been synthesized by using an isolated sulfonyl iodide or from the sulfonyl hydrazide? In the characterization, it mentions General Procedure E (sulfonyl hydrazide), but footnote b) is not placed at this compound in Figure 3 in the manuscript, and entry 41 is omitted in the title of section 7.2 in the SI.

Answer: We are sorry for the inconvenience caused. The compound **41** was synthesized by using Naphthalene-1-sulfonyl iodide. We have revised General Procedure from E to D in the Supporting Information and also included a footnote in Figure 3.

- The yields of the both reactions in section 7.4 is not corresponding to those reported in Figure 4b in the main manuscript! This for both reactions.

Answer: We are sorry for the inconvenience caused. We have revised according to manuscript yields in the revised Supporting information.

- In 10.1, the yield of the adduct 92 is not given in the supporting information. Additionally, a ‘-‘ is given instead of 0 % for the reaction with this radical trap.

Answer: We are thankful for your careful observation. We have included 0% yield in the respective place.

- In 10.2: typo in scheme: ‘observed’.

Answer: We are thankful for your careful observation. We corrected in the revised Supporting Information.

- In the characterization, the compounds **1a** and **1b** are indicated to be isolated ‘as a brown color’. Please exchange this by ‘as a brown colored solid’ or ‘as a brown solid’.

Answer: We are thankful for your suggestion. We have exchanged with ‘as a brown solid’ in the entire revised Supporting Information.

Reviewer: 2

The Authors have addressed the queries. The quality of the revised manuscript has been significantly improved. In my opinion, this fine piece of work is now suitable for publication in Nature Communications.

Answer: We thank this reviewer for supporting our work in the prestigious journal “*Nature communications*” as an article.

In summary, we feel all the suggestions and concerns from the reviewers have been carefully addressed or explained with experimental support. Finally, thank you very much again for the time you spend reviewing our manuscript.

Sincerely,

Prof. Jeh-Jeng Wang.

REVIEWER COMMENTS

Reviewer #1 (Remarks to the Author):

The manuscript was carefully reworked, especially in terms of language.

The following paragraph has not been corrected to the same level with respect to language as the rest of the manuscript. Presumably this has been added after language editing:

“The reaction tested with traditional heating at 28 °C (Supplementary Figs. 5-6) failed to produce product **3** (Table 1, entry 16) (~~observed both starting material~~ **both starting materials visible** on TLC). Next, **a** blue LED light source which has no second emission peak <400 nm also produced **product 3** in 80% yield (Table 1, entry 17) (Supplementary Figs. 4 and 8). According to our observations, the reaction speed and conversion is greatly enhanced by irradiation with visible light **to initiate the radical reaction**; an added advantage is that the reaction also gives a higher final conversion in a **shorter** reaction time.⁶²⁻⁶⁸”

Some mechanistic issues persist:

*The structures of **2da**, **2bd** and **2ea** were clarified in a reference. However, the authors once again state in the manuscript that “[...] the absorbance of specific wavelengths of the blue LED light source [...] was not observed, which suggests that blue LED light only induces radical generation.” In the rebuttal, they furthermore note that “We believe blue LED light source helps shorten the reaction time and also enhancement of the product yields by providing photo-energy in the cleavage of weak RSO₂-I bond in compound **2a** via homolytic bond fission”.

This seems to be a contradiction. While we acknowledge the observation that blue light (and not a rise in temperature) seems to cause a fast(er) reaction, it is not possible for a light source to provide photonic energy to cleave a bond if there is no absorption by a component in the mixture. The RSO₂-I is labile, but any increased fission due to light must be accompanied by the absorption of light. The reason why this is not observed is unclear, maybe these compounds are indeed too labile to accurately measure UV-vis absorption spectra? Nevertheless, for stable thiols, sulfonyl bromides and sulfonyl hydrazides there must be (a measurable ... can be very small) visible light absorption in order to be any effect of light. This aspect should be further clarified. If this is not the case (absorbing) maybe one needs to consider things like in situ formed other compounds such as PhSH delivering PhS-SPh with air splitted by light or complexes between these species (thiols, sulfonyl bromides and sulfonyl hydrazides) and other components present in the mixture absorbing light.

*The suggested scrambling reaction with disulfide in the presence of thiol was executed, and the incorporation of both thiol and disulfide was observed. This result suggests that disulfide can be a precursor of thiyl radicals under blue LED irradiation, a well-established phenomenon. The authors correctly write the equilibrium between the disulfide and its thiyl radical.

There is still reasonable doubt about the order of the thiyl radical formation: either thiol directly splits homolytically under blue LED absorption, or it first oxidizes towards disulfide under the present air atmosphere, which is known to absorb blue LED light, and forms an equilibrium with the thiyl radicals (see also paragraph above). The former manner seems unlikely as thiols do not feature absorption in the visible area and their S-H bond is a lot stronger than the labile SO₂-I bond. The latter possibility (disulfide) explains the absence of Blue LED light absorption of thiol **2ea**. To prove the direct involvement of thiol splitting an identical reaction under inert atmosphere can be conducted (carefully excluding all air avoiding the disulfide formation).

A POINT-BY-POINT RESPONSE TO REVIEWER COMMENTS

Dear Reviewer

Once again thank you very much for your time and efforts to review our manuscript. We have revised the manuscript according to your comments and suggestions. The corrections in detail were given in the revised manuscript and highlighted it in yellow color. We believe now the manuscript is suitable for publication.

Reviewer: 1

The manuscript was carefully reworked, especially in terms of language.

Answer: We are thankful for your appreciation and positive comments.

The following paragraph has not been corrected to the same level with respect to language as the rest of the manuscript. Presumably this has been added after language editing:

“The reaction tested with traditional heating at 28 °C (Supplementary Figs. 5-6) failed to produce product 3 (Table 1, entry 16) (~~observed both starting material~~ **both starting materials visible** on TLC). Next, **a** blue LED light source which has no second emission peak <400 nm also produced **ed** product 3 in 80% yield (Table 1, entry 17) (Supplementary Figs. 4 and 8). According to our observations, the reaction speed and conversion is greatly enhanced by irradiation with visible light ~~to initiate the radical reaction~~; an added advantage is that the reaction also gives a higher final conversion in a shorter **er** reaction time.⁶²⁻⁶⁸”

Answer: We are thankful for your kind help in the English language editing. We revised according to your suggestions and highlighted it yellow color in the revised manuscript.

Some mechanistic issues persist:

*The structures of **2da**, **2bd** and **2ea** were clarified in a reference. However, the authors once again state in the manuscript that “[...] the absorbance of specific wavelengths of the blue LED light source [...] was not observed, which suggests that blue LED light only induces radical generation.” In the rebuttal, they furthermore note that “We believe blue LED light source helps shorten the reaction time and also enhancement of the product yields by providing photo-energy in the cleavage of weak RSO₂-I bond in compound **2a** via homolytic bond fission“.

Answer: We are thankful for your query. Based on our careful UV-visible absorption studies in various molar concentration, different solvent and previous literature support, we have revised the manuscript text as shown below.

“[...] The spectra have marginally absorption of light was supported by UV-visible absorption spectra, which suggests that blue LED light activate the radical precursors by absorption of light”. Form this outcome, the blue LED light source helps shorten the reaction time and also enhancement of the product yields by providing photonic energy in homolytic bond fission of weak RSO₂-I bond in compound **2a**.

This seems to be a contradiction. While we acknowledge the observation that blue light (and not a rise in temperature) seems to cause a fast(er) reaction, it is not possible for a light source to provide photonic energy to cleave a bond if there is no absorption by a component in the mixture. The RSO₂-I is labile, but any increased fission due to light must be accompanied by the absorption of light. The reason why this is not observed is unclear, maybe these compounds are indeed too labile to accurately measure UV-vis absorption spectra? Nevertheless, for stable thiols, sulfonyl bromides and sulfonyl hydrazides there must be (a measurable ... can be very small) visible light absorption in order to be any effect of light. This aspect should be further clarified. If this is not the case (absorbing) maybe one needs to consider things like in situ formed other compounds such as PhSH delivering PhS-SPh with air splitted by light or complexes between these species (thiols, sulfonyl bromides and sulfonyl hydrazides) and other components present in the mixture absorbing light.

Answer: We are thankful for your valuable suggestions and details explanation to find the role of light in this reaction transformation. After considering your valuable suggestions and careful literature search (**The Journal of Physical Chemistry, Vol. 82, No. 3, 1978**)

Studies of Sulfonyl Radicals. 4. Flash Photolysis of Aromatic Sulfones

Ho Huu Thoi, Osamu Ito, Masashi Iino, and Minoru Matsuda*

Chemical Research Institute of Non-Aqueous Solutions, Tohoku University, Sendai 980, Japan (Received May 25, 1977)

Publication costs assisted by the Chemical Research Institut of Non-Aqueous Solutions

The flash photolysis of arenesulfonyl iodides and some other aromatic sulfones showed a transient absorption spectrum with a maximum near 330 nm and a continuous band up to 500 nm which was attributed to the arenesulfonyl radical. No wavelength shift in the absorption maximum was observed for substituted benzenesulfonyl radicals. The second-order decay rates of the radicals are shown to be diffusion controlled and the rate constants are about $5 \times 10^9 \text{ M}^{-1} \text{ s}^{-1}$.

The Journal of Physical Chemistry, Vol. 82, No. 3, 1978 (Copied from this article)

The RSO₂-I is labile but can measure the UV-vis absorption spectra as shown in above article. Considering above facts, we have checked the absorption spectra of the compound **2a** in different molar concentration as well as different solvents (MeCN, THF and DCM) as shown below. In higher molar concentration of the solvent we

observed better absorption (The peak shape is not good below 10^{-3} M concentration but for better understanding we included in the Supporting Information along with 10^{-2} M concentration UV-vis absorption spectra for all of the radical precursors) compared to lower concentration of the solvent (Previously the UV-vis absorption spectra were recorded by using less molar concentration (10^{-5} M of DCM). In this concentration, there is a very minor absorption from radical precursor but unable to notice in the previous UV-vis absorption spectra). For your kind information, our standard reaction condition is very concentrated. We chose 0.05 M solvent based on ynamides **1** in the reaction condition.

Absorption spectra of 4-methylbenzenesulfonyl iodide (**2a**) in DCM solvent

Supplementary Figure 14. Absorption spectra of 4-methylbenzenesulfonyl iodide (**2a**) (10^{-3} M) in DCM, and blue LED emission.

Supplementary Figure 15. Absorption spectra of 4-methylbenzenesulfonyl iodide (**2a**) (10^{-4} M) in DCM, and blue LED emission.

Supplementary Figure 16. Absorption spectra of 4-methylbenzenesulfonyl iodide (**2a**) (10^{-5} M) in DCM, and blue LED emission.

Absorption spectra of 4-methylbenzenesulfonyl iodide (2a) in MeCN solvent

Supplementary Figure 18. Absorption spectra of 4-methylbenzenesulfonyl iodide (**2a**) (10^{-3} M) in MeCN, and blue LED emission.

Supplementary Figure 19. Absorption spectra of 4-methylbenzenesulfonyl iodide (**2a**) (10^{-4} M) in MeCN, and blue LED emission.

Supplementary Figure 20. Absorption spectra of 4-methylbenzenesulfonyl iodide (**2a**) (10^{-5} M) in MeCN, and blue LED emission.

Absorption spectra of 4-methylbenzenesulfonyl iodide (2a) in THF solvent

Supplementary Figure 22. Absorption spectra of 4-methylbenzenesulfonyl iodide (**2a**) (10^{-3} M) in THF, and blue LED emission.

Supplementary Figure 23. Absorption spectra of 4-methylbenzenesulfonyl iodide (**2a**) (10^{-4} M) in THF, and blue LED emission.

Supplementary Figure 24. Absorption spectra of 4-methylbenzenesulfonyl iodide (**2a**) (10^{-5} M) in THF, and blue LED emission.

Based on the above UV-vis absorption spectra results of the compound **2a** other radical precursors (**2ca**, **2da**, **2bd** and **2ea**) absorption spectra were recorded by using DCM solvent (10^{-2} to 10^{-5} M). All the radical precursors have marginally absorption of blue LED light, which suggests that blue LED light induces the radical generation from radical precursor by photonic energy. But compound **1a** didn't not have absorption at specific wavelength of blue light.

*The suggested scrambling reaction with disulfide in the presence of thiol was executed, and the incorporation of both thiol and disulfide was observed. This result suggests that disulfide can be a precursor of thiyl radicals under blue LED irradiation, a well-established phenomenon. The authors correctly write the equilibrium between the disulfide and its thiyl radical.

Answer: We are sorry for the mistake. Corrected the equilibrium between the disulfide and its thiyl radical. Please check in the revised Supporting Information.

There is still reasonable doubt about the order of the thiyl radical formation: either thiol directly splits homolytically under blue LED absorption, or it first oxidizes towards disulfide under the present air atmosphere, which is known to absorb blue LED light, and forms an equilibrium with the thiyl radicals (see also paragraph above). The former manner seems unlikely as thiols do not feature absorption in the visible area and their S-H bond is a lot stronger than the labile $\text{SO}_2\text{-I}$ bond. The latter possibility (disulfide) explains the absence of Blue

LED light absorption of thiol **2ea**. To prove the direct involvement of thiol splitting an identical reaction under inert atmosphere can be conducted (carefully excluding all air avoiding the disulfide formation).

Answer: We are thankful for your query and valuable suggestions. According to your query an identical reaction was performed by using compounds **1a**, **S20** and **2ea** in nitrogen atmosphere under blue light irradiation. In this reaction also observed mixture of the products **76** and **81** with 6.3:1 ratio based on proton NMR. With this outcome, the first step is thiyl radical generation and it may form minor disulfide equilibrium with thiyl radical. The below shown pictures are reaction setup and handling. We have provided ^1H NMR and ^{13}C NMR spectra for your kind reference and included in the Supporting Information.

Supplementary Figure 48: 1H NMR and ^{13}C NMR spectra of mixture of 76 and 81.

In summary, we believe all the suggestions and concerns from this reviewer have been carefully addressed or explained with experimental support. Finally, thank you very much again for the time you spend reviewing our manuscript.

Sincerely,

Prof. Jeh-Jeng Wang.

REVIEWER COMMENTS

Reviewer #1 (Remarks to the Author):

The results of the scrambling experiment under inert atmosphere and additional UV-vis absorption spectra were provided. The mechanistic issues are largely resolved. One issue remains:

For sulfonyl iodide **2a** there is now a clear link between the performance of the reaction in both DCM, MeCN and THF, based upon the more concentrated absorption spectra in each of these solvents. Absorption of sulfonyl bromide **2ca** in the blue area is for some reason more pronounced in the low concentrated sample, but nevertheless also present, explaining the result in **Figure 4a**. Selenosulfonate **2da** shows absorption as well. The authors furthermore remark that the reaction takes place in an even more concentrated (0.05 M based on **1**) solution than used for UV/VIS measurement, which supports the (sometimes) low absorption features of the sulfur containing reactants measured but presumably relevant at high concentration in the reactions. However, the absorption of the thiol **2ea** is very low (or non-existent?) in DCM, even in the 10^{-2} M sample. This is not surprising, as the reaction only yielded traces of the desired product (Section 6 of the SI) in DCM. In order to support homolytic fission, as supported by the conducted scrambling experiments, please provide a comparable absorption spectrum of **2ea** in MeCN (10^{-2} M) in which you are performing your actual reactions.

A POINT-BY-POINT RESPONSE TO REVIEWER COMMENTS

Dear Reviewer

We thank and greatly appreciate for your efforts and time to review our manuscript. We have revised the manuscript according to your comments and suggestions. The corrections in detail were given in the revised manuscript and highlighted it in yellow color. We have addressed all remaining points from this reviewer and that the revised manuscript will be suitable for publication in *Nature Communications*.

Reviewer: 1

The results of the scrambling experiment under inert atmosphere and additional UV-vis absorption spectra were provided. The mechanistic issues are largely resolved. One issue remains:

Answer: We are thankful for your appreciation and positive comments.

For sulfonyl iodide **2a** there is now a clear link between the performance of the reaction in both DCM, MeCN and THF, based upon the more concentrated absorption spectra in each of these solvents. Absorption of sulfonyl bromide **2ca** in the blue area is for some reason more pronounced in the low concentrated sample, but nevertheless also present, explaining the result in **Figure 4a**. Selenosulfonate **2da** shows absorption as well. The authors furthermore remark that the reaction takes place in an even more concentrated (0.05 M based on **1**) solution than used for UV/VIS measurement, which supports the (sometimes) low absorption features of the sulfur containing reactants measured but presumably relevant at high concentration in the reactions.

Answer: We are thankful for your suggestions. We revised according to your suggestions and highlighted it yellow color in manuscript. We included in the text description of **Figure 4a** “The UV-vis absorption spectra of sulfonyl iodide **2a** in various solvents (DCM, MeCN and THF in 10^{-2} M to 10^{-5} M) were checked (Supplementary Figs. 13-24). In more concentration solution observed better absorption in each of these solvents in blue LED area (our actual reaction condition is even more concentrated solution than used for UV/VIS measurement. In case of sulfonyl bromide **2ca** have marginal absorption in the blue LED area in various concentration (10^{-2} M to 10^{-5} M in DCM) (Supplementary Figs. 25-28)”. (We cross checked and provided new absorption spectra for **2ca**).

However, the absorption of the thiol **2ea** is very low (or non-existent?) in DCM, even in the 10^{-2} M sample. This is not surprising, as the reaction only yielded traces of the desired product (Section 6 of the SI) in DCM. In order to support homolytic fission, as supported by the conducted scrambling experiments, please provide a

comparable absorption spectrum of **2ea** in MeCN (10^{-2} M) in which you are performing your actual reactions.

Answer: We are thankful for your query. For your kind information, we used 0.1 M of MeCN in our standard reaction condition based on compound **1** (In that 2.5 of equivalence thiol was used). As per your query, compound **2ea** absorption spectra in acetonitrile (10^{-1} M to 10^{-5} M) were checked as shown below Supplementary Figs. We observed marginal absorption in the blue LED area (Supplementary Figs. 36-40). We assume that polar aprotic solvent acetonitrile would stabilize the radical species (thiyl) which may lower the activation energy and faster the reaction to produce the product compared to dichloromethane (This text included in Supplementary Table 1 description). Absorption spectra of **2ea** in MeCN (Supplementary Figs. 36-40) and in DCM (Supplementary Figs. 41-44) were attached in the Supporting Information.

9.2.5. Absorption spectra of benzenethiol (2ea) in MeCN

Supplementary Figure 36. Absorption spectra of benzenethiol (**2ea**) (10^{-1} M) in MeCN, blue LED emission and zoom appearance

Supplementary Figure 37. Absorption spectra of benzenethiol (**2ea**) (10^{-2} M) in MeCN, blue LED emission and zoom appearance

Supplementary Figure 38. Absorption spectra of benzenethiol (**2ea**) (10^{-3} M) in MeCN, blue LED emission and zoom appearance

Supplementary Figure 39. Absorption spectra of benzenethiol (**2ea**) (10^{-4} M) in MeCN, blue LED emission and zoom appearance

Supplementary Figure 40. Absorption spectra of benzenethiol (**2ea**) (10^{-5} M) in MeCN, blue LED emission and zoom appearance

In summary, we believe all the suggestions and concerns from this reviewer have been carefully addressed or explained with experimental support. Hoping to have addressed all remaining points raised by the reviewer and that this revised manuscript will be suitable for publication in *Nature Communications*.

Sincerely,

Prof. Jeh-Jeng Wang.

REVIEWER COMMENTS

Reviewer #1 (Remarks to the Author):

The authors added the required spectra and a marginal absorption of thiophenol in MeCN was observed. The manuscript is ready for publication in Nature Communication, provided language correction of the following two paragraphs is performed:

- The UV-vis absorption spectra of sulfonyl iodide **2a** in various solvents (DCM, MeCN and THF in 10^{-2} M to 10^{-5} M) were ~~measured~~ ~~checked~~ (Supplementary Figs. 13-24). In more concentrated ~~solutions~~, ~~observed~~ better absorption in each of these solvents ~~was observed in the under~~ blue LED area (our actual reaction condition is even more concentrated ~~solution~~ than ~~the samples~~ used for UV/~~VIS~~ -vis measurement). In case of sulfonyl bromide **2ca**, ~~have~~ marginal absorption in the blue LED area in various concentrations (10^{-2} M to 10^{-5} M in DCM ~~and MeCN~~) ~~was observed~~ (Supplementary Figs. 25-28).
- The UV-vis absorption spectra ~~show have~~ marginally ~~absorption of~~ blue light ~~was supported by UV-visible absorption spectra~~, which suggests that blue LED light activates the radical precursors ~~by absorption of light~~. In case of compound 4-methyl-N-(phenylethynyl)-N-(2-(phenylethynyl)phenyl)benzenesulfonamide (**1a**), UV-vis absorption was not observed (Supplementary Figs. 45-48), which suggests that blue LED light only activates the radical precursors but not ynamides (**1**).

A POINT-BY-POINT RESPONSE TO REVIEWER COMMENTS

The authors added the required spectra and a marginal absorption of thiophenol in MeCN was observed. The manuscript is ready for publication in Nature Communication, provided language correction of the following two paragraphs is performed:

Answer: We thanks this reviewer for accepting our work in the prestigious journal “*Nature communications*” .

- The UV-vis absorption spectra of sulfonyl iodide **2a** in various solvents (DCM, MeCN and THF in 10^{-2} M to 10^{-5} M) were ~~measured checked~~ (Supplementary Figs. 13-24). In more concentrated ~~ion~~ solutions, ~~observed~~ better absorption in each of these solvents ~~was observed in the~~ under blue LED area (our actual reaction condition is even more concentrated solution than the samples used for UV/~~VIS~~ -vis measurement). In case of sulfonyl bromide **2ca**, ~~have~~-marginal absorption in the blue LED area in various concentrations (10^{-2} M to 10^{-5} M in DCM and MeCN) ~~was observed~~ (Supplementary Figs. 25-28).

Answer: We are thankful for your help in the English editing. We revised according to your suggestions.

- The UV-vis absorption spectra ~~show have~~ marginally absorption of blue light ~~was supported by UV-visible absorption spectra~~, which suggests that blue LED light activates the radical precursors ~~by absorption of light~~. In case of compound 4-methyl-N-(phenylethynyl)-N-(2-(phenylethynyl)phenyl)benzenesulfonamide (1a), UV-vis absorption was not observed (Supplementary Figs. 45-48), which suggests that blue LED light only activates the radical precursors but not ynamides (1).

Answer: We are thankful for your help in the English editing. We revised according to your suggestions.